# Near-Optimal Goal-Oriented Reinforcement Learning in Non-Stationary Environments

**Liyu Chen**
University of Southern California
liyuc@usc.edu

**Haipeng Luo**
University of Southern California
haipengl@usc.edu

## Abstract

We initiate the study of dynamic regret minimization for goal-oriented reinforcement learning modeled by a non-stationary stochastic shortest path problem with changing cost and transition functions. We start by establishing a lower bound $\Omega((B_\star SAT_\star(\Delta_c + B_\star^2 \Delta_P))^{1/3}K^{2/3})$, where $B_\star$ is the maximum expected cost of the optimal policy of any episode starting from any state, $T_\star$ is the maximum hitting time of the optimal policy of any episode starting from the initial state, $SA$ is the number of state-action pairs, $\Delta_c$ and $\Delta_P$ are the amount of changes of the cost and transition functions respectively, and $K$ is the number of episodes. The different roles of $\Delta_c$ and $\Delta_P$ in this lower bound inspire us to design algorithms that estimate costs and transitions separately. Specifically, assuming the knowledge of $\Delta_c$ and $\Delta_P$, we develop a simple but sub-optimal algorithm and another more involved minimax optimal algorithm (up to logarithmic terms). These algorithms combine the ideas of finite-horizon approximation [Chen et al., 2022a], special Bernstein-style bonuses of the MVP algorithm [Zhang et al., 2020], adaptive confidence widening [Wei and Luo, 2021], as well as some new techniques such as properly penalizing long-horizon policies. Finally, when $\Delta_c$ and $\Delta_P$ are unknown, we develop a variant of the MASTER algorithm [Wei and Luo, 2021] and integrate the aforementioned ideas into it to achieve $\tilde{\mathcal{O}}(\min\{B_\star S\sqrt{ALK}, (B_\star^2 S^2 AT_\star(\Delta_c + B_\star \Delta_P))^{1/3}K^{2/3}\})$ regret, where $L$ is the unknown number of changes of the environment.

## 1 Introduction

Goal-oriented reinforcement learning studies how to achieve a certain goal with minimal total cost in an unknown environment via sequential interactions. It has often been modeled as online learning in an episodic Stochastic Shortest Path (SSP) model, where in each episode, starting from a fixed initial state, the learner sequentially takes an action, suffers a cost, and transits to the next state, until the goal state is reached. The performance of the learner can be measured by her *regret*, generally defined as the difference between her total cost and that of a sequence of benchmark policies (one for each episode).

Despite the recent surge of studies on this problem, all previous works consider minimizing *static regret*, a special case where the benchmark policy is the same for every episode. This is reasonable only for (near) stationary environments where one single policy performs well over all episodes. In reality, however, the environment is often non-stationary with both the cost function and the transition function changing over episodes, making static regret an unreasonable metric. Instead, the desired objective is to minimize *dynamic regret*, where the benchmark policy for each episode is the optimal policy for that corresponding environment, and the hope is to obtain sublinear dynamic regret whenever the non-stationarity is not too large.

36th Conference on Neural Information Processing Systems (NeurIPS 2022).

Based on this motivation, we initiate the study of dynamic regret minimization for non-stationary SSP and develop the first set of results. Specifically, our contributions are as follows:

- To get a sense on the difficulty of the problem, we start by establishing a dynamic regret lower bound in Section 3. Specifically, we prove that $\Omega((B_\star SAT_\star(\Delta_c + B_\star^2\Delta_P))^{1/3}K^{2/3})$ regret is unavoidable, where $B_\star$ is the maximum expected cost of the optimal policy of any episode starting from any state, $T_\star$ is the maximum hitting time of the optimal policy of any episode starting from the initial state, $S$ and $A$ are the number of states and actions respectively, $\Delta_c$ and $\Delta_P$ are the amount of changes of the cost and transition functions respectively, and $K$ is the number of episodes. Note the different roles of $\Delta_c$ and $\Delta_P$ here — the latter is multiplied with an extra $B_\star^2$ factor, which we find surprising for a technical reason discussed in Section 3. More importantly, this inspires us to estimate costs and transitions independently in subsequent algorithm design.

- For algorithms, we first present a simple one (Algorithm 2 in Section 5) that achieves sub-optimal regret of $\tilde{\mathcal{O}}((B_\star SAT_{\max}(\Delta_c + B_\star^2\Delta_P))^{1/3}K^{2/3})$, where $T_{\max} \geq T_\star$ is the maximum hitting time of the optimal policy of any episode starting from any state. Except for replacing $T_\star$ with the larger quantity $T_{\max}$, this bound is optimal in all other parameters. Moreover, this also translates to a minimax optimal regret bound in the finite-horizon setting (a special case of SSP), making Algorithm 2 the first model-based algorithm with the optimal $(SA)^{1/3}$ dependency.

- To improve the $T_{\max}$ dependency to $T_\star$, in Section 6, we present a more involved algorithm (Algorithm 4) that achieves a near minimax optimal regret bound matching the earlier lower bound up to logarithmic terms.

- Both algorithms above require the knowledge of $\Delta_c$ and $\Delta_P$. Moreover, for a special kind of non-stationary environments where the cost/transition function only changes $L$ times, they are not able to achieve a more favorable dynamic regret bound of the form $\sqrt{LK}$. To overcome these issues altogether, in Section 7, we develop a variant of the MASTER algorithm [Wei and Luo, 2021] and integrate the earlier algorithmic ideas into it, which finally leads to a (sub-optimal) $\tilde{\mathcal{O}}(\min\{B_\star S\sqrt{ALK}, (B_\star^2 S^2 AT_\star(\Delta_c + B_\star\Delta_P))^{1/3}K^{2/3}\})$ regret bound without knowing the non-stationarity $\Delta_c$, $\Delta_P$, or $L$.

**Techniques**   All our algorithms are built on top of a finite-horizon approximation scheme first proposed by Cohen et al. [2021] and later improved by Chen et al. [2022a]; see Section 4. Both the sub-optimal Algorithm 2 and the optimal Algorithm 4 are then developed based on ideas from the MVP algorithm [Zhang et al., 2020] (for the finite-horizon setting), which adopts a UCBVI-style update rule [Azar et al., 2017] with a special Bernstein-style bonus term. The sub-optimal algorithm further integrates the idea of adaptive confidence widening [Wei and Luo, 2021] into the UCBVI-style update by subtracting a bias from the cost function uniformly over all state-action pairs, which helps control the magnitude of the estimated value function. The minimax optimal algorithm, on the other hand, adds a positive correction term to the cost function to penalize long-horizon policies, which helps improve the $T_{\max}$ dependency to $T_\star$. It also incorporates several non-stationarity tests to ensure that the algorithm resets its knowledge of the environment when the amount of non-stationarity is large. Both algorithms maintain (update and reset) cost and transition estimation independently, which is the key to achieve the correct $B_\star$ dependency for both the $\Delta_c$-related and $\Delta_P$-related terms.

To handle unknown non-stationarity, we adopt the idea of the MASTER algorithm from [Wei and Luo, 2021]. Although the nature of MASTER is a blackbox reduction, we cannot apply it directly due to the presence of the correction term that changes continuously and brings extra challenges in tracking the learner's performance. We handle this by redesigning the first non-stationarity test of the MASTER algorithm. Specifically, we maintain multiple running averages of the estimated value function to detect different levels of non-stationarity.

**Related Work**   Static regret minimization in SSP has been heavily studied in recent years, for both stochastic costs [Tarbouriech et al., 2020, Cohen et al., 2020, 2021, Tarbouriech et al., 2021, Chen et al., 2021a, Jafarnia-Jahromi et al., 2021, Vial et al., 2021, Min et al., 2021, Chen et al., 2022a] and adversarial costs [Rosenberg and Mansour, 2021, Chen et al., 2021b, Chen and Luo, 2021, Chen et al., 2022b]. To the best of our knowledge, we are the first to study dynamic regret for non-stationary SSP.

There is also a surge of studies on online learning in non-stationary environments, ranging from bandits [Auer et al., 2019, Chen et al., 2019, 2021c, Russac et al., 2020, Faury et al., 2021, Abbasi-Yadkori et al., 2022, Suk and Kpotufe, 2021] to reinforcement learning [Gajane et al., 2018, Ortner

et al., 2020, Cheung et al., 2020, Fei et al., 2020, Mao et al., 2021, Zhou et al., 2020, Touati and Vincent, 2020, Domingues et al., 2021, Wei and Luo, 2021, Ding and Lavaei, 2022, Lykouris et al., 2021, Wei et al., 2022]. Compared to previous work, the model we study is quite general and subsumes multi-armed bandit and finite-horizon reinforcement learning. On the other hand, it also introduces extra and unique challenges as we will discuss.

## 2  Preliminaries

A non-stationary SSP instance consists of state space $\mathcal{S}$, action space $\mathcal{A}$, initial state $s_{\text{init}} \in \mathcal{S}$, goal state $g \notin \mathcal{S}$, a set of cost mean functions $\{c_k\}_{k=1}^K$ with $c_k \in [0,1]^{\mathcal{S} \times \mathcal{A}}$, and a set of transition functions $\{P_k\}_{k=1}^K$ with $P_k = \{P_{k,s,a}\}_{(s,a) \in \mathcal{S} \times \mathcal{A}}$ and $P_{k,s,a} \in \Delta_{\mathcal{S}_+}$, where $\mathcal{S}_+ = \mathcal{S} \cup \{g\}$, $\Delta_{\mathcal{S}_+}$ is the simplex over $\mathcal{S}_+$, and $K$ is the number of episodes. The set of cost and transition functions are unknown to the learner and determined by the environment before learning starts.

The learning protocol is as follows: the learner interacts with the environment for $K$ episodes. In episode $k$, starting from the initial state $s_{\text{init}}$, the learner sequentially takes an action, incurs a cost, and transits to the next state until reaching the goal state. We denote by $(s_i^k, a_i^k, c_i^k, s_{i+1}^k)$ the $i$-th state-action-cost-afterstate tuple observed in episode $k$, where $c_i^k$ is sampled from an unknown distribution with support $[0,1]$ and mean $c_k(s_i^k, a_i^k)$, and $s_{i+1}^k$ is sampled from $P_{k,s_i^k,a_i^k}$. We denote by $I_k$ the total number of steps in episode $k$, such that $s_{I_k+1}^k = g$.

**Learning Objective**   Intuitively, in each episode the learner aims at finding a policy that minimizes the total cost of reaching the goal state. Formally, a policy $\pi \in \mathcal{A}^{\mathcal{S}}$ assigns an action $\pi(s)$ to each state $s \in \mathcal{S}$, and its expected cost for episode $k$ starting from a state $s$ is denoted as $V_k^\pi(s) = \mathbb{E}\left[\sum_{i=1}^{I_k} c_k(s_i^k, \pi(s_i^k)) | P_k, s_1^k = s\right]$ where the expectation is with respect to the randomness of next states $s_{i+1}^k \sim P_{k,s_i^k,\pi(s_i^k)}$ and the number of steps $I_k$ before reaching $g$. The optimal policy $\pi_k^\star$ for episode $k$ is then the policy that minimizes $V_k^\pi(s)$ for all $s$. Using $V_k^\star$ as a shorthand for $V_k^{\pi_k^\star}$, we formally define the dynamic regret of the learner as

$$R_K = \sum_{k=1}^K \left( \sum_{i=1}^{I_k} c_i^k - V_k^\star(s_{\text{init}}) \right).$$

When $I_k = \infty$ for some $k$, we let $R_K = \infty$.

**Remark 1.** *Note that our learning setting does not fall into the general non-stationary reinforcement learning framework in [Wei and Luo, 2021]. In their framework, they fix a policy to play throughout an episode, and the cost incurs by any policy is bounded. While in our case, the learner may follow several different policies within an episode. This is necessary since under unknown and changing transition, the learner may not be able to identify a proper policy (which reaches the goal state with probability $1$) at the beginning of an episode, and committing to a single policy within an episode may lead to infinite regret.*

Several parameters play a key role in characterizing the difficulty of this problem: $B_\star = \max_{k,s} V_k^\star(s)$, the maximum cost of the optimal policy of any episode starting from any state; $T_\star = \max_k T_k^{\pi_k^\star}(s_{\text{init}})$ (where $T_k^\pi(s)$ is expected number of steps it takes for policy $\pi$ to reach the goal in episode $k$ starting from state $s$), the maximum hitting time of the optimal policy of any episode starting from the initial state; $T_{\max} = \max_{k,s} T_k^{\pi_k^\star}(s)$, the maximum hitting time of the optimal policy of any episode starting from any state; $\Delta_c = \sum_{k=1}^{K-1} \|c_{k+1} - c_k\|_\infty$, the amount of non-stationarity in the cost functions; and finally $\Delta_P = \sum_{k=1}^{K-1} \max_{s,a} \|P_{k+1,s,a} - P_{k,s,a}\|_1$, the amount of non-stationarity in the transition functions. Throughout the paper we assume the knowledge of $B_\star$, $T_\star$, and $T_{\max}$, and also $B_\star \geq 1$ for simplicity. $\Delta_c$ and $\Delta_P$ are assumed to be known for the first two algorithms we develop, but unknown for the last one.

**Other Notations**   For a value function $V \in \mathbb{R}^{\mathcal{S}_+}$ and a distribution $P$ over $\mathcal{S}_+$, define $PV = \mathbb{E}_{s' \sim P}[V(s')]$ (mean) and $\mathbb{V}(P,V) = \mathbb{E}_{s' \sim P}[V(s')^2] - (PV)^2$ (variance). Let $S = |\mathcal{S}|$ and $A = |\mathcal{A}|$ be the number of states and actions respectively. The notation $\tilde{\mathcal{O}}(\cdot)$ hides all logarithmic dependency including $\ln K$ and $\ln \frac{1}{\delta}$ for some failure probability $\delta \in (0,1)$. Also define a value function upper bound $B = 16B_\star$. For integers $s$ and $e$, we define $[s,e] = \{s, s+1, \ldots, e\}$ and $[e] = \{1, \ldots, e\}$.

---

**Algorithm 1** Finite-Horizon Approximation of SSP

---

**Input:** Algorithm $\mathfrak{A}$ for finite-horizon MDP $\mathring{\mathcal{M}}$ with horizon $H = 4T_{\max} \ln(8K)$.
**Initialize:** interval counter $m \leftarrow 1$.
**for** $k = 1, \ldots, K$ **do**
  1     Set $s_1^m \leftarrow s_{\mathrm{init}}$.
  2     **while** $s_1^m \neq g$ **do**
  3         Feed initial state $s_1^m$ to $\mathfrak{A}$, $h \leftarrow 1$.
  4         **while** *True* **do**
  5             Receive action $a_h^m$ from $\mathfrak{A}$, play it, and observe cost $c_h^m$ and next state $s_{h+1}^m$.
  6             Feed $c_h^m$ and $s_{h+1}^m$ to $\mathfrak{A}$.
  7             **if** $h = H$ *or* $s_{h+1}^m = g$ *or* $\mathfrak{A}$ *requests to start a new interval* **then**
  8                 $H_m \leftarrow h$. **break**.
  9             **else** $h \leftarrow h + 1$.
  10         Set $s_1^{m+1} = s_{H_m+1}^m$ and $m \leftarrow m + 1$.

---

## 3   Lower Bound

To better understand the difficulty of learning non-stationary SSP, we first establish the following dynamic regret lower bound.

**Theorem 1.** *In the worst case, the learner's regret is at least* $\Omega((B_\star S A T_\star (\Delta_c + B_\star^2 \Delta_P))^{1/3} K^{2/3})$.

The lower bound construction is similar to that in [Mao et al., 2021], where the environment is piecewise stationary. In each stationary period, the learner is facing a hard SSP instance with a slightly better hidden state. Details are deferred to Appendix B.2.

In a technical lemma in Appendix B.1, we show that for any two episodes $k_1$ and $k_2$, the change of the optimal value function due to non-stationarity satisfies $V_{k_1}^\star(s_{\mathrm{init}}) - V_{k_2}^\star(s_{\mathrm{init}}) \leq (\Delta_c + B_\star \Delta_P) T_\star$, with only one extra $B_\star$ factor for the $\Delta_P$-related term. We thus find our lower bound somewhat surprising since an extra $B_\star^2$ factor shows up for the $\Delta_P$-related term. This comes from the fact that constructing the hard instance with perturbed costs requires a larger amount of perturbation compared to that with perturbed transitions; see Theorem 7 and Theorem 8 for details.

More importantly, this observation implies that simply treating these two types of non-stationarity as a whole and only consider the non-stationarity in value function as done in [Wei and Luo, 2021] does not give the right $B_\star$ dependency. This further inspires us to consider cost and transition estimation independently in our subsequent algorithm design.

## 4   Basic Framework: Finite-Horizon Approximation

Our algorithms are all built on top of the finite-horizon approximation scheme of [Cohen et al., 2021], whose analysis is greatly simplified and improved by [Chen et al., 2022a], making it applicable to our non-stationary setting as well. This scheme makes use of an algorithm $\mathfrak{A}$ that deals with a special case of SSP where each episode ends within $H = \tilde{\mathcal{O}}(T_{\max})$ steps, and applies it to the original SSP following Algorithm 1. Specifically, call each "mini-episode" $\mathfrak{A}$ is facing an *interval*. At each step $h$ of interval $m$, the learner receives the decision $a_h^m$ from $\mathfrak{A}$, takes this action, observes the cost $c_h^m$, transits to the next state $s_{h+1}^m$, and then feed the observation $c_h^m$ and $s_{h+1}^m$ to $\mathfrak{A}$ (Line 5 and Line 6). The interval $m$ ends whenever one of the following happens (Line 7): the goal state is reached, $H$ steps have passed, or $\mathfrak{A}$ requests to start a new interval.[1] In the first case, the initial state $s_1^{m+1}$ of the next interval $m + 1$ will be set to $s_{\mathrm{init}}$, while in the other two cases, it is naturally set to the learner's current state, which is also $s_{H_m+1}^m$ where $H_m$ is the length of interval $m$ (see Line 10). At the end of each interval, we artificially let $\mathfrak{A}$ suffer a *terminal cost* $c_f(s_{H_m+1}^m)$ where $c_f(s) = 2B_\star \mathbb{I}\{s \neq g\}$.

---

[1]This last condition is not present in prior works. We introduce it since later our instantiation of $\mathfrak{A}$ will change its policy in the middle of an interval, and creating a new interval in this case allows us to make sure that the policy in each interval is always fixed, which simplifies the analysis.

---

**Algorithm 2** Non-Stationary MVP

---

**Parameters:** window sizes $W_c$ (for costs) and $W_P$ (for transitions), and failure probability $\delta$.

**Initialize:** for all $(s, a, s')$, $\mathbf{C}(s, a) \leftarrow 0$, $\mathbf{M}(s, a) \leftarrow 0$, $\mathbf{N}(s, a) \leftarrow 0$, $\mathbf{N}(s, a, s') \leftarrow 0$.

**Initialize:** Update(1).

**for** $m = 1, \ldots, M$ **do**

 **for** $h = 1, \ldots, H$ **do**

1  Play action $a_h^m \leftarrow \operatorname{argmin}_a Q_h(s_h^m, a)$, receive cost $c_h^m$ and next state $s_{h+1}^m$.

  $\mathbf{C}(s_h^m, a_h^m) \overset{+}{\leftarrow} c_h^m$, $\mathbf{M}(s_h^m, a_h^m) \overset{+}{\leftarrow} 1$, $\mathbf{N}(s_h^m, a_h^m) \overset{+}{\leftarrow} 1$, $\mathbf{N}(s_h^m, a_h^m, s_{h+1}^m) \overset{+}{\leftarrow} 1$.[2]

2  **if** $s_{h+1}^m = g$ *or* $\mathbf{M}(s_h^m, a_h^m) = 2^l$ *or* $\mathbf{N}(s_h^m, a_h^m) = 2^l$ *for some integer* $l \geq 0$ **then**

   **break** (which starts a new interval).

3 **if** $W_c$ *divides* $m$ **then** reset $\mathbf{C}(s, a) \leftarrow 0$ and $\mathbf{M}(s, a) \leftarrow 0$ for all $(s, a)$.

4 **if** $W_P$ *divides* $m$ **then** reset $\mathbf{N}(s, a, s') \leftarrow 0$ and $\mathbf{N}(s, a) \leftarrow 0$ for all $(s, a, s')$.

 Update($m + 1$).

**Procedure** Update($m$)

 $V_{H+1}(s) \leftarrow 2B_\star \mathbb{I}\{s \neq g\}$, $V_h(g) \leftarrow 0$ for $h \leq H$, $\iota \leftarrow 2^{11} \cdot \ln\left(\frac{2SAHKm}{\delta}\right)$, and $x \leftarrow \frac{1}{mH}$.

 **for** *all* $(s, a)$ **do**

  $\mathbf{N}^+(s, a) \leftarrow \max\{1, \mathbf{N}(s, a)\}$, $\mathbf{M}^+(s, a) \leftarrow \max\{1, \mathbf{M}(s, a)\}$, $\bar{c}(s, a) \leftarrow \frac{\mathbf{C}(s, a)}{\mathbf{M}^+(s, a)}$,

  $\widehat{c}(s, a) \leftarrow \max\left\{0, \bar{c}(s, a) - \sqrt{\frac{\bar{c}(s, a)\iota}{\mathbf{M}^+(s, a)}} - \frac{\iota}{\mathbf{M}^+(s, a)}\right\}$, $\bar{P}_{s, a}(\cdot) \leftarrow \frac{\mathbf{N}(s, a, \cdot)}{\mathbf{N}^+(s, a)}$.

 **while** *True* **do**

  **for** $h = H, \ldots, 1$ **do**

5   $b_h(s, a) \leftarrow \max\left\{7\sqrt{\frac{\mathbb{V}(\bar{P}_{s,a}, V_{h+1})\iota}{\mathbf{N}^+(s, a)}}, \frac{49B\sqrt{S}\iota}{\mathbf{N}^+(s, a)}\right\}$ for all $(s, a)$.

6   $Q_h(s, a) \leftarrow \max\{0, \widehat{c}(s, a) + \bar{P}_{s,a}V_{h+1} - b_h(s, a) - x\}$ for all $(s, a)$.

   $V_h(s) \leftarrow \min_a Q_h(s, a)$ for all $s$.

7  **if** $\max_{s, a, h} Q_h(s, a) \leq B/4$ **then break; else** $x \leftarrow 2x$.

---

This procedure (adaptively) generates a non-stationary finite-horizon Markov Decision Process (MDP) that $\mathfrak{A}$ faces: $\mathring{\mathcal{M}} = (\mathcal{S}, \mathcal{A}, g, \{c^m\}_{m=1}^M, \{P^m\}_{m=1}^M, c_f, H)$. Here, $c^m = c_{k(m)}$ and $P^m = P_{k(m)}$ where $k(m)$ is the unique episode that interval $m$ belongs to, and $M$ is the total number of intervals over $K$ episodes, a random variable determined by the interactions. Note that $c^m$ and $P^m$ always lie in the oblivious sets $\{c_k\}_{k=1}^K$ and $\{P_k\}_{k=1}^K$ respectively, but $c^m$ and $P^m$ are not oblivious since their values depend on the interaction history. Let $V_1^{\pi, m}(s)$ be the expected cost (including the terminal cost) of following policy $\pi$ starting from state $s$ in interval $m$. Define the regret of $\mathfrak{A}$ over the first $M'$ intervals in $\mathring{\mathcal{M}}$ as $\mathring{R}_{M'} = \sum_{m=1}^{M'}(\sum_{h=1}^{H_m+1} c_h^m - V_1^{\pi_{k(m)}^\star, m}(s_1^m))$ where we use $c_{H_m+1}^m$ as a shorthand for the terminal cost $c_f(s_{H_m+1}^m)$. Following similar arguments as in [Cohen et al., 2021, Chen et al., 2022a], the regret in $\mathcal{M}$ and $\mathring{\mathcal{M}}$ are close in the following sense.

**Lemma 1.** *Algorithm 1 ensures $R_K \leq \mathring{R}_M + B_\star$.*

See Appendix C for the proof. Based on this lemma, in following sections we focus on developing the finite-horizon algorithm $\mathfrak{A}$ and analyzing how large $\mathring{R}_M$ is. Note, however, that while this finite-horizon reduction is very useful, it does not mean that our problem is as easy as learning non-stationary finite-horizon MDPs and that we can directly plug in an existing algorithm as $\mathfrak{A}$. Great care is still needed when designing $\mathfrak{A}$ in order to obtain tight regret bounds as we will show.

## 5 A Simple Sub-Optimal Algorithm

In this section, we present a relatively simple finite-horizon algorithm $\mathfrak{A}$ for $\mathring{\mathcal{M}}$ which, in combination with the reduction of Algorithm 1, achieves a regret bound that almost matches our lower bound except that $T_\star$ is replaced by $T_{\max}$. The key steps are shown in Algorithm 2. It follows the ideas

---

[2] $z \overset{+}{\leftarrow} y$ is a shorthand for $z \leftarrow z + y$.

of the MVP algorithm [Zhang et al., 2020] and adopts a UCBVI-style update rule (Line 6) with a Bernstein-type bonus term (Line 5) to maintain a set of $Q_h$ functions, which then determines the action at each step in a greedy manner (Line 1). The two crucial new elements are the following. First, in the update rule Line 6, we subtract a positive value $x$ uniformly over all state-action pairs so that $\|Q_h\|_\infty$ is of order $\mathcal{O}(B_\star)$ (recall $B = 16B_\star$), and we find the (almost) smallest such $x$ via a doubling trick (Line 7). This is similar to the adaptive confidence widening technique of [Wei and Luo, 2021], where they increase the size of the transition confidence set to ensure a bounded magnitude on the estimated value function; our approach is an adaptation of their idea to the UCBVI style update rule.

Second, we periodically restart the algorithm (by resetting some counters and statistics) in Line 3 and Line 4. While periodic restart is a standard idea to deal with non-stationarity, the novelty here is a two-scale restart schedule: we set one window size $W_c$ related to costs and another one $W_P$ related to transitions, and restart after every $W_c$ intervals or every $W_P$ intervals. As mentioned, this two-scale schedule is inspired by the lower bound in Section 3, which indicates that cost estimation and transition estimation play different roles in the final regret and should be treated separately.

Another small modification is that we start a new interval when the visitation to some $(s,a)$ doubles (Line 2), which helps remove $T_{\max}$ dependency in lower-order terms and is important for following sections. With all these elements, we prove the following regret guarantee of Algorithm 2.

**Theorem 2.** *For any $M' \leq M$, with probability at least $1 - 22\delta$ Algorithm 2 ensures $\mathring{R}_{M'} = \tilde{\mathcal{O}}\left(M'\left(\sqrt{B_\star SA(1/W_c + B_\star/W_P)} + B_\star SA(1/W_c + S/W_P)\right) + (\Delta_c W_c + B_\star \Delta_P W_P)T_{\max}\right).$*

Thus, with a proper tunning of $W_c$ and $W_P$ (that is in term of $M'$), Algorithm 2 ensures $\mathring{R}_{M'} = \tilde{\mathcal{O}}((B_\star SAT_{\max}(\Delta_c + B_\star^2 \Delta_P))^{1/3}M'^{2/3})$. However, this does not directly imply a bound on $\mathring{R}_M$ since $M$ is a random variable (and the tunning above would depend on $M$). Fortunately, to resolve this it suffices to perform a doubling trick on the number of intervals, that is, first make a guess on $M$, and then double the guess whenever $M$ exceeds it. We summarize this idea in Algorithm 3. Finally, combining it with Algorithm 1, Lemma 1, and the simplified analysis of [Chen et al., 2022a] which is able to bound the total number of intervals $M$ in terms of the total number of episodes $K$ (Lemma 16), we obtain the following result (all proofs are deferred to Appendix D).

**Theorem 3.** *With probability at least $1-22\delta$, applying Algorithm 1 with $\mathfrak{A}$ being Algorithm 3 ensures $R_{K'} = \tilde{\mathcal{O}}((B_\star SAT_{\max}(\Delta_c + B_\star^2 \Delta_P))^{1/3}K'^{2/3})$ (ignoring lower order terms) for any $K' \leq K$.*

Note that Theorem 3 actually provides an anytime regret guarantee (that is, holds for any $K' \leq K$), which is important in following sections. Compared to our lower bound in Theorem 1, the only sub-optimality is in replacing $T_\star$ with the larger quantity $T_{\max}$. Despite its sub-optimality for SSP, however, as a side result our algorithm in fact implies the first model-based finite-horizon algorithm that achieves the optimal dependency on $SA$ and matches the minimax lower bound of [Mao et al., 2021]. Specifically, in previous works, the optimal $SA$ dependency is only achievable by model-free algorithms, which unfortunately have sub-optimal dependency on the horizon by the current analysis (see [Mao et al., 2021, Lemma 10]). On the other hand, existing model-based algorithms for finite state-action space all follow the idea of extended value iteration, which gives sub-optimal dependency on $S$ and also brings difficulty in incorporating entry-wise Bernstein confidence sets.[3] Our approach, however, resolves all these issues. See Appendix D.4 for more discussions.

**Technical Highlights** The key step of our proof for Theorem 2 is to bound the term $\sum_{m=1}^{M'} \sum_{h=1}^{H_m} \mathbb{V}(P_{s_h^m, a_h^m}^m, V_{h+1}^{\star,m} - V_{h+1}^m)$, where $V_{h+1}^m$ is the value of $V_{h+1}$ at the beginning of interval $m$, and $V_{h+1}^{\star,m}$ is the optimal value function of $\mathring{\mathcal{M}}$ in interval $m$ (formally defined in Appendix A). The standard analysis on bounding this term requires $V_{h+1}^{\star,m}(s) - V_{h+1}^m(s) \geq 0$, which is only true in a stationary environment due to optimism. To handle this in non-stationarity environments, we carefully choose a set of constants $\{z_h^m\}$ so that $V_{h+1}^{\star,m}(s) + z_h^m - V_{h+1}^m(s) \geq 0$ (Lemma 18), and then apply similar analysis on $\sum_{m=1}^{M'} \sum_{h=1}^{H_m} \mathbb{V}(P_{s_h^m, a_h^m}^m, V_{h+1}^{\star,m} - V_{h+1}^m) = \sum_{m=1}^{M'} \sum_{h=1}^{H_m} \mathbb{V}(P_{s_h^m, a_h^m}^m, V_{h+1}^{\star,m} + z_h^m - V_{h+1}^m)$. See Lemma 20 for more details.

---

[3]Note that the transition non-stationarity $\Delta_P$ is defined via $L_1$ norm. Thus, naively applying entry-wise confidence widening to Bernstein confidence sets introduces extra dependency on $S$.

---

**Algorithm 3** Non-Stationary MVP with a Doubling Trick

---

**for** $n = 1, 2, \ldots$ **do**

    Initialize an instance of Algorithm 2 with $W_c = \lceil (B_\star SA)^{1/3}(2^{n-1}/(\Delta_c T_{\max}))^{2/3} \rceil$ and $W_P = \lceil (SA)^{1/3}(2^{n-1}/(\Delta_P T_{\max}))^{2/3} \rceil$, and execute it in intervals $m = 2^{n-1}, \ldots, 2^n - 1$.

---

---

**Algorithm 4** MVP with Non-Stationarity Tests

---

**Parameters:** window sizes $W_c$ and $W_P$, coefficients $c_1$, $c_2$, sample probability $p$, and failure probability $\delta$.

**Initialize:** `ResetC()`, `ResetP()`, `Update(1)`.

**for** $m = 1, \ldots, M$ **do**

    **for** $h = 1, \ldots, H$ **do**

        Play action $a_h^m \leftarrow \operatorname{argmin}_a \check{Q}_h(s_h^m, a)$, receive cost $c_h^m$ and next state $s_{h+1}^m$.

        $\mathbf{C}(s_h^m, a_h^m) \overset{+}{\leftarrow} c_h^m, \mathbf{M}(s_h^m, a_h^m) \overset{+}{\leftarrow} 1, \mathbf{N}(s_h^m, a_h^m) \overset{+}{\leftarrow} 1, \mathbf{N}(s_h^m, a_h^m, s_{h+1}^m) \overset{+}{\leftarrow} 1.$

**1**        $\widehat{\chi}^c \overset{+}{\leftarrow} c_h^m - \widehat{c}(s_h^m, a_h^m), \widehat{\chi}^P \overset{+}{\leftarrow} \check{V}_{h+1}(s_{h+1}^m) - \bar{P}_{s_h^m, a_h^m} \check{V}_{h+1}.$

        **if** $s_{h+1}^m = g$ *or* $\mathbf{M}(s_h^m, a_h^m) = 2^l$ *or* $\mathbf{N}(s_h^m, a_h^m) = 2^l$ *for some integer* $l \geq 0$ **then**

            **break** (which start a new interval).

**2**    **if** $\widehat{\chi}^c > \chi_m^c$ *(defined in Lemma 24)* **then** `ResetC()`. **(Test 1)**

**3**    **if** $\widehat{\chi}^P > \chi_m^P$ *(defined in Lemma 25)* **then** `ResetC()` and `ResetP()`. **(Test 2)**

**4**    **if** $\nu^c = W_c$ **then** `ResetC()`.

**5**    **if** $\nu^P = W_P$ **then** `ResetC()` and `ResetP()`.

    $\nu^c \overset{+}{\leftarrow} 1, \nu^P \overset{+}{\leftarrow} 1, \texttt{Update}(m + 1).$

**6**    **if** $\left\| \check{V}_h \right\|_\infty > B/2$ *for some* $h$ *(Test 3)* **then**

        `ResetC()`, with probability $p$ execute `ResetP()`, and `Update(m + 1)`.

**Procedure** `Update`$(m)$

    $\check{V}_{H+1}(s) \leftarrow 2B_\star \mathbb{I}\{s \neq g\}, \check{V}_h(g) \leftarrow 0$ for all $h \leq H$, and $\iota \leftarrow 2^{11} \cdot \ln\left(\frac{2SAHKm}{\delta}\right).$

**7**    $\rho^c \leftarrow \min\{\frac{c_1}{\sqrt{\nu^c}}, \frac{1}{2^8 H}\}, \rho^P \leftarrow \min\{\frac{c_2}{\sqrt{\nu^P}}, \frac{1}{2^8 H}\}, \eta \leftarrow \rho^c + B\rho^P.$

    **for** *all* $(s, a)$ **do**

        $\mathbf{N}^+(s, a) \leftarrow \max\{1, \mathbf{N}(s, a)\}, \mathbf{M}^+(s, a) \leftarrow \max\{1, \mathbf{M}(s, a)\}, \bar{c}(s, a) \leftarrow \frac{\mathbf{C}(s, a)}{\mathbf{M}^+(s, a)},$

        $\bar{P}_{s, a}(\cdot) \leftarrow \frac{\mathbf{N}(s, a, \cdot)}{\mathbf{N}^+(s, a)}, \widehat{c}(s, a) \leftarrow \max\left\{0, \bar{c}(s, a) - \sqrt{\frac{\bar{c}(s, a)\iota}{\mathbf{M}^+(s, a)}} - \frac{\iota}{\mathbf{M}^+(s, a)}\right\},$

**8**        $\check{c}(s, a) \leftarrow \widehat{c}(s, a) + 8\eta.$

    **for** $h = H, \ldots, 1$ **do**

        $b_h(s, a) \leftarrow \max\left\{7\sqrt{\frac{\mathbb{V}(\bar{P}_{s, a}, \check{V}_{h+1})\iota}{\mathbf{N}^+(s, a)}}, \frac{49B\sqrt{S}\iota}{\mathbf{N}^+(s, a)}\right\}$ for all $(s, a)$.

        $\check{Q}_h(s, a) = \max\{0, \check{c}(s, a) + \bar{P}_{s, a} \check{V}_{h+1} - b_h(s, a)\}$ all $(s, a)$.

        $\check{V}_h(s) = \operatorname{argmin}_a \check{Q}_h(s, a)$ for all $s$.

**Procedure** `ResetC`()

    $\nu^c \leftarrow 1, \widehat{\chi}^c \leftarrow 0, \mathbf{C}(s, a) \leftarrow 0, \mathbf{M}(s, a) \leftarrow 0$ for all $(s, a)$.

**Procedure** `ResetP`()

    $\nu^P \leftarrow 1, \widehat{\chi}^P \leftarrow 0, \mathbf{N}(s, a, s') \leftarrow 0, \mathbf{N}(s, a) \leftarrow 0$ for all $(s, a, s')$.

---

## 6 A Minimax Optimal Algorithm

In this section, we present an improved algorithm that achieves the minimax optimal regret bound up to logarithmic terms, starting with a refined version of Algorithm 2 shown in Algorithm 4. Below, we focus on describing the new elements introduced in Algorithm 4 (that is, Lines 1-3 and 6-4).[4]

The main challenge in replacing $T_{\max}$ with $T_\star$ is that the regret due to non-stationarity accumulates along the learner's trajectory, which can be as large as $\mathcal{O}((\Delta_c + B_\star \Delta_P)H)$ since the horizon is $H$

---

[4]Line 4 and Line 5, althogh written in a different form, are similar to Line 3 and Line 4 of Algorithm 2.

| **Algorithm 5** A Two-Phase Variant of Algorithm 1 |
|---|
| **Initialize:** Phase 1 algorithm instance $\mathfrak{A}_1$ and Phase 2 algorithm instance $\mathfrak{A}_2$. 
 Execute Algorithm 1 with $\mathfrak{A} = \mathfrak{A}_1$ for every first interval of an episode, and $\mathfrak{A} = \mathfrak{A}_2$ otherwise. |

(recall $H = \tilde{\mathcal{O}}(T_{\max})$). Moreover, bounding the number of steps needed for the learner's policy to reach the goal is highly non-trivial due to the changing transitions. Our main idea to address these issues is to incorporate a correction term $\eta$ (computed in Line 7) into the estimated cost (Line 4) to penalize policies that take too long to reach the goal. This correction term is set to be an upper bound of the learner's average regret per interval (defined through $\rho^c$ and $\rho^P$ in Line 7). It introduces the effect of canceling the non-stationarity along the learner's trajectory when it is not too large. When the non-stationarity is large, on the other hand, we detect it through two non-stationary tests (Line 2 and Line 3), and reset the knowledge of the environment (more details to follow).

However, this correction leads to one issue: we cannot perform adaptive confidence widening (that is, the $-x$ bias) anymore as it would cancel out the correction term. To address this, we introduce another test (Line 6, **Test 3**) to directly check whether the magnitude of the estimated value function is bounded as desired. If not, we reset again since that is also an indication of large non-stationarity.

We now provide some intuitions on the design of **Test 1** and **Test 2**. First, one can show that the two quantities $\widehat{\chi}^c$ and $\widehat{\chi}^P$ we maintain in Line 1 are such that their sum is roughly an upper bound on the estimated accumulated regret. So directly checking whether $\widehat{\chi}^c + \widehat{\chi}^P$ is too large would be similar to the second test of the MASTER algorithm [Wei and Luo, 2021]. Here, however, we again break it into two tests where **Test 1** only guards the non-stationarity in cost, and **Test 2** mainly guards the non-stationarity in transition. Note that **Test 2** also involves cost information through $\tilde{V}$, but our observation is that we can still achieve the desired regret bound as long as the ratio of the number of resets caused by procedures ResetC() and ResetP() is of order $\tilde{\mathcal{O}}(B_\star)$. This inspires us to reset both the cost and the transition estimation when **Test 2** fails, but reset the transition estimation only with some probability $p$ (eventually set to $1/B_\star$) when **Test 3** fails.

For analysis, we first establish a regret guarantee of Algorithm 4 in an ideal situation where the first state of each interval is always $s_{\text{init}}$. (Proofs of this section are deferred to Appendix E.)

**Theorem 4.** *Let* $c_1 = \sqrt{B_\star SA}/T_\star$, $c_2 = \sqrt{SA}/T_\star$, $W_c = \lceil (B_\star SA)^{1/3}(K/(\Delta_c T_\star))^{2/3} \rceil$, $W_P = \lceil (SA)^{1/3}(K/(\Delta_P T_\star))^{2/3} \rceil$, *and* $p = 1/B_\star$. *Suppose* $s_1^m = s_{init}$ *for all* $m \leq K$, *then Algorithm 4 ensures* $\mathring{R}_K = \tilde{\mathcal{O}}((B_\star SAT_\star(\Delta_c + B_\star^2 \Delta_P))^{1/3}K^{2/3})$ *(ignoring lower order terms) with probability at least* $1 - 40\delta$.

The reason that we only analyze this ideal case is that, if the initial state is not $s_{\text{init}}$, then even the optimal policy does not guarantee $T_\star$ hitting time by definition. This also inspires us to eventually deploy a two-phase algorithm slightly modifying Algorithm 1: feed the first interval of each episode into an instance of Algorithm 4, and the rest of intervals into an instance of Algorithm 3 (see Algorithm 5). Thanks to the large terminal cost, we are able to show that the regret in the second phase is upper bounded by a constant, leading to the following final result.

**Theorem 5.** *Algorithm 5 with* $\mathfrak{A}_1$ *being Algorithm 4 and* $\mathfrak{A}_2$ *being Algorithm 3 ensures* $R_K = \tilde{\mathcal{O}}((B_\star SAT_\star(\Delta_c + B_\star^2 \Delta_P))^{1/3}K^{2/3})$ *(ignoring lower order terms) with probability at least* $1 - 64\delta$.

Ignoring logarithmic and lower-order terms, our bound is minimax optimal. Also note that the bound is sub-linear (in $K$) as long as $\Delta_c$ and $\Delta_P$ are sub-linear (that is, not the worst case).

## 7 Learning without Knowing $\Delta_c$ and $\Delta_P$

To handle unknown non-stationarity, we combine our algorithmic ideas in previous sections with a new variant of the MASTER algorithm [Wei and Luo, 2021]. The original MASTER algorithm is a blackbox reduction that takes a base algorithm for (near) stationary environments as input, and turns it into another algorithm for non-stationarity environments. For many problems (including multi-armed bandits, contextual bandits, linear bandits, finite-horizon or infinite-horizon MDPs), Wei and Luo [2021] show that the final algorithm achieves optimal regret without knowing the non-stationarity. While powerful, MASTER can not be directly used in our problem to achieve the

same strong result. As we will discuss, some modification is needed, and even with this modification, some extra difficulty unique to SSP still prevents us from eventually obtaining the optimal regret.

Specifically, in order to obtain $T_\star$ dependency, we again follow the two-phase procedure Algorithm 5 and instantiate a MASTER algorithm with a different base algorithm in each phase. In Phase 1, since it is unclear how to update cost and transition estimation independently under the framework of MASTER, we adopt a simpler version of Algorithm 4 as the base algorithm, which performs synchronized cost and transition estimation and a simpler non-stationarity test; see Algorithm 6 (all algorithms/proofs in this section are deferred to Appendix F due to space limit). In Phase 2, we use Algorithm 2 as the base algorithm.

Our version of the MASTER algorithm (Algorithm 8) requires a different **Test 1** compared to that in [Wei and Luo, 2021], which is essential due to the presence of the correction terms in Algorithm 6. Specifically, it no longer makes sense to simply maintain the maximum of estimated value functions over the past intervals, since the cost function combined with the correction term is changing adaptively, and a large correction term will interfere with the detection of a small amount of non-stationarity. Our key observation is that for a base algorithm scheduled on a given range by MASTER, the average of its correction terms within the same range is of the desired order that does not interfere with non-stationarity detection. This inspires us to maintain multiple running averages of the estimated value functions with different scales (see Line 2 of Algorithm 8). Then, to detect a certain level of non-stationarity, we refer to the running average with the matching scale (see Line 3).

We show that the algorithm described above achieves the following regret guarantee without knowledge of the non-stationarity.

**Theorem 6.** *Let $\mathfrak{A}_1$ be an instance of Algorithm 8 with Algorithm 6 as the base algorithm and $\mathfrak{A}_2$ be an instance of Algorithm 8 with Algorithm 2 as the base algorithm. Then Algorithm 5 with $\mathfrak{A}_1$ and $\mathfrak{A}_2$ ensures with high probability (ignoring lower order terms):*

$$R_K = \tilde{\mathcal{O}}\left(\min\left\{B_\star S\sqrt{ALK}, B_\star S\sqrt{AK} + (B_\star^2 S^2 A(\Delta_c + B_\star \Delta_P)T_\star)^{1/3}K^{2/3}\right\}\right),$$

*where $L = 1 + \sum_{k=1}^{K-1}\mathbb{I}\{P_{k+1} \neq P_k \text{ or } c_{k+1} \neq c_k\}$ is the number changes of the environment (plus one). Moreover, this is achieved without the knowledge of $\Delta_c$, $\Delta_P$, or $L$.*

The advantage of this result compared to Theorem 5 is two-fold. First, it adapts to different levels of non-stationarity ($\Delta_c$, $\Delta_P$, and $L$) automatically. Second, it additionally achieves a bound of order $\tilde{\mathcal{O}}(B_\star S\sqrt{ALK})$, which could be much better than that in Theorem 5; for example, when $L = \mathcal{O}(1)$, the former is a $\sqrt{K}$-order bound while the latter is of order $K^{2/3}$. As discussed in [Wei and Luo, 2021], this is a unique benefit brought by the MASTER algorithm and is not achieved by any other algorithms even with the knowledge of $L$.

The disadvantage of Theorem 6, on the other hand, is its sub-optimality in the $B_\star$ dependency for the $\Delta_c$-related term and the $S$ dependency for both terms. The extra $B_\star$ dependency is due to the synchronized cost and transition estimation. As mentioned, it is unclear how to update cost and transition estimation independently as we do in Algorithm 4 under the framework of MASTER, which we leave as an important future direction. On the other hand, the extra $S$ dependency comes from the fact that the lower-order term in the regret bound of the base algorithm affects the final regret bound (see the statement of Theorem 13). Specifically, the lower-order term is $B_\star S^2 A$ instead of $B_\star SA$, which eventually leads to extra $S$ dependency. How to remove the extra $S$ factor in the base algorithm, or eliminate the undesirable lower-order term effect brought by the MASTER algorithm, is another important future direction.

## 8 Conclusion

In this work, we develop the first set of results for dynamic regret minimization in non-stationary SSP, including a (near) minimax optimal algorithm and two others that are either simpler or advantageous in some other cases. Besides the immediate next step such as improving our results when the non-stationarity is unknown, our work also opens up many other possible future directions on this topic, such as extension to more general settings with function approximation. It would also be interesting to study more adaptive dynamic regret bounds in this setting. For example, our $B_\star$ and $T_\star$ are defined as the maximum optimal expected cost and hitting time over all episodes, which is undesirable if only

a few episodes admit a large optimal expected cost or hitting time. Ideally, some kind of (weighted) average would be a more reasonable measure in these cases.

## Acknowledgments and Disclosure of Funding

The authors thank Aviv Rosenberg and Chen-Yu Wei for many helpful discussions. HL is supported by NSF Award IIS-1943607 and a Google Research Scholar Award.

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
