# Contents of Appendix

## A  Preliminaries

**Extra Notations**  We first define (or restate) some notations used throughout the whole Appendix.

- Let $\Delta_{c,[i,j]} = \sum_{\tau=i}^{j-1} \left\| c^{\tau+1} - c^\tau \right\|_\infty$, $\Delta_{P,[i,j]} = \sum_{\tau=i}^{j-1} \max_{s,a} \left\| P_{s,a}^{\tau+1} - P_{s,a}^\tau \right\|_1$. It is straightforward to verify that $\Delta_{c,[1,M]} = \Delta_c$ and $\Delta_{P,[1,M]} = \Delta_P$.

- Define $\Delta_{c,m} = \Delta_{c,[i_m^c,m]}$ and $\Delta_{P,m} = \Delta_{P,[i_m^P,m]}$, where $i_m^c$ and $i_m^P$ are the first intervals after the last resets of $\mathbf{M}$ and $\mathbf{N}$ before interval $m$ respectively.

- For all algorithms, denote by $\widehat{c}^m, \bar{c}^m, \bar{P}_{s,a}^m, b_h^m, \mathbf{N}_m^+, \mathbf{M}_m^+, \iota_m$ the value of $\widehat{c}, \bar{c}, \bar{P}_{s,a}, b_h, \mathbf{N}^+, \mathbf{M}^+, \iota$ at the beginning of interval $m$, and define $\widehat{c}_h^m = \widehat{c}^m(s_h^m, a_h^m), \bar{c}_h^m = \bar{c}^m(s_h^m, a_h^m)$, $\mathbf{N}_h^m = \mathbf{N}^+(s_h^m, a_h^m)$, and $\mathbf{M}_h^m = \mathbf{M}^+(s_h^m, a_h^m)$. We also slightly abuse the notation and write $b^m(s_h^m, a_h^m)$ as $b_h^m$ when there is no confusion.

- Define $\widetilde{c}^m(s,a) = \frac{1}{\mathbf{M}_m^+(s,a)} \sum_{m'=i_m^c}^{m-1} \sum_{h=1}^{H_{m'}} c^{m'}(s,a)\mathbb{I}\{(s_h^{m'}, a_h^{m'}) = (s,a)\}$, $\widetilde{c}_h^m = \widetilde{c}^m(s_h^m, a_h^m)$, $\widetilde{P}_{s,a}^m = \frac{1}{\mathbf{N}_m^+(s,a)} \sum_{m'=i_m^P}^{m-1} \sum_{h=1}^{H_{m'}} P_{s,a}^{m'}\mathbb{I}\{(s_h^{m'}, a_h^{m'}) = (s,a)\}$, $\bar{P}_h^m = \bar{P}_{s_h^m, a_h^m}^m$, and $\widetilde{P}_h^m = \widetilde{P}_{s_h^m, a_h^m}^m$.

- Denote by $L_{c,[i,j]}$ and $L_{P,[i,j]}$ one plus the number of resets of $\mathbf{M}$ and $\mathbf{N}$ within intervals $[i,j]$ respectively, and define $L_{c,m} = L_{c,[1,m]}$, $L_{P,m} = L_{P,[1,m]}$, $L_m = L_{c,m} + L_{P,m}$ for any $m \geq 1$.

- Define $f^c(m)$ (or $f^P(m)$) as the earliest interval at or after interval $m$ in which the learner resets $\mathbf{M}$ (or $\mathbf{N}$).

- Define $\mathbf{m}_h^m = \mathbb{I}\{\mathbf{M}^m(s_h^m, a_h^m) = 0\}$, $\mathbf{n}_h^m = \mathbb{I}\{\mathbf{N}^m(s_h^m, a_h^m) = 0\}$, $C_{M'} = \sum_{m=1}^{M'} \sum_{h=1}^{H_m+1} c_h^m$, and bonus function $b^m(s,a,V) = \max\left\{7\sqrt{\frac{\mathbb{V}(\bar{P}_{s,a}^m, V)\iota_m}{\mathbf{N}_m^+(s,a)}}, \frac{49B\sqrt{S}\iota_m}{\mathbf{N}_m^+(s,a)}\right\}$.

- Define $T_h^{\pi^\star, m}(s)$ (or $T_h^{\pi^\star, m}(s,a)$) as the hitting time (reaching $g$ or layer $H+1$) of $\pi_{k(m)}^\star$ starting from state $s$ (or state-action pair $(s,a)$) in layer $h$ w.r.t transition $P^m$, such that $T_h^{\pi^\star, m}(s,a) = 1 + P_{s,a}^m T_{h+1}^{\pi^\star, m}$, $T_h^{\pi^\star, m}(s) = T_h^{\pi^\star, m}(s, \pi_{k(m)}^\star(s))$, and $T_{H+1}^{\pi^\star, m}(s) = T_{H+1}^{\pi^\star, m}(s,a) = T_h^{\pi^\star, m}(g) = T_h^{\pi^\star, m}(g,a) = 0$.

- For notational convenience, we often write $V_h^{\pi_{k(m)}^\star, m}$ as $V_h^{\pi^\star, m}$.

- Define $(x)_+ = \max\{0, x\}$.

**Optimal Value Functions of $\mathring{\mathcal{M}}$** We denote by $Q_h^{\star, m}$ and $V_h^{\star, m}$ the optimal value functions in interval $m$. It is not hard to see that they can be defined recursively as follows: $V_{H+1}^{\star, m} = c_f$ and for $h \leq H$,

$$Q_h^{\star, m}(s,a) = c^m(s,a) + P_{s,a}^m V_{h+1}^{\star, m}, \qquad V_h^{\star, m}(s) = \min_a Q_h^{\star, m}(s,a).$$

For notational convenience, we also let $Q_{H+1}^{\star, m}(s,a) = V_{H+1}^{\star, m}(s)$ for any $(s,a) \in \mathcal{S} \times \mathcal{A}$.

**Lemma 2.** *For any $m \geq 1$ and $h \leq H+1$, $Q_h^{\star, m}(s,a) \leq Q_h^{\pi^\star, m}(s,a) \leq 4B_\star$.*

*Proof.* This is simply by $Q_h^{\pi^\star, m}(s,a) \leq 1 + \max_s V_k^\star(s) + 2B_\star \leq 4B_\star$. $\square$

**Auxiliary Lemmas** Below we provide auxiliary lemmas used throughout the whole Appendix and for all algorithms.

**Lemma 3.** *With probability at least $1 - 3\delta$, $\sum_{m=1}^{M'} \sum_{h=1}^{H_m}(c^m(s_h^m, a_h^m) - \widehat{c}_h^m) \leq 3\sum_{m=1}^{M'} \sum_{h=1}^{H_m}\left(\sqrt{\frac{\widehat{c}_h^m \iota_m}{\mathbf{M}_h^m}} + \frac{\iota_m}{\mathbf{M}_h^m}\right) + \sum_{m=1}^{M'} \sum_{h=1}^{H_m} \Delta_{c,m} \leq \tilde{\mathcal{O}}\left(\sqrt{SAL_{c,M'}C_{M'}} + SAL_{c,M'}\right) + 2\sum_{m=1}^{M'} \sum_{h=1}^{H_m} \Delta_{c,m}$ and $\sum_{m=1}^{M'} \sum_{h=1}^{H_m}\left(\sqrt{\frac{\widehat{c}_h^m \iota_m}{\mathbf{M}_h^m}} + \frac{\iota_m}{\mathbf{M}_h^m}\right) \leq \tilde{\mathcal{O}}(\sqrt{SAL_{c,M'}C_{M'}} + SAL_{c,M'} + \sqrt{SAL_{c,M'} \sum_{m=1}^{M'} \sum_{h=1}^{H_m} \Delta_{c,m}})$ for any $M' \leq M$.*

*Proof.* First note that by Lemma 49, with probability at least $1 - \delta$, for any $m \geq 1$ and $(s,a) \in \mathcal{S} \times \mathcal{A}$,

$$\widetilde{c}^m(s,a) - \bar{c}^m(s,a) \leq \sqrt{\frac{\bar{c}^m(s_h^m, a_h^m)}{\mathbf{M}_m^+(s,a)}} + \frac{1}{\mathbf{M}_m^+(s,a)}. \tag{1}$$

For the first inequality in the first statement, note that

$$\sum_{m=1}^{M'} \sum_{h=1}^{H_m} (c^m(s_h^m, a_h^m) - \widehat{c}_h^m)$$

$$\leq \sum_{m=1}^{M'} \sum_{h=1}^{H_m} \left( \widetilde{c}^m(s_h^m, a_h^m) - \bar{c}^m(s_h^m, a_h^m) + \sqrt{\frac{\bar{c}_h^m \iota_m}{\mathbf{M}_h^m}} + \frac{\iota_m}{\mathbf{M}_h^m} + \mathbf{m}_h^m \right) + \sum_{m=1}^{M'} \sum_{h=1}^{H_m} \Delta_{c,m}$$

(definition of $\widehat{c}_h^m$ and $c^m(s_h^m, a_h^m) \leq \widetilde{c}^m(s_h^m, a_h^m) + \Delta_{c,m} + \mathbf{m}_h^m$)

$$\leq 3 \sum_{m=1}^{M'} \sum_{h=1}^{H_m} \left( \sqrt{\frac{\bar{c}_h^m \iota_m}{\mathbf{M}_h^m}} + \frac{\iota_m}{\mathbf{M}_h^m} \right) + \sum_{m=1}^{M'} \sum_{h=1}^{H_m} \Delta_{c,m}. \qquad \text{(Eq. (1) and } \mathbf{m}_h^m \leq \frac{1}{\mathbf{M}_h^m})$$

The second inequality in the first statement simply follows from applying AM-GM inequality on the second statement. To prove the second statement, first note that by Lemma 49, Cauchy-Schwarz inequality, and Lemma 11, with probability at least $1 - \delta$,

$$\sum_{m=1}^{M'} \sum_{h=1}^{H_m} \bar{c}_h^m = \tilde{\mathcal{O}} \left( \sum_{m=1}^{M'} \sum_{h=1}^{H_m} \left( \widetilde{c}_h^m + \sqrt{\frac{\bar{c}_h^m}{\mathbf{M}_h^m}} + \frac{1}{\mathbf{M}_h^m} \right) \right)$$

$$= \tilde{\mathcal{O}} \left( \sum_{m=1}^{M'} \sum_{h=1}^{H_m} \widetilde{c}_h^m + \sqrt{SAL_{c,M'} \sum_{m=1}^{M'} \sum_{h=1}^{H_m} \bar{c}_h^m} + SAL_{c,M'} \right).$$

Solving a quadratic inequality w.r.t $\sum_{m=1}^{M'} \sum_{h=1}^{H_m} \bar{c}_h^m$ (Lemma 45) gives $\sum_{m=1}^{M'} \sum_{h=1}^{H_m} \bar{c}_h^m = \tilde{\mathcal{O}}(\sum_{m=1}^{M'} \sum_{h=1}^{H_m} \widetilde{c}_h^m + SAL_{c,M'})$. Therefore, with probability at least $1 - \delta$,

$$\sum_{m=1}^{M'} \sum_{h=1}^{H_m} \left( \sqrt{\frac{\bar{c}_h^m \iota_m}{\mathbf{M}_h^m}} + \frac{\iota_m}{\mathbf{M}_h^m} \right) = \tilde{\mathcal{O}} \left( \sqrt{SAL_{c,M'} \sum_{m=1}^{M'} \sum_{h=1}^{H_m} \bar{c}_h^m} + SAL_{c,M'} \right)$$

(Cauchy-Schwarz inequality and Lemma 11)

$$= \tilde{\mathcal{O}} \left( \sqrt{SAL_{c,M'} \sum_{m=1}^{M'} \sum_{h=1}^{H_m} \widetilde{c}_h^m} + SAL_{c,M'} \right)$$

$$= \tilde{\mathcal{O}} \left( \sqrt{SAL_{c,M'} \sum_{m=1}^{M'} \sum_{h=1}^{H_m} \Delta_{c,m}} + \sqrt{SAL_{c,M'} \sum_{m=1}^{M'} \sum_{h=1}^{H_m} c^m(s_h^m, a_h^m)} + SAL_{c,M'} \right)$$

$$= \tilde{\mathcal{O}} \left( \sqrt{SAL_{c,M'} \sum_{m=1}^{M'} \sum_{h=1}^{H_m} \Delta_{c,m}} + \sqrt{SAL_{c,M'} C_{M'}} + SAL_{c,M'} \right). \qquad \text{(Lemma 50)}$$

This completes the proof. $\qquad \square$

**Lemma 4.** *With probability at least $1 - \delta$, for any $m \geq 1$, $(s,a) \in \mathcal{S} \times \mathcal{A}$ and $s' \in \mathcal{S}_+$,*
$$\left| \widetilde{P}_{s,a}^m(s') - \bar{P}_{s,a}^m(s') \right| \leq \sqrt{\frac{\widetilde{P}_{s,a}^m(s') \iota_m}{2\mathbf{N}_m^+(s,a)}} + \frac{\iota_m}{2\mathbf{N}_m^+(s,a)} \leq \sqrt{\frac{\bar{P}_{s,a}^m(s') \iota_m}{\mathbf{N}_m^+(s,a)}} + \frac{\iota_m}{\mathbf{N}_m^+(s,a)}.$$

*Proof.* The first inequality hold with probability at least $1 - \delta/2$ by applying Lemma 49 for each $(s,a) \in \mathcal{S} \times \mathcal{A}$ and $s' \in \mathcal{S}_+$. Also by Lemma 50, we have $\widetilde{P}_{s,a}^m(s') \leq 2\bar{P}_{s,a}^m(s') + \frac{\iota_m}{2\mathbf{N}_m^+(s,a)}$ for any $(s,a) \in \mathcal{S} \times \mathcal{A}, s' \in \mathcal{S}_+$ with probability at least $1 - \delta/2$. Substituting this back and applying $\sqrt{a+b} \leq \sqrt{a} + \sqrt{b}$ proves the second inequality. $\qquad \square$

**Lemma 5.** *With probability at least $1 - \delta$, for any $(s,a) \in \mathcal{S} \times \mathcal{A}$ and $m \geq 1$, $\widehat{c}^m(s,a) \leq c^m(s,a) + \Delta_{c,m}$.*

*Proof.* For any $(s,a)$ and $m \geq 1$, when $\mathbf{M}_m(s,a) = 0$, the statement clearly holds since $\bar{c}^m(s,a) = 0$. Otherwise, by [Lemma 49](#) and [Lemma 50](#), with probability at least $1 - \delta$, for all $(s,a)$ and $m \geq 1$ simultaneously,

$$|\bar{c}^m(s,a) - \widetilde{c}^m(s,a)| \leq 3\sqrt{\frac{\widetilde{c}^m(s,a)}{\mathbf{M}_m^+(s,a)} \ln \frac{32SAm^5}{\delta}} + \frac{2\ln \frac{32SAm^5}{\delta}}{\mathbf{M}_m^+(s,a)}$$

$$\leq 3\sqrt{\frac{\left(2\bar{c}^m(s,a) + \frac{12\ln \frac{4SAm}{\delta}}{\mathbf{M}_m^+(s,a)}\right)}{\mathbf{M}_m^+(s,a)} \ln \frac{32SAm^5}{\delta}} + \frac{2\ln \frac{32SAm^5}{\delta}}{\mathbf{M}_m^+(s,a)} \leq \sqrt{\frac{\bar{c}^m(s,a)\iota_m}{\mathbf{M}_m^+(s,a)}} + \frac{\iota_m}{\mathbf{M}_m^+(s,a)}. \tag{2}$$

Therefore, by $\max\{0,a\} - \max\{0,b\} \leq \max\{0, a-b\}$,

$$\widehat{c}^m(s,a) - c^m(s,a) \leq \widehat{c}^m(s,a) - \widetilde{c}^m(s,a) + \Delta_{c,m}$$

$$\leq \max\left\{0, \bar{c}^m(s,a) - \widetilde{c}^m(s,a) - \sqrt{\frac{\bar{c}^m(s,a)\iota_m}{\mathbf{M}_m^+(s,a)}} - \frac{\iota_m}{\mathbf{M}_m^+(s,a)}\right\} + \Delta_{c,m} \leq \Delta_{c,m},$$

where the last step is by [Eq. (2)](#). $\qquad\square$

**Lemma 6.** *Given function $V \in [-B, B]^{\mathcal{S}_+}$ for some $B > 0$, we have with probability at least $1 - \delta$,*
$$|(\widetilde{P}_{s,a}^m - \bar{P}_{s,a}^m)V| \leq \tilde{\mathcal{O}}\left(\sqrt{\frac{S\mathbb{V}(P_{s,a}^m,V)}{\mathbf{N}_m^+(s,a)}} + \frac{SB}{\mathbf{N}_m^+(s,a)}\right) + \frac{B\Delta_{P,m}}{64} \text{ for any } m \geq 1.$$

*Proof.* Note that with probability at least $1 - \delta$,

$$|(\widetilde{P}_{s,a}^m - \bar{P}_{s,a}^m)V| = |(\widetilde{P}_{s,a}^m - \bar{P}_{s,a}^m)(V - P_{s,a}^m V)|$$

$$= \tilde{\mathcal{O}}\left(\sum_{s'}\left(\sqrt{\frac{\widetilde{P}_{s,a}^m(s')}{\mathbf{N}_m^+(s,a)}}|V(s') - P_{s,a}^m V| + \frac{B}{\mathbf{N}_m^+(s,a)}\right)\right) \qquad \text{([Lemma 4](#))}$$

$$= \tilde{\mathcal{O}}\left(\sqrt{\frac{S\widetilde{P}_h^m(V - P_{s,a}^m V)^2}{\mathbf{N}_m^+(s,a)}} + \frac{SB}{\mathbf{N}_m^+(s,a)}\right) \qquad \text{(Cauchy-Schwarz inequality)}$$

$$= \tilde{\mathcal{O}}\left(\sqrt{\frac{SP_h^m(V - P_{s,a}^m V)^2}{\mathbf{N}_m^+(s,a)}} + \frac{SB}{\mathbf{N}_m^+(s,a)} + B\sqrt{\frac{S\Delta_{P,m}}{\mathbf{N}_m^+(s,a)}}\right).$$

Applying AM-GM inequality completes the proof. $\qquad\square$

**Lemma 7.** *With probability at least $1 - \delta$, $\mathbb{V}(\bar{P}_h^m, V_{h+1}^m) \leq 2\mathbb{V}(P_h^m, V_{h+1}^m) + \tilde{\mathcal{O}}\left(\frac{SB^2}{\mathbf{N}_h^m}\right) + 2B^2\Delta_{P,m}$ for any $m \geq 1$.*

*Proof.* Note that:

$$\mathbb{V}(\bar{P}_h^m, V_{h+1}^m) \leq \bar{P}_h^m(V_{h+1}^m - P_h^m V_{h+1}^m)^2 \qquad \left(\frac{\sum_i p_i x_i}{\sum_i p_i} = \arg\min_z \sum_i p_i(x_i - z)^2\right)$$

$$= \mathbb{V}(P_h^m, V_{h+1}^m) + (\bar{P}_h^m - P_h^m)(V_{h+1}^m - P_h^m V_{h+1}^m)^2$$

$$\leq \mathbb{V}(P_h^m, V_{h+1}^m) + (\bar{P}_h^m - \widetilde{P}_h^m)(V_{h+1}^m - P_h^m V_{h+1}^m)^2 + B^2\Delta_{P,m}$$

$$\leq \mathbb{V}(P_h^m, V_{h+1}^m) + \tilde{\mathcal{O}}\left(B\sqrt{\frac{S\widetilde{P}_h^m(V_{h+1}^m - P_h^m V_{h+1}^m)^2}{\mathbf{N}_h^m}} + \frac{SB^2}{\mathbf{N}_h^m}\right) + B^2\Delta_{P,m}$$

$$\qquad \text{([Lemma 4](#) and Cauchy-Schwarz inequality)}$$

$$\leq \mathbb{V}(P_h^m, V_{h+1}^m) + \tilde{\mathcal{O}}\left(B\sqrt{\frac{S\mathbb{V}(P_h^m, V_{h+1}^m)}{\mathbf{N}_h^m}} + B^2\sqrt{\frac{S\Delta_{P,m}}{\mathbf{N}_h^m}} + \frac{SB^2}{\mathbf{N}_h^m}\right) + B^2\Delta_{P,m}$$

$$\leq 2\mathbb{V}(P_h^m, V_{h+1}^m) + \tilde{\mathcal{O}}\left(\frac{SB^2}{\mathbf{N}_h^m}\right) + 2B^2\Delta_{P,m}. \qquad \text{(AM-GM inequality)}$$

$\qquad\square$

**Lemma 8.** *Given an oblivious set of value functions $\mathcal{V}$ with $|\mathcal{V}| \le (2HK)^6$ and $\|V\|_\infty \le B$ for any $V \in \mathcal{V}$, we have with probability at least $1 - \delta$, for any $V \in \mathcal{V}$, $(s,a) \in \mathcal{S} \times \mathcal{A}$, and $m \ge 1$, $|(\bar{P}^m_{s,a} - \widetilde{P}^m_{s,a})V| \le \sqrt{\frac{\mathbb{V}(P^m_{s,a},V)\iota_m}{\mathbf{N}^+_m(s,a)}} + \frac{17B\iota_m}{\mathbf{N}^+_m(s,a)} + \frac{B\Delta_{P,m}}{64}$ and $|(\bar{P}^m_{s,a} - \widetilde{P}^m_{s,a})V| \le \sqrt{\frac{2\mathbb{V}(\bar{P}^m_{s,a},V)\iota_m}{\mathbf{N}^+_m(s,a)}} + \frac{3B\sqrt{S}\iota_m}{\mathbf{N}^+_m(s,a)}$.*

*Proof.* For each $(s,a) \in \mathcal{S} \times \mathcal{A}$ and $V \in \mathcal{V}$, by Lemma 49, with probability at least $1 - \frac{\delta}{2SA(2HK)^6}$, for any $m \ge 1$,

$$|(\bar{P}^m_{s,a} - \widetilde{P}^m_{s,a})V| \le \frac{1}{\mathbf{N}^+_m(s,a)}\left( \sqrt{\sum_{i=1}^{\mathbf{N}_m(s,a)} \mathbb{V}(P^{m_i}_{s,a},V)\iota_m} + B\iota_m \right). \tag{3}$$

Denote by $m_i$ the interval where the $i$-th visits to $(s,a)$ lies in among those $\mathbf{N}_m(s,a)$ visits, we have

$$\frac{1}{\mathbf{N}^+_m(s,a)} \sum_{i=1}^{\mathbf{N}_m(s,a)} \mathbb{V}(P^{m_i}_{s,a},V) = \frac{1}{\mathbf{N}^+_m(s,a)} \sum_{i=1}^{\mathbf{N}_m(s,a)} P^{m_i}_{s,a}(V - P^{m_i}_{s,a}V)^2$$

$$\le \frac{1}{\mathbf{N}^+_m(s,a)} \sum_{i=1}^{\mathbf{N}_m(s,a)} P^{m_i}_{s,a}(V - P^m_{s,a}V)^2 \le \mathbb{V}(P^m_{s,a},V) + B^2\Delta_{P,m},$$

where the second last inequality is by $\frac{\sum_i p_i x_i}{\sum_i p_i} = \text{argmin}_z \sum_i p_i(x_i - z)^2$. Thus by Eq. (3),

$$|(\bar{P}^m_{s,a} - \widetilde{P}^m_{s,a})V| \le \sqrt{\frac{\mathbb{V}(P^m_{s,a},V)\iota_m}{\mathbf{N}^+_m(s,a)}} + \frac{B\iota_m}{\mathbf{N}^+_m(s,a)} + B\sqrt{\frac{\Delta_{P,m}\iota_m}{\mathbf{N}^+_m(s,a)}}$$

$$\le \sqrt{\frac{\mathbb{V}(P^m_{s,a},V)\iota_m}{\mathbf{N}^+_m(s,a)}} + \frac{17B\iota_m}{\mathbf{N}^+_m(s,a)} + \frac{B\Delta_{P,m}}{64}. \qquad \text{(AM-GM inequality)}$$

Moreover, again by $\frac{\sum_i p_i x_i}{\sum_i p_i} = \text{argmin}_z \sum_i p_i(x_i - z)^2$,

$$\frac{1}{\mathbf{N}^+_m(s,a)} \sum_{i=1}^{\mathbf{N}_m(s,a)} \mathbb{V}(P^{m_i}_{s,a},V) \le \frac{1}{\mathbf{N}^+_m(s,a)} \sum_{i=1}^{\mathbf{N}_m(s,a)} P^{m_i}_{s,a}(V - \bar{P}^m_{s,a}V)^2$$

$$\le \mathbb{V}(\bar{P}^m_{s,a},V) + (\widetilde{P}^m_{s,a} - \bar{P}^m_{s,a})(V - \bar{P}^m_{s,a}V)^2 \le \mathbb{V}(\bar{P}^m_{s,a},V) + B\sqrt{\frac{S\mathbb{V}(\bar{P}^m_{s,a},V)\iota_m}{\mathbf{N}^+_m(s,a)}} + \frac{SB^2\iota_m}{\mathbf{N}^+_m(s,a)}$$

(Lemma 4 and Cauchy-Schwarz inequality)

$$\le 2\mathbb{V}(\bar{P}^m_{s,a},V) + \frac{2SB^2\iota_m}{\mathbf{N}^+_m(s,a)}. \qquad \text{(AM-GM inequality)}$$

Thus by Eq. (3), $|(\bar{P}^m_{s,a} - \widetilde{P}^m_{s,a})V| \le \sqrt{\frac{2\mathbb{V}(\bar{P}^m_{s,a},V)\iota_m}{\mathbf{N}^+_m(s,a)}} + \frac{3B\sqrt{S}\iota_m}{\mathbf{N}^+_m(s,a)}$. $\qquad\square$

**Lemma 9.** *For any sequence of value functions $\{V^m_h\}_{m,h}$ with $\|V^m_h\|_\infty \in [0,B]$, we have with probability at least $1 - \delta$, for all $M' \ge 1$, $\sum_{m=1}^{M'} \sum_{h=1}^{H_m} \mathbb{V}(P^m_h,V^m_{h+1}) = \tilde{\mathcal{O}}\left( \sum_{m=1}^{M'} V^m_{H_m+1}(s^m_{H_m+1})^2 + \sum_{m=1}^{M'} \sum_{h=1}^{H_m} B(V^m_h(s^m_h) - P^m_h V^m_{h+1})_+ + B^2 \right)$.*

*Proof.* We decompose the sum of variance as follows:

$$\sum_{m=1}^{M'} \sum_{h=1}^{H_m} \mathbb{V}(P^m_h,V^m_{h+1}) = \sum_{m=1}^{M'} \sum_{h=1}^{H_m} \left( P^m_h(V^m_{h+1})^2 - V^m_{h+1}(s^m_{h+1})^2 \right)$$

$$+ \sum_{m=1}^{M'} \sum_{h=1}^{H_m} \left( V^m_{h+1}(s^m_{h+1})^2 - V^m_h(s^m_h)^2 \right) + \sum_{m=1}^{M'} \sum_{h=1}^{H_m} \left( V^m_h(s^m_h)^2 - (P^m_h V^m_{h+1})^2 \right).$$

For the first term, by Lemma 49 and Lemma 47, with probability at least $1 - \delta$,

$$\sum_{m=1}^{M'} \sum_{h=1}^{H_m} \left( P_h^m (V_{h+1}^m)^2 - V_{h+1}^m(s_{h+1}^m)^2 \right) = \tilde{\mathcal{O}} \left( \sqrt{\sum_{m=1}^{M'} \sum_{h=1}^{H_m} \mathbb{V}(P_h^m, (V_{h+1}^m)^2) + B^2} \right)$$

$$= \tilde{\mathcal{O}} \left( B \sqrt{\sum_{m=1}^{M'} \sum_{h=1}^{H_m} \mathbb{V}(P_h^m, V_{h+1}^m) + B^2} \right).$$

The second term is clearly upper bounded by $\sum_{m=1}^{M'} V_{H_m+1}^m(s_{H_m+1}^m)^2$, and the third term is upper bounded by $2B \sum_{m=1}^{M'} \sum_{h=1}^{H_m} (V_h^m(s_h^m) - P_h^m V_{h+1}^m)_+$ by $a^2 - b^2 \leq (a+b)(a-b)_+$. Putting everything together and solving a quadratic inequality (Lemma 45) w.r.t $\sum_{m=1}^{M'} \sum_{h=1}^{H_m} \mathbb{V}(P_h^m, V_{h+1}^m)$ completes the proof. □

**Lemma 10.** *For any value functions $\{V_h^m\}_{m,h}$ such that $\|V_h^m\|_\infty \leq B$, with probability at least $1 - \delta$, for any $M' \geq 1$,*

$$\sum_{m=1}^{M'} \sum_{h=1}^{H_m} b^m(s_h^m, a_h^m, V_{h+1}^m)$$

$$= \tilde{\mathcal{O}} \left( \sqrt{SAL_{P,M'} \sum_{m=1}^{M'} \sum_{h=1}^{H_m} \mathbb{V}(P_h^m, V_{h+1}^m) + BS^{1.5} AL_{P,M'} + B \sqrt{SAL_{P,M'} \sum_{m=1}^{M'} \sum_{h=1}^{H_m} \Delta_{P,m}}} \right).$$

*Proof.* Note that:

$$\sum_{m=1}^{M'} \sum_{h=1}^{H} b^m(s_h^m, a_h^m, V_{h+1}^m) = \tilde{\mathcal{O}} \left( \sum_{m=1}^{M'} \sum_{h=1}^{H_m} \left( \sqrt{\frac{\mathbb{V}(\bar{P}_h^m, V_{h+1}^m)}{\mathbf{N}_h^m}} + \frac{B\sqrt{S}}{\mathbf{N}_h^m} \right) \right)$$

$$= \tilde{\mathcal{O}} \left( \sqrt{SAL_{P,M'} \sum_{m=1}^{M'} \sum_{h=1}^{H_m} \mathbb{V}(\bar{P}_h^m, V_{h+1}^m) + BS^{1.5} AL_{P,M'}} \right)$$

(Cauchy-Schwarz inequality and Lemma 11)

$$= \tilde{\mathcal{O}} \left( \sqrt{SAL_{P,M'} \sum_{m=1}^{M'} \sum_{h=1}^{H_m} \mathbb{V}(P_h^m, V_{h+1}^m) + BS^{1.5} AL_{P,M'} + B \sqrt{SAL_{P,M'} \sum_{m=1}^{M'} \sum_{h=1}^{H_m} \Delta_{P,m}}} \right).$$

(Lemma 7, Lemma 11, and $\sqrt{a+b} \leq \sqrt{a} + \sqrt{b}$)

□

**Lemma 11.** *For any $M' \geq 1$, $\sum_{m=1}^{M'} \sum_{h=1}^{H_m} \frac{1}{\mathbf{M}_h^m} = \tilde{\mathcal{O}}(SAL_{c,M'})$ and $\sum_{m=1}^{M'} \sum_{h=1}^{H_m} \frac{1}{\mathbf{N}_h^m} = \tilde{\mathcal{O}}(SAL_{P,M'})$.*

*Proof.* This simply follows from the fact that the sum of $\frac{1}{\mathbf{M}_h^m}$ (or $\frac{1}{\mathbf{N}_h^m}$) between consecutive resets of $\mathbf{M}_h^m$ (or $\mathbf{N}_h^m$) is of order $\tilde{\mathcal{O}}(SA)$. □

**Lemma 12.** *$\sum_{m=1}^{M'} \mathbb{I}\{H_m < H, s_{H_m+1}^m \neq g\} = \tilde{\mathcal{O}}(SAL_{M'})$ for any $M' \leq M$.*

*Proof.* This simply follows from the fact that between consecutive resets of $\mathbf{M}$ or $\mathbf{N}$, the number of times that the number of visits to some $(s,a)$ is doubled is $\tilde{\mathcal{O}}(SA)$. □

**Lemma 13.** *Suppose $r(m) = \min\{\frac{c_1}{\sqrt{m}} + c_2, c_3\}$, $\Delta \in \mathbb{R}_+^{\mathbb{N}_+}$ is a non-stationarity measure, and define $\Delta_{[i,j]} = \sum_{i=1}^{j-1} \Delta(i)$. If for a given interval $\mathcal{J}$, there is a way to partition $\mathcal{J}$ into $\ell$ intervals $\{\mathcal{I}_i\}_{i=1}^\ell$ with $\mathcal{I}_i = [s_i, e_i]$ such that $\Delta_{[s_i, e_i+1]} > r(|\mathcal{I}_i|+1)$ for $i \leq \ell - 1$ (note that $|\mathcal{I}_i| = e_i - s_i + 1$), then $\ell \leq 1 + (2c_1^{-1}\Delta_{\mathcal{J}})^{2/3} |\mathcal{J}|^{1/3} + c_3^{-1} \Delta_{\mathcal{J}}$.*

*Proof.* Note that

$$\Delta_{\mathcal{J}} \geq \sum_{i=1}^{\ell-1} \Delta_{[s_i, e_i+1]} > \sum_{i=1}^{\ell-1} r(|\mathcal{I}_i| + 1) \geq \sum_{i=1}^{\ell-1} \min\left\{ c_1(|\mathcal{I}_i| + 1)^{-1/2}, c_3 \right\}$$

$$\geq \sum_{i=1}^{\ell-1} \min\left\{ \frac{c_1}{2} |\mathcal{I}_i|^{-1/2}, c_3 \right\} = \sum_{i=1}^{\ell_1} \frac{c_1}{2} |\mathcal{I}_i|^{-1/2} + \ell_2 c_3,$$

where in the last step we assume $|\mathcal{I}_i|$ is decreasing in $i$ without loss of generality and $\ell_1 + \ell_2 = \ell - 1$. The inequality above implies $\ell_2 \leq c_3^{-1} \Delta_{\mathcal{J}}$ and

$$\ell_1 = \sum_{i=1}^{\ell_1} |\mathcal{I}_i|^{-\frac{1}{3}} |\mathcal{I}_i|^{\frac{1}{3}} \leq \left( \sum_{i=1}^{\ell_1} |\mathcal{I}_i|^{-1/2} \right)^{\frac{2}{3}} \left( \sum_{i=1}^{\ell_1} |\mathcal{I}_i| \right)^{\frac{1}{3}} \leq \left( \frac{2\Delta_{\mathcal{J}}}{c_1} \right)^{\frac{2}{3}} |\mathcal{J}|^{\frac{1}{3}}$$

(Hölder's inequality with $p = \frac{3}{2}$ and $q = 3$)

Combining them completes the proof. $\square$

# B    Omitted Details in Section 3

In this section we provide omitted proofs and discussions in Section 3.

## B.1    Optimal Value Change w.r.t Non-stationarity

Below we provide a bound on the change of optimal value functions w.r.t cost and transition non-stationarity.

**Lemma 14.** *For any* $k_1, k_2 \in [K]$, $V_{k_1}^\star(s_{init}) - V_{k_2}^\star(s_{init}) \leq (\Delta_c + B_\star \Delta_P) T_\star$.

*Proof.* Denote by $q_{k_2}^\star(s, a)$ (or $q_{k_2}^\star(s)$) the number of visits to $(s, a)$ (or $s$) before reaching $g$ following $\pi_{k_2}^\star$. By the extended value difference lemma [Shani et al., 2020, Lemma 1] (note that their result is for finite-horizon MDP, but the nature generalization to SSP holds), we have

$$V_{k_1}^\star(s_{\text{init}}) - V_{k_2}^\star(s_{\text{init}})$$
$$= \sum_s q_{k_2}^\star(s)(V_{k_1}^\star(s) - Q_{k_1}^\star(s, \pi_{k_2}^\star(s))) + \sum_{s,a} q_{k_2}^\star(s, a)(Q_{k_1}^\star(s, a) - c_{k_2}(s, a) - P_{k_2,s,a} V_{k_1}^\star)$$
$$\leq \sum_{s,a} q_{k_2}^\star(s, a)(c_{k_1}(s, a) - c_{k_2}(s, a) + (P_{k_1,s,a} - P_{k_2,s,a})V_{k_1}^\star) \leq (\Delta_c + B_\star \Delta_P) T_\star.$$

where in the last inequality we apply $\|c_{k_1} - c_{k_2}\|_\infty \leq \Delta_c, (P_{k_1,s,a} - P_{k_2,s,a})V_{k_1}^\star \leq \max_{s,a} \|P_{k_1,s,a} - P_{k_2,s,a}\|_1 \|V_{k_1}^\star\|_\infty \leq B_\star \Delta_P$, and $\sum_{s,a} q_{k_2}^\star(s, a) \leq T_\star$. $\square$

We also give an example showing that the bound in Lemma 14 is tight up to a multiplication factor. Consider an SSP instance with only one state $s_{\text{init}}$ and one action $a_g$, such that $c(s_{\text{init}}, a_g) = \frac{B_\star}{T_\star}$, $P(g|s_{\text{init}}, a_g) = \frac{1}{T_\star}$, and $P(s_{\text{init}}|s_{\text{init}}, a_g) = 1 - P(g|s_{\text{init}}, a_g)$ with $1 \leq B_\star \leq T_\star$. The optimal value of this instance is clearly $B_\star$. Now consider another SSP instance with perturbed cost function $c'(s_{\text{init}}, a_g) = \frac{B_\star}{T_\star} + \Delta_c$ and perturbed transition function $P'(g|s_{\text{init}}, a_g) = \frac{1}{T_\star} - \frac{\Delta_P}{2}$, $P'(s_{\text{init}}|s_{\text{init}}, a_g) = 1 - P'(g|s_{\text{init}}, a_g)$ with $\max\{\Delta_c, \Delta_P\} \leq \frac{1}{T_\star}$. The optimal value function in this instance is

$$\frac{\frac{B_\star}{T_\star} + \Delta_c}{\frac{1}{T_\star} - \frac{\Delta_P}{2}} = \frac{B_\star + T_\star \Delta_c}{1 - \frac{T_\star \Delta_P}{2}} \leq (B_\star + T_\star \Delta_c)(1 + T_\star \Delta_P) = B_\star + (\Delta_c + B_\star \Delta_P)T_\star + T_\star^2 \Delta_c \Delta_P$$
$$\leq B_\star + 2(\Delta_c + B_\star \Delta_P)T_\star,$$

where in the first inequality we apply $\frac{1}{1-x} \leq 1 + 2x$ for $x \in [0, \frac{1}{2}]$. Thus the optimal value difference between these two SSPs is of the same order of the upper bound in Lemma 14.

## B.2 Proof of Theorem 1

For the ease of analysis, in this section we consider SSP instances with different action set at different state similar to [Chen et al., 2021b]. The meaning of $SA$ is still the total number of state-action pairs in the SSP instance.

For any $B_\star, T_\star, SA, K$ with $B_\star \geq 1$, $T_\star \geq 3B_\star$, and $K \geq SA \geq 10$, we define a set of SSP instances $\{\mathcal{M}_{i,j}^K\}_{i,j}$ with $i, j \in \{0, 1, \ldots, N\}$ and $N = SA$. The instance $\mathcal{M}_{i^\star, j^\star}^K$ is constructed as follows:

- There are $N + 1$ states $\{s_{\text{init}}, s_1, \ldots, s_N\}$.
- At $s_{\text{init}}$, there are $N$ actions $a_1, \ldots, a_N$; at $s_i$ for $i \in [N]$ there is only one action $a_g$.
- $c(s_{\text{init}}, a_i) = 0$ and $c(s_i, a_g) \sim \text{Bernoulli}(\frac{B_\star + \epsilon_{c,K} \mathbb{I}\{i \neq i^\star\}}{T_\star})$ for $i \in [N]$, where $\epsilon_{c,K} = \frac{1-1/N}{4}\sqrt{NB_\star/K}$.
- $P(s_i | s_{\text{init}}, a_i) = 1$, $P(g | s_j, a_g) = \frac{1 + \epsilon_{P,K} \mathbb{I}\{j = j^\star\}}{T_\star}$, and $P(s_j | s_j, a_g) = 1 - P(g | s_j, a_g)$, where $\epsilon_{P,K} = \frac{1-1/N}{4}\sqrt{N/K}$.

Note that for any $\mathcal{M}_{i,j}^K$, the expected hitting time is upper bounded by $T_\star + 1$, the expected cost of optimal policy is upper bounded by $2B_\star$, and the number of state-action pairs is upper bounded by $2N$. We then use $\{\mathcal{M}_{i,j}^K\}_{i,j}$ to prove static regret lower bounds (note that static regret and dynamic regret are the same without non-stationarity, that is, $\Delta_c = \Delta_P = 0$) based on cost perturbation and transition perturbation respectively, which serve as the cornerstones of the proof of Theorem 1.

**Theorem 7.** *For any $B_\star, T_\star, SA, K$ with $B_\star \geq 1$, $T_\star \geq 3B_\star$, $K \geq SA \geq 10$, and any learner, there exists an SSP instance based on cost perturbation such that the regret of the learner after $K$ episodes is at least $\Omega(\sqrt{B_\star SAK})$.*

*Proof.* Consider a distribution of SSP instances which is uniform over $\{\mathcal{M}_{i,0}^K\}_i$ for $i \in [N]$. Let $\mathbb{E}_i$ be the expectation w.r.t $\mathcal{M}_{i,0}^K$, $P_i$ be the distribution of learner's observations w.r.t $\mathcal{M}_{i,0}^K$, and $K_i$ the number of visits to state $i$ in $K$ episodes. Also let $\epsilon_c = \epsilon_{c,K}$. The expected regret over this distribution of SSPs can be lower bounded as

$$\mathbb{E}[R_K] = \frac{1}{N}\sum_{i=1}^N \mathbb{E}_i[R_K] \geq \frac{1}{N}\sum_{i=1}^N \mathbb{E}_i[K - K_i]\epsilon_c = \epsilon_c\left(K - \frac{1}{N}\sum_{i=1}^N \mathbb{E}_i[K_i]\right).$$

Note that $\mathcal{M}_{0,0}^K$ has no "good" state. By Pinsker's inequality:

$$\mathbb{E}_i[K_i] - \mathbb{E}_0[K_i] \leq K\|P_i - P_0\|_1 \leq K\sqrt{2\text{KL}(P_0, P_i)}.$$

By the divergence decomposition lemma [Lattimore and Szepesvári, 2020, Lemma 15.1], we have:

$$\text{KL}(P_0, P_i) = \mathbb{E}_0[K_i] \cdot T_\star \cdot \text{KL}(\text{Bernoulli}((B_\star + \epsilon_c)/T_\star), \text{Bernoulli}(B_\star/T_\star))$$
$$\leq \mathbb{E}_0[K_i] \cdot T_\star \cdot \frac{\epsilon_c^2/T_\star^2}{\frac{B_\star}{T_\star}(1 - \frac{B_\star}{T_\star})} \leq \frac{2\epsilon_c^2}{B_\star}\mathbb{E}_0[K_i].$$
$$(\text{[Gerchinovitz and Lattimore, 2016, Lemma 6]})$$

Therefore, by Cauchy-Schwarz inequality,

$$\sum_{i=1}^N \mathbb{E}_i[K_i] \leq \sum_{i=1}^N \left(\mathbb{E}_0[K_i] + 2\epsilon_c K\sqrt{\mathbb{E}_0[K_i]/B_\star}\right) \leq K + 2\epsilon_c K\sqrt{NK/B_\star}.$$

Plugging this back and by the definition of $\epsilon_c$, we obtain

$$\mathbb{E}[R_K] \geq \epsilon_c K\left(1 - \frac{1}{N} - 2\epsilon_c\sqrt{\frac{K}{NB_\star}}\right) = \frac{(1-1/N)^2}{8}\sqrt{B_\star NK} = \Omega(\sqrt{B_\star SAK}).$$

This completes the proof. $\qquad\square$

**Theorem 8.** *For any $B_\star, T_\star, SA, K$ with $B_\star \geq 1$, $T_\star \geq 3B_\star$, $K \geq SA \geq 10$, and any learner, there exists an SSP instance based on transition perturbation such that the regret of the learner after $K$ episodes is at least $\Omega(B_\star\sqrt{SAK})$.*

*Proof.* Consider a distribution of SSP instances which is uniform over $\{\mathcal{M}_{0,j}^K\}_j$ for $j \in [N]$. Let $\mathbb{E}_j$ be the expectation w.r.t $\mathcal{M}_{0,j}^K$, $P_j$ be the distribution of learner's observations w.r.t $\mathcal{M}_{0,j}^K$, and $K_j$ the number of visits to state $j$ in $K$ episodes. Also let $\epsilon_P = \epsilon_{P,K}$. The expected regret over this distribution of SSPs can be lower bounded as

$$\mathbb{E}[R_K] = \frac{1}{N}\sum_{j=1}^N \mathbb{E}_j[R_K] \geq \frac{1}{N}\sum_{j=1}^N \mathbb{E}_j[K - K_j] \cdot B_\star\left(1 - \frac{1}{1+\epsilon_P}\right)$$

$$\geq \frac{B_\star\epsilon_P}{2}\left(K - \frac{1}{N}\sum_{j=1}^N \mathbb{E}_j[K_j]\right).$$

Note that $\mathcal{M}_{0,0}^K$ has no "good" state. By Pinsker's inequality:

$$\mathbb{E}_j[K_j] - \mathbb{E}_0[K_j] \leq K\|P_j - P_0\|_1 \leq K\sqrt{2\mathrm{KL}(P_0, P_j)}.$$

By the divergence decomposition lemma [Lattimore and Szepesvári, 2020, Lemma 15.1], we have:

$$\mathrm{KL}(P_0, P_j) = \mathbb{E}_0[K_j] \cdot \mathrm{KL}(\mathrm{Geometric}(1/T_\star), \mathrm{Geometric}((1+\epsilon_P)/T_\star))$$
$$= \mathbb{E}_0[K_j] \cdot T_\star \cdot \mathrm{KL}(\mathrm{Bernoulli}(1/T_\star), \mathrm{Bernoulli}((1+\epsilon_P)/T_\star))$$
$$\leq \mathbb{E}_0[K_j] \cdot T_\star \cdot \frac{\epsilon_P^2/T_\star^2}{\frac{1+\epsilon_P}{T_\star}(1 - \frac{1+\epsilon_P}{T_\star})} \leq 2\epsilon_P^2 \mathbb{E}_0[K_j].$$
$$\text{([Gerchinovitz and Lattimore, 2016, Lemma 6] and } \epsilon_P \leq \tfrac{1}{4})$$

Therefore, by Cauchy-Schwarz inequality,

$$\sum_{j=1}^N \mathbb{E}_j[K_j] \leq \sum_{j=1}^N \left(\mathbb{E}_0[K_j] + 2\epsilon_P K\sqrt{\mathbb{E}_0[K_j]}\right) \leq K + 2\epsilon_P K\sqrt{NK}.$$

Plugging this back and by the definition of $\epsilon_P$, we obtain

$$\mathbb{E}[R_K] \geq \frac{B_\star\epsilon_P K}{2}\left(1 - \frac{1}{N} - 2\epsilon_P\sqrt{\frac{K}{N}}\right) \geq \frac{(1-1/N)^2}{16}B_\star\sqrt{NK} = \Omega(B_\star\sqrt{SAK}).$$

This completes the proof. $\qquad\square$

Now we are ready to prove Theorem 1.

*Proof of Theorem 1.* We construct a hard non-stationary SSP instance as follows: we divide $K$ episodes into $L = L_c + L_P$ epochs. Each of the first $L_c$ epochs has length $\frac{K}{2L_c}$, and the corresponding SSP is uniformly sampled from $\{\mathcal{M}_{i,0}^{K/(2L_c)}\}_{i\in[N]}$ independently; each of the last $L_P$ epochs has length $\frac{K}{2L_P}$, and the corresponding SSP is uniformly sampled from $\{\mathcal{M}_{0,j}^{K/(2L_P)}\}_{j\in[N]}$ independently. By Theorem 7 and Theorem 8, the regrets in each of the first $L_c$ epochs and each of the last $L_P$ epochs are of order $\Omega(\sqrt{B_\star SAK/L_c})$ and $\Omega(B_\star\sqrt{SAK/L_P})$ respectively. Moreover, the total change in cost and transition functions are upper bounded by $\frac{\epsilon_c L_c}{T_\star}$ and $\frac{2\epsilon_P L_P}{T_\star}$ respectively with $\epsilon_c = \epsilon_{c,\frac{K}{2L_c}}$ and $\epsilon_P = \epsilon_{P,\frac{K}{2L_P}}$. Now let $\frac{\epsilon_c L_c}{T_\star} = \Delta_c$ and $\frac{2\epsilon_P L_P}{T_\star} = \Delta_P$, we have $L_c = (\frac{4\Delta_c T_\star}{1-1/N})^{2/3}(\frac{K}{2NB_\star})^{1/3}$ and $L_P = (\frac{2\Delta_P T_\star}{1-1/N})^{2/3}(\frac{K}{2N})^{1/3}$, and the dynamic regret is of order $\Omega(L_c \cdot \sqrt{B_\star SAK/L_c} + L_P \cdot B_\star\sqrt{SAK/L_P}) = \Omega((B_\star SAT_\star(\Delta_c + B_\star^2\Delta_P))^{1/3}K^{2/3})$. $\qquad\square$

## C Omitted Details in Section 4

**Notations**    Under the protocol of Algorithm 1, for any $k \in [K]$, denote by $M_k$ the number of intervals in the first $k$ episodes. Clearly, $M = M_K$.

The following lemma is a more general version of Lemma 1.

**Lemma 15.** *For any $K' \in [K]$, $R_{K'} \le \mathring{R}_{M_{K'}} + B_\star$.*

*Proof.* Let $\mathcal{I}_k$ be the set of intervals in episode $k$. Then the regret in episode $k$ satisfies

$$\sum_{m \in \mathcal{I}_k} \sum_{h=1}^{H_m} c_h^m - V_k^\star(s_1^k) = \sum_{m \in \mathcal{I}_k} \left( \sum_{h=1}^{H_m} c_h^m - V_1^{\pi^\star, m}(s_1^m) \right) + \sum_{m \in \mathcal{I}_k} V_1^{\pi^\star, m}(s_1^m) - V_k^\star(s_1^k)$$

$$\le \sum_{m \in \mathcal{I}_k} (C^m - V_1^{\pi^\star, m}(s_1^m)) + \frac{B_\star}{2K},$$

where the last step is by the definition of $c_{H_m+1}^m$ and $V_1^{\pi^\star, m}(s_1^m) \le V_k^\star(s_1^m) + \frac{B_\star}{2K} \le \frac{3}{2} B_\star$ by Lemma 46. Summing up over $k$ completes the proof. $\square$

**Lemma 16.** *Suppose algorithm $\mathfrak{A}$ ensures $\mathring{R}_{M'} = \tilde{\mathcal{O}}(\gamma_0 + \gamma_1 M'^{1/3} + \gamma_{\frac{1}{2}} M'^{1/2} + \gamma_2 M'^{2/3})$ for any number of intervals $M' \le M$ with cetain probability. Then with the same probability, $M_{K'} = \tilde{\mathcal{O}}(K' + \gamma_0/B_\star + (\gamma_1/B_\star)^{3/2} + (\gamma_{\frac{1}{2}}/B_\star)^2 + (\gamma_2/B_\star)^3)$ and $\mathring{R}_{M_{K'}} = \tilde{\mathcal{O}}(\gamma_1 K'^{1/3} + \gamma_{\frac{1}{2}} K'^{1/2} + \gamma_2 K'^{2/3} + \gamma_1^{3/2}/B_\star^{1/2} + \gamma_{\frac{1}{2}}^2/B_\star + \gamma_2^3/B_\star^2 + \gamma_0)$ for any $K' \in [K]$.*

*Proof.* Fix a $K' \in [K]$. For any $M' \le M_{K'}$, let $\mathcal{C}_g = \{m \in [M'] : s_{H_m+1}^m = g\}$. Then,

$$\mathring{R}_{M'} = \sum_{m \in \mathcal{C}_g} (C^m - V_1^{\pi^\star, m}(s_1^m)) + \sum_{m \notin \mathcal{C}_g} (C^m - V_1^{\pi^\star, m}(s_1^m))$$

$$= \tilde{\mathcal{O}} \left( \gamma_0 + \gamma_1 M'^{1/3} + \gamma_{\frac{1}{2}} M'^{1/2} + \gamma_2 M'^{2/3} \right). \tag{4}$$

Note that $V_1^{\pi^\star, m}(s_1^m) \le V_{k(m)}^\star(s_1^m) + \frac{B_\star}{2K} \le \frac{3}{2} B_\star$ by Lemma 46. Moreover, $C^m \ge 2B_\star$ when $m \notin \mathcal{C}_g$. Therefore, $C^m - V_1^{\pi^\star, m}(s_1^m) \ge -\frac{3B_\star}{2}$ for $m \in \mathcal{C}_g$ and $C^m - V_1^{\pi^\star, m}(s_1^m) \ge \frac{B_\star}{2}$ for $m \notin \mathcal{C}_g$. Reorganizing terms and by $|\mathcal{C}_g| \le K'$, we get:

$$\frac{B_\star M'}{2} \le 2B_\star K' + \tilde{\mathcal{O}} \left( \gamma_0 + \gamma_1 M'^{1/3} + \gamma_{\frac{1}{2}} M'^{1/2} + \gamma_2 M'^{2/3} \right).$$

Solving a quadratic inequality w.r.t. $M'$, we get $M' = \tilde{\mathcal{O}}(K' + \gamma_0/B_\star + (\gamma_1/B_\star)^{3/2} + (\gamma_{\frac{1}{2}}/B_\star)^2 + (\gamma_2/B_\star)^3)$. Define $\gamma = \gamma_0/B_\star + (\gamma_1/B_\star)^{3/2} + (\gamma_{\frac{1}{2}}/B_\star)^2 + (\gamma_2/B_\star)^3$. Plugging the bound on $M'$ back to Eq. (4), we have

$$\mathring{R}_{M'} = \tilde{\mathcal{O}} \left( \gamma_0 + \gamma_1 K'^{1/3} + \gamma_{\frac{1}{2}} K'^{1/2} + \gamma_2 K'^{2/3} + \gamma_1 \gamma^{1/3} + \gamma_{\frac{1}{2}} \gamma^{1/2} + \gamma_2 \gamma^{2/3} \right)$$

$$= \tilde{\mathcal{O}} \left( \gamma_0 + \gamma_1 K'^{1/3} + \gamma_{\frac{1}{2}} K'^{1/2} + \gamma_2 K'^{2/3} + \gamma_1^{3/2}/B_\star^{1/2} + \gamma_{\frac{1}{2}}^2/B_\star + \gamma_2^3/B_\star^2 + B_\star \gamma \right)$$

$$= \tilde{\mathcal{O}} \left( \gamma_0 + \gamma_1 K'^{1/3} + \gamma_{\frac{1}{2}} K'^{1/2} + \gamma_2 K'^{2/3} + \gamma_1^{3/2}/B_\star^{1/2} + \gamma_{\frac{1}{2}}^2/B_\star + \gamma_2^3/B_\star^2 \right),$$

where in the second last step we apply Young's inequality for product ($xy \le x^p/p + y^q/q$ for $x \ge 0$, $y \ge 0$, $p > 1$, $q > 1$, and $\frac{1}{p} + \frac{1}{q} = 1$). Putting everything together and setting $M' = M_{K'}$ completes the proof. $\square$

## D Omitted Details in Section 5

**Extra Notations**    Let $Q_h^m$, $V_h^m$, $x_m$ be the value of $Q_h$, $V_h$, and $x$ at the beginning of interval $m$, and $Q_{H+1}^m(s, a) = V_{H+1}^m(s)$ for any $(s, a) \in \mathcal{S} \times \mathcal{A}$.

## D.1 Proof of Theorem 2

We first prove two lemmas related to the optimism of $Q_h^m$. Define the following reference value function: $\mathring{Q}_h^m(s,a) = (\widehat{c}^m(s,a) + \bar{P}_{s,a}^m \mathring{V}_{h+1}^m - b^m(s,a,\mathring{V}_{h+1}^m) - \mathring{x}_m)_+$ for $h \in [H]$, where $\mathring{V}_h^m(s) = \operatorname{argmin}_a \mathring{Q}_h^m(s,a)$ for $h \in [H]$, $\mathring{V}_{H+1}^m = c_f$, $\mathring{Q}_{H+1}^m(s,a) = \mathring{V}_{H+1}^m(s)$ for any $(s,a) \in \mathcal{S} \times \mathcal{A}$, and $\mathring{x}_m = \Delta_{c,m} + 4B_\star \Delta_{P,m}$.

**Lemma 17.** *With probability at least $1 - 2\delta$, $\mathring{Q}_h^m(s,a) \leq Q_h^{\star,m}(s,a)$ for $m \leq M$.*

*Proof.* We prove this by induction on $h$. The base case of $h = H + 1$ is clearly true. For $h \leq H$, by Lemma 48, for any $(s,a) \in \mathcal{S} \times \mathcal{A}$:

$$
\begin{aligned}
\mathring{Q}_h^m(s,a) &= \widehat{c}^m(s,a) + \bar{P}_{s,a}^m \mathring{V}_{h+1}^m - b^m(s,a,\mathring{V}_{h+1}^m) - \mathring{x}_m \\
&\leq \widehat{c}^m(s,a) + \bar{P}_{s,a}^m V_{h+1}^{\star,m} - b^m(s,a,V_{h+1}^{\star,m}) - \mathring{x}_m \qquad \text{(by the induction step)} \\
&= \widehat{c}^m(s,a) + \widetilde{P}_{s,a}^m V_{h+1}^{\star,m} + (\bar{P}_{s,a}^m - \widetilde{P}_{s,a}^m)V_{h+1}^{\star,m} - b^m(s,a,V_{h+1}^{\star,m}) - \mathring{x}_m \\
&\overset{(i)}{\leq} \widehat{c}^m(s,a) + \widetilde{P}_{s,a}^m V_{h+1}^{\star,m} - \mathring{x}_m \overset{(ii)}{\leq} c^m(s,a) + P_{s,a}^m V_{h+1}^{\star,m} = Q_h^{\star,m}(s,a),
\end{aligned}
$$

where in (i) we apply Lemma 8 with $|\{V_h^{\star,m}\}_{m,h}| \leq HK + 1$ to obtain $(\bar{P}_{s,a}^m - \widetilde{P}_{s,a}^m)V_{h+1}^{\star,m} - b^m(s,a,V_{h+1}^{\star,m}) \leq 0$; in (ii) we apply Lemma 5, Lemma 2, and the definition of $\mathring{x}_m$. $\qquad\square$

**Lemma 18.** *With probability at least $1 - 2\delta$, $Q_h^m(s,a) \leq Q_h^{\star,m}(s,a) + (\Delta_{c,m} + 4B_\star\Delta_{P,m})(H - h + 1)$ and $x_m \leq \max\{\frac{1}{mH}, 2(\Delta_{c,m} + 4B_\star\Delta_{P,m})\}$.*

*Proof.* The second statement simply follows from Lemma 17, $Q_h^{\star,m}(s,a) \leq Q_h^{\pi^\star,m}(s,a) \leq 4B_\star = B/4$ by Lemma 2, and the computing procedure of $x_m$. We now prove $Q_h^m(s,a) \leq \mathring{Q}_h^m(s,a) + (\Delta_{c,m} + 4B_\star\Delta_{P,m})(H - h + 1)$ by induction on $h$, and the first statement simply follows from $\mathring{Q}_h^m(s,a) \leq Q_h^{\star,m}(s,a)$ (Lemma 17). The statement is clearly true for $h = H + 1$. For $h \leq H$, by the induction step and $\|V_{h+1}^m\|_\infty \leq B/4$ from the update rule, we have $V_{h+1}^m(s) \leq \min\{B/4, \mathring{V}_{h+1}^m(s) + (\Delta_{c,m} + 4B_\star\Delta_{P,m})(H - h)\} \leq \mathring{V}_{h+1}^m(s) + y_{h+1}^m \leq B$ for any $s \in \mathcal{S}_+$, where $y_h^m = \min\{B/4, (\Delta_{c,m} + 4B_\star\Delta_{P,m})(H - h + 1)\}$. Thus,

$$
\begin{aligned}
\bar{P}_{s,a}^m V_{h+1}^m - b^m(s,a,V_{h+1}^m) - x_m &\leq \bar{P}_{s,a}^m(\mathring{V}_{h+1}^m + y_{h+1}^m) - b^m(s,a,\mathring{V}_{h+1}^m + y_{h+1}^m) \\
&\qquad\qquad\qquad\qquad\qquad (\text{Lemma 48 and } x_m \geq 0) \\
&\leq \bar{P}_{s,a}^m \mathring{V}_{h+1}^m - b^m(s,a,\mathring{V}_{h+1}^m) - \mathring{x}_m + (\Delta_{c,m} + 4B_\star^m\Delta_{P,m})(H - h + 1),
\end{aligned}
$$

where in the last inequality we apply definition of $\mathring{x}_m$ and $b^m(s,a,\mathring{V}_{h+1}^m + y_{h+1}^m) = b^m(s,a,\mathring{V}_{h+1}^m)$ since constant offset does not change the variance. Then, $Q_h^m(s,a) \leq \mathring{Q}_h^m(s,a) + (\Delta_{c,m} + 4B_\star\Delta_{P,m})(H - h + 1)$ by the update rule of $Q_h^m$ and the definition of $\mathring{Q}_h^m$. $\qquad\square$

We are now ready to prove the main theorem, from which Theorem 2 is a simple corollary.

**Theorem 9.** *Algorithm 2 ensures with probability at least $1 - 22\delta$, for any $M' \leq M$, $\mathring{R}_{M'} = \tilde{\mathcal{O}}(\sqrt{B_\star SAL_{c,M'}M'} + B_\star\sqrt{SAL_{P,M'}M'} + B_\star SAL_{c,M'} + B_\star S^2 AL_{P,M'} + \sum_{m=1}^{M'}(\Delta_{c,m} + B_\star\Delta_{P,m})H)$.*

*Proof.* Note that with probability at least $1 - 2\delta$:

$$
\mathring{R}_{M'} \leq \sum_{m=1}^{M'} \left( \sum_{h=1}^{H_m} c_h^m + c_{H_m+1}^m - V_1^{\star,m}(s_1^m) \right) \qquad (V_1^{\star,m}(s_1^m) \leq V_1^{\pi^\star,m}(s_1^m))
$$

$$
\leq \sum_{m=1}^{M'} \left( \sum_{h=1}^{H_m} c_h^m + c_{H_m+1}^m - V_1^m(s_1^m) \right) + \sum_{m=1}^{M'} (\Delta_{c,m} + 4B_\star \Delta_{P,m})H \qquad \text{(Lemma 18)}
$$

$$
\leq \sum_{m=1}^{M'} \sum_{h=1}^{H_m} \left( c_h^m + V_{h+1}^m(s_{h+1}^m) - V_h^m(s_h^m) \right) + \sum_{m=1}^{M'} (\Delta_{c,m} + 4B_\star \Delta_{P,m})H + \tilde{\mathcal{O}}\left( B_\star SAL_{M'} \right)
$$

$$
(c_{H_m+1}^m = \tilde{\mathcal{O}}(B_\star) \text{ and Lemma 12})
$$

$$
\leq \sum_{m=1}^{M'} \sum_{h=1}^{H_m} \left( (c_h^m - \widehat{c}_h^m) + (V_{h+1}^m(s_{h+1}^m) - P_h^m V_{h+1}^m) + (P_h^m - \bar{P}_h^m)V_{h+1}^m + b_h^m \right)
$$

$$
+ 2\sum_{m=1}^{M'} (\Delta_{c,m} + 4B_\star \Delta_{P,m})H + \tilde{\mathcal{O}}\left( B_\star SAL_{M'} \right),
$$

where the last step is by the definitions of $V_h^m(s_h^m)$, $x_m \leq \max\{\frac{1}{mH}, 2(\Delta_{c,m} + 4B_\star \Delta_{P,m})\}$ (Lemma 18), $\max\{a,b\} \leq \frac{a+b}{2}$, and $\sum_{m=1}^{M'} \sum_{h=1}^{H_m} \frac{1}{mH} = \tilde{\mathcal{O}}(1)$. Now we bound the first three sums separately. For the first term, with probability at least $1 - 4\delta$,

$$
\sum_{m=1}^{M'} \sum_{h=1}^{H_m} (c_h^m - \widehat{c}_h^m) = \sum_{m=1}^{M'} \sum_{h=1}^{H_m} (c_h^m - c^m(s_h^m, a_h^m)) + \sum_{m=1}^{M'} \sum_{h=1}^{H_m} (c^m(s_h^m, a_h^m) - \widehat{c}_h^m)
$$

$$
\leq \tilde{\mathcal{O}}\left( \sqrt{C_{M'}} + \sqrt{SAL_{c,M'}C_{M'}} + SAL_{c,M'} \right) + 2\sum_{m=1}^{M'} \Delta_{c,m}H. \qquad \text{(Lemma 49 and Lemma 3)}
$$

For the second term, by Lemma 49, with probability at least $1 - \delta$,

$$
\sum_{m=1}^{M'} \sum_{h=1}^{H_m} (V_{h+1}^m(s_{h+1}^m) - P_h^m V_{h+1}^m) = \tilde{\mathcal{O}}\left( \sqrt{\sum_{m=1}^{M'} \sum_{h=1}^{H_m} \mathbb{V}(P_h^m, V_{h+1}^m)} + B_\star \right)
$$

$$
= \tilde{\mathcal{O}}\left( \sqrt{\sum_{m=1}^{M'} \sum_{h=1}^{H_m} \mathbb{V}(P_h^m, V_{h+1}^{\star,m})} + \sqrt{\sum_{m=1}^{M'} \sum_{h=1}^{H_m} \mathbb{V}(P_h^m, V_{h+1}^{\star,m} - V_{h+1}^m)} + B_\star \right),
$$

$$
(\text{VAR}[X + Y] \leq 2(\text{VAR}[X] + \text{VAR}[Y]) \text{ and } \sqrt{a+b} \leq \sqrt{a} + \sqrt{b})
$$

which is dominated by the upper bound of the third term below. For the third term, by $P_h^m V_{h+1}^m \leq \widetilde{P}_h^m V_{h+1}^m + 4B_\star(\Delta_{P,m} + \mathbf{n}_h^m)$, with probability at least $1 - 2\delta$,

$$
\sum_{m=1}^{M'} \sum_{h=1}^{H_m} (P_h^m - \bar{P}_h^m) V_{h+1}^m \leq \sum_{m=1}^{M'} \sum_{h=1}^{H_m} (\widetilde{P}_h^m - \bar{P}_h^m) V_{h+1}^m + \sum_{m=1}^{M'} \sum_{h=1}^{H_m} 4B_\star(\Delta_{P,m} + \mathbf{n}_h^m)
$$

$$
\leq \sum_{m=1}^{M'} \sum_{h=1}^{H_m} \left( (\widetilde{P}_h^m - \bar{P}_h^m) V_{h+1}^{\star,m} + (\widetilde{P}_h^m - \bar{P}_h^m)(V_{h+1}^m - V_{h+1}^{\star,m}) + 4B_\star \mathbf{n}_h^m \right) + \sum_{m=1}^{M'} 4B_\star \Delta_{P,m} H
$$

$$
= \tilde{\mathcal{O}} \left( \sum_{m=1}^{M'} \sum_{h=1}^{H_m} \left( \sqrt{\frac{\mathbb{V}(P_h^m, V_{h+1}^{\star,m})}{\mathbf{N}_h^m}} + \frac{SB_\star}{\mathbf{N}_h^m} + \sqrt{\frac{S\mathbb{V}(P_h^m, V_{h+1}^m - V_{h+1}^{\star,m})}{\mathbf{N}_h^m}} \right) + \sum_{m=1}^{M'} B_\star \Delta_{P,m} H \right)
$$

$$
(\mathbf{n}_h^m \leq \tfrac{1}{\mathbf{N}_h^m}, \text{Lemma 8 with } |\{V_{h+1}^{\star,m}\}_{m,h}| \leq HK + 1, \text{ and Lemma 6})
$$

$$
= \tilde{\mathcal{O}} \left( \sqrt{SAL_{P,M'} \sum_{m=1}^{M'} \sum_{h=1}^{H_m} \mathbb{V}(P_h^m, V_{h+1}^{\star,m})} + \sqrt{S^2 AL_{P,M'} \sum_{m=1}^{M'} \sum_{h=1}^{H_m} \mathbb{V}(P_h^m, V_{h+1}^{\star,m} - V_{h+1}^m)} \right)
$$

$$
+ \tilde{\mathcal{O}} \left( B_\star S^2 AL_{P,M'} + \sum_{m=1}^{M'} B_\star \Delta_{P,m} H \right). \qquad \text{(Cauchy-Schwarz inequality and Lemma 11)}
$$

Moreover, by Lemma 10, with probability at least $1 - \delta$,

$$
\sum_{m=1}^{M'} \sum_{h=1}^{H_m} b_h^m = \tilde{\mathcal{O}} \left( \sqrt{SAL_{P,M'} \sum_{m=1}^{M'} \sum_{h=1}^{H_m} \mathbb{V}(P_h^m, V_{h+1}^m)} + B_\star S^{1.5} AL_{P,M'} + B_\star \sqrt{SAHL_{P,M'} \sum_{m=1}^{M'} \Delta_{P,m}} \right)
$$

$$
= \tilde{\mathcal{O}} \left( \sqrt{SAL_{P,M'} \sum_{m=1}^{M'} \sum_{h=1}^{H_m} \mathbb{V}(P_h^m, V_{h+1}^{\star,m})} + \sqrt{SAL_{P,M'} \sum_{m=1}^{M'} \sum_{h=1}^{H_m} \mathbb{V}(P_h^m, V_{h+1}^m - V_{h+1}^{\star,m})} \right)
$$

$$
+ \tilde{\mathcal{O}} \left( B_\star S^{1.5} AL_{P,M'} + \sum_{m=1}^{M'} B_\star \Delta_{P,m} H \right).
$$

$$
(\mathrm{VAR}[X + Y] \leq 2\mathrm{VAR}[X] + 2\mathrm{VAR}[Y], \sqrt{a+b} \leq \sqrt{a} + \sqrt{b}, \text{ and AM-GM inequality})
$$

which is dominated by the upper bound of the third term above. Putting everything together, we have with probability at least $1 - 11\delta$,

$$
\mathring{R}_{M'} = \tilde{\mathcal{O}} \left( \sqrt{SAL_{c,M'} C_{M'}} + B_\star SAL_{c,M'} + \sqrt{SAL_{P,M'} \sum_{m=1}^{M'} \sum_{h=1}^{H_m} \mathbb{V}(P_h^m, V_{h+1}^{\star,m}) + B_\star S^2 AL_{P,M'}} \right)
$$

$$
+ \tilde{\mathcal{O}} \left( \sqrt{S^2 AL_{P,M'} \sum_{m=1}^{M'} \sum_{h=1}^{H_m} \mathbb{V}(P_h^m, V_{h+1}^{\star,m} - V_{h+1}^m)} + \sum_{m=1}^{M'} (\Delta_{c,m} + B_\star \Delta_{P,m}) H \right)
$$

$$
= \tilde{\mathcal{O}} \left( \sqrt{SAL_{c,M'} C_{M'}} + \sqrt{B_\star SAL_{P,M'} C_{M'}} + B_\star SAL_{c,M'} + B_\star S^2 AL_{P,M'} \right)
$$

$$
+ \tilde{\mathcal{O}} \left( \sum_{m=1}^{M'} (\Delta_{c,m} + B_\star \Delta_{P,m}) H \right). \qquad \text{(Lemma 19, Lemma 20 and AM-GM inequality)}
$$

Note that $\mathring{R}_{M'} = \sum_{m=1}^{M'} (C^m - V_1^{\pi^\star,m}(s_1^m)) \geq C_{M'} - 4B_\star M'$ (Lemma 2). Reorganizing terms and solving a quadratic inequality (Lemma 45) w.r.t $C_{M'}$ gives $C_{M'} = \tilde{\mathcal{O}}(B_\star M')$ ignoring lower order terms. Plugging this back completes the proof. $\qquad \square$

*Proof of Theorem 2.* Note that by by Line 3 and Line 4 of Algorithm 2, we have $L_c \leq \lceil \frac{M'}{W_c} \rceil$, $L_P \leq \lceil \frac{M'}{W_P} \rceil$, and the number of intervals between consecutive resets of $\mathbf{M}$ (or $\mathbf{N}$) are upper bounded

by $W_c$ (or $W_P$), which gives

$$\sum_{m=1}^{M'} (\Delta_{c,m} + B_\star \Delta_{P,m})H \le \sum_{m=1}^{M'} (\Delta_{c,f^c(m)} + B_\star \Delta_{P,f^P(m)})H \le (W_c \Delta_c + B_\star W_P \Delta_P)H$$

Applying Theorem 9 completes the proof. $\qquad\square$

## D.2 Proof of Theorem 3

We first show that Algorithm 3 ensures an anytime regret bound in $\mathring{\mathcal{M}}$.

**Theorem 10.** *With probability at least $1 - 22\delta$, Algorithm 3 ensures for any $M' \le M$,*
$\mathring{R}_{M'} = \tilde{\mathcal{O}}((B_\star SAT_{\max}\Delta_c)^{1/3}M'^{2/3} + B_\star(SAT_{\max}\Delta_P)^{1/3}M'^{2/3} + (B_\star SAT_{\max}\Delta_c)^{2/3}M'^{1/3} + B_\star(S^{2.5}AT_{\max}\Delta_P)^{2/3}M'^{1/3} + (\Delta_c + B_\star \Delta_P)T_{\max})$.

*Proof.* It suffices to prove the desired inequality for $M' \in \{2^n - 1\}_{n \in \mathbb{N}_+}$. Suppose $M' = 2^N - 1$ for some $N \ge 1$. By the doubling scheme, we run Algorithm 2 on intervals $[2^{n-1}, 2^n - 1]$ for $n = 1, \ldots, N$, and the regret on intervals $[2^{n-1}, 2^n - 1]$ is of order $\tilde{\mathcal{O}}((B_\star SAT_{\max}\Delta_c)^{1/3}(2^{n-1})^{2/3} + B_\star(SAT_{\max}\Delta_P)^{1/3}(2^{n-1})^{2/3} + (B_\star SAT_{\max}\Delta_c)^{2/3}(2^{n-1})^{1/3} + B_\star(S^{2.5}AT_{\max}\Delta_c)^{2/3}(2^{n-1})^{1/3} + (\Delta_c + B_\star \Delta_P)T_{\max})$ by Theorem 2 and the choice of $W_c$ and $W_P$. Summing over $n$ completes the proof. $\qquad\square$

*Proof of Theorem 3.* By Lemma 16 and Theorem 10 with $\gamma_0 = (\Delta_c + B_\star \Delta_P)T_{\max}$, $\gamma_1 = (B_\star SAT_{\max}\Delta_c)^{2/3} + B_\star(S^{2.5}AT_{\max}\Delta_P)^{2/3}$, $\gamma_{\frac{1}{2}} = 0$, and $\gamma_2 = (B_\star SAT_{\max}\Delta_c)^{1/3} + B_\star(SAT_{\max}\Delta_P)^{1/3}$, we have $\gamma_1^{3/2}/B_\star^{1/2} = \tilde{\mathcal{O}}(B_\star^{1/2}SAT_{\max}\Delta_c + B_\star S^{2.5}AT_{\max}\Delta_P)$, $\gamma_2^3/B_\star^2 = \tilde{\mathcal{O}}(SAT_{\max}\Delta_c/B_\star^2 + B_\star SAT_{\max}\Delta_P)$, and thus $\mathring{R}_{M_{K'}} = \tilde{\mathcal{O}}((B_\star SAT_{\max}\Delta_c)^{1/3}K'^{2/3} + B_\star(SAT_{\max}\Delta_P)^{1/3}K'^{2/3} + (B_\star SAT_{\max}\Delta_c)^{2/3}K'^{1/3} + B_\star(S^{2.5}AT_{\max}\Delta_P)^{2/3}K'^{1/3} + B_\star^{1/2}SAT_{\max}\Delta_c + B_\star S^{2.5}AT_{\max}\Delta_P)$ for any $K' \in [K]$. Then by Lemma 15, we obtain the same bound as $\mathring{R}_{M_{K'}}$ for $R_{K'}$. $\qquad\square$

## D.3 Auxiliary Lemmas

**Lemma 19.** *With probability at least $1 - 2\delta$, $\sum_{m=1}^{M'} \sum_{h=1}^{H_m} \mathbb{V}(P_h^m, V_{h+1}^{\star,m}) = \tilde{\mathcal{O}}(B_\star C_{M'} + B_\star^2)$ for any $M' \le M$.*

*Proof.* Applying Lemma 9 with $\|V_h^{\star,m}\|_\infty \le 4B_\star$ (Lemma 2), with probability at least $1 - 2\delta$,

$$\sum_{m=1}^{M'} \sum_{h=1}^{H_m} \mathbb{V}(P_h^m, V_{h+1}^{\star,m})$$
$$= \tilde{\mathcal{O}}\left(\sum_{m=1}^{M'} V_{H_m+1}^{\star,m}(s_{H_m+1}^m)^2 + \sum_{m=1}^{M'} \sum_{h=1}^{H_m} B_\star(V_h^{\star,m}(s_h^m) - P_h^m V_{h+1}^{\star,m})_+ + B_\star^2\right)$$
$$= \tilde{\mathcal{O}}\left(B_\star C_{M'} + B_\star^2\right),$$

where in the last step we apply

$$(V_h^{\star,m}(s_h^m) - P_h^m V_{h+1}^{\star,m})_+ \le (Q_h^{\star,m}(s_h^m, a_h^m) - P_h^m V_{h+1}^{\star,m})_+ \le c^m(s_h^m, a_h^m),$$

and $\sum_{m=1}^{M'} \sum_{h=1}^{H_m} c^m(s_h^m, a_h^m) = \tilde{\mathcal{O}}(\sum_{m=1}^{M'} \sum_{h=1}^{H_m} c_h^m)$ by Lemma 50. $\qquad\square$

**Lemma 20.** *With probability at least $1 - 9\delta$, for any $M' \le M$, $\sum_{m=1}^{M'} \sum_{h=1}^{H_m} \mathbb{V}(P_h^m, V_{h+1}^{\star,m} - V_{h+1}^m) = \tilde{\mathcal{O}}(B_\star \sqrt{B_\star SAL_{P,M'}C_{M'}} + B_\star \sqrt{SAL_{c,M'}C_{M'}} + B_\star^2 S^2 AL_{P,M'} + B_\star^2 SAL_{c,M'} + \sum_{m=1}^{M'} B_\star(\Delta_{c,m} + B_\star \Delta_{P,m})H)$.*

*Proof.* Let $z_h^m = \min\{B/4, (\Delta_{c,m} + 4B_\star \Delta_{P,m})H\}\mathbb{I}\{h \le H\}$. By Lemma 18 and $\|V_h^m\|_\infty \le B/4$, we have $V_h^{\star,m}(s) + z_h^m \ge V_h^m(s)$ for all $s \in \mathcal{S}_+$. Moreover, by Lemma 12,

$$\sum_{m=1}^{M'} (V_{H_m+1}^{\star,m}(s_{H_m+1}^m) + z_{H_m+1}^m - V_{H_m+1}^m(s_{H_m+1}^m))^2$$

$$\le \sum_{m=1}^{M'} (z_{H_m+1}^m)^2 \mathbb{I}\{s_{H_m+1}^m = g\} + \tilde{\mathcal{O}}\left(B_\star^2 \sum_{m=1}^{M'} \mathbb{I}\{H_m < H, s_{H_m+1}^m \ne g\}\right)$$

$$= 4B_\star \sum_{m=1}^{M'} (\Delta_{c,m} + 4B_\star \Delta_{P,m})H + \tilde{\mathcal{O}}\left(B_\star^2 SAL_{M'}\right).$$

Also note that

$$(*) = \sum_{m=1}^{M'} B_\star \sum_{h=1}^{H_m} (V_h^{\star,m}(s_h^m) - V_h^m(s_h^m) - P_h^m V_{h+1}^{\star,m} + P_h^m V_{h+1}^m + z_h^m - z_{h+1}^m)_+$$

$$\le \sum_{m=1}^{M'} B_\star \sum_{h=1}^{H_m} \left(c^m(s_h^m, a_h^m) + \widetilde{P}_h^m V_{h+1}^m - V_h^m(s_h^m) + 4B_\star \mathbf{n}_h^m\right)_+ + 2\sum_{m=1}^{M'} B_\star(\Delta_{c,m} + 4B_\star \Delta_{P,m})H$$

$$\quad (V_h^{\star,m}(s_h^m) \le Q_h^{\star,m}(s_h^m, a_h^m), z_h^m \ge z_{h+1}^m, \text{ and } P_h^m V_{h+1}^m \le \widetilde{P}_h^m V_{h+1}^m + 4B_\star(\mathbf{n}_h^m + \Delta_{P,m}))$$

$$\le \sum_{m=1}^{M'} B_\star \sum_{h=1}^{H_m} (c^m(s_h^m, a_h^m) - \widehat{c}_h^m + (\widetilde{P}_h^m - \bar{P}_h^m)V_{h+1}^{\star,m} + (\widetilde{P}_h^m - \bar{P}_h^m)(V_{h+1}^m - V_{h+1}^{\star,m}) + b_h^m)_+$$

$$\quad + 3\sum_{m=1}^{M'} B_\star(\Delta_{c,m} + 4B_\star \Delta_{P,m})H + 4B_\star^2 \sum_{m=1}^{M'} \sum_{h=1}^{H_m} \mathbf{n}_h^m + \tilde{\mathcal{O}}(B_\star),$$

where the last step is by the definitions of $V_h^m(s_h^m)$, $x_m \le \max\{\frac{1}{mH}, 2(\Delta_{c,m} + 4B_\star \Delta_{P,m})\}$ (Lemma 18), $\max\{a, b\} \le \frac{a+b}{2}$, and $\sum_{m=1}^{M'} \sum_{h=1}^{H_m} \frac{1}{mH} = \tilde{\mathcal{O}}(1)$. Now by Lemma 3, Lemma 8, Lemma 6, and $\mathbf{n}_h^m \le \frac{1}{\mathbf{N}_h^m}$, we continue with

$$(*) = \tilde{\mathcal{O}}\left(B_\star\left(\sqrt{SAL_{c,M'}C_{M'}} + SAL_{c,M'} + \sum_{m=1}^{M'}\sum_{h=1}^{H_m}\left(\sqrt{\frac{\mathbb{V}(P_h^m, V_{h+1}^{\star,m})}{\mathbf{N}_h^m}} + \sqrt{\frac{S\mathbb{V}(P_h^m, V_{h+1}^m - V_{h+1}^{\star,m})}{\mathbf{N}_h^m}}\right)\right)\right)$$

$$\quad + \tilde{\mathcal{O}}\left(\sum_{m=1}^{M'}\sum_{h=1}^{H_m}\frac{B_\star^2 S}{\mathbf{N}_h^m} + \sum_{m=1}^{M'} B_\star(\Delta_{c,m} + B_\star \Delta_{P,m})H + B_\star \sum_{m=1}^{M'}\sum_{h=1}^{H_m} b_h^m\right)$$

$$= \tilde{\mathcal{O}}\left(B_\star\sqrt{SAL_{c,M'}C_{M'}} + B_\star SAL_{c,M'} + B_\star\sqrt{SAL_{P,M'}\sum_{m=1}^{M'}\sum_{h=1}^{H_m}\mathbb{V}(P_h^m, V_{h+1}^{\star,m}) + B_\star^2 S^2 AL_{P,M'}}\right)$$

$$\quad + \tilde{\mathcal{O}}\left(B_\star\sqrt{S^2 AL_{P,M'}\sum_{m=1}^{M'}\sum_{h=1}^{H_m}\mathbb{V}(P_h^m, V_{h+1}^{\star,m} - V_{h+1}^m)} + \sum_{m=1}^{M'} B_\star(\Delta_{c,m} + B_\star \Delta_{P,m})H\right),$$

where in the last step we apply Cauchy-Schwarz inequality, Lemma 11, Lemma 10, $\mathrm{VAR}[X + Y] \le 2\mathrm{VAR}[X] + 2\mathrm{VAR}[Y]$, and AM-GM inequality. Finally, by Lemma 19, we continue with

$$(*) = \tilde{\mathcal{O}}\left(B_\star\sqrt{SAL_{c,M'}C_{M'}} + B_\star SAL_{c,M'} + B_\star\sqrt{B_\star SAL_{P,M'}C_{M'}} + B_\star^2 S^2 AL_{P,M'}\right)$$

$$\quad + \tilde{\mathcal{O}}\left(B_\star\sqrt{S^2 AL_{P,M'}\sum_{m=1}^{M'}\sum_{h=1}^{H_m}\mathbb{V}(P_h^m, V_{h+1}^{\star,m} - V_{h+1}^m)} + \sum_{m=1}^{M'} B_\star(\Delta_{c,m} + B_\star \Delta_{P,m})H\right).$$

Applying Lemma 9 on value functions $\{V_h^{\star,m} + z_h^m - V_h^m\}_{m,h}$ (constant offset does not change the variance) and plugging in the bounds above, we have

$$\sum_{m=1}^{M'} \sum_{h=1}^{H_m} \mathbb{V}(P_h^m, V_{h+1}^{\star,m} - V_{h+1}^m) = \sum_{m=1}^{M'} \sum_{h=1}^{H_m} \mathbb{V}(P_h^m, V_{h+1}^{\star,m} + z_h^m - V_{h+1}^m)$$

$$= \tilde{\mathcal{O}}\left(B_\star \sqrt{SAL_{c,M'}C_{M'}} + B_\star^2 SAL_{c,M'} + B_\star \sqrt{B_\star SAL_{P,M'}C_{M'}} + B_\star^2 S^2 AL_{P,M'}\right)$$

$$+ \tilde{\mathcal{O}}\left(B_\star \sqrt{S^2 AL_{P,M'} \sum_{m=1}^{M'} \sum_{h=1}^{H_m} \mathbb{V}(P_h^m, V_{h+1}^{\star,m} - V_{h+1}^m)} + \sum_{m=1}^{M'} B_\star(\Delta_{c,m} + B_\star \Delta_{P,m})H\right).$$

Then solving a quadratic inequality w.r.t $\sum_{m=1}^{M'} \sum_{h=1}^{H_m} \mathbb{V}(P_h^m, V_{h+1}^{\star,m} - V_{h+1}^m)$ (Lemma 45) completes the proof. □

### D.4 Minimax Optimal Bound in Finite-Horizon MDP

Here we give a high level arguments on why Algorithm 2 implies a minimax optimal dynamic regret bound in the finite-horizon setting. To adapt Algorithm 2 to the non-homogeneous finite-horizon setting, we maintain empirical cost and transition functions for each layer $h \in [H]$ and let $c_f(s) = 0$. Following similar arguments and substituting $B_\star, T_{\max}$ by horizon $H$, Theorem 2 implies (ignoring lower order terms)

$$\mathring{R}_{M'} = \tilde{\mathcal{O}}\left(\sqrt{SAH^2/W_c}M' + \sqrt{SAH^3/W_P}M' + (\Delta_c W_c + H\Delta_P W_P)H\right)$$

$$= \tilde{\mathcal{O}}\left(H(SA\Delta_c)^{1/3}M'^{2/3} + (SAH^5\Delta_P)^{1/3}M'^{2/3}\right),$$

where the extra $\sqrt{H}$ dependency in the first two terms comes from estimating the cost and transition functions of each layer independently, and we set $W_c = (SA)^{1/3}(M'/\Delta_c)^{2/3}$, $W_P = (SA/H)^{1/3}(M'/\Delta_P)^{2/3}$. Note that the lower bound construction in [Mao et al., 2021] only make use of non-stationary transition. The lower bound they prove is $\Omega((SA\Delta)^{1/3}(HT)^{2/3})$ (their Theorem 5), which actually matches our upper bound $\tilde{\mathcal{O}}((SAH^5\Delta_P)^{1/3}M'^{2/3})$ for non-stationary transition since $T = M'H$ and $\Delta = H\Delta_P$ by their definition of non-stationarity. It is also straightforward to show that the lower bound for non-stationary cost matches our upper bound following similar arguments in proving Theorem 1.

## E Omitted Details in Section 6

**Notations** Denote by $\rho_m^c$ and $\rho_m^P$ the values of $\rho^c$ and $\rho^P$ at the beginning of interval $m$ respectively, that is, $\rho_m^c = g^c(\nu_m^c)$ and $\rho_m^P = g^P(\nu_m^P)$, where $g^c(m) = \min\{\frac{c_1}{\sqrt{m}}, \frac{1}{2^8 H}\}$ and $g^P(m) = \min\{\frac{c_2}{\sqrt{m}}, \frac{1}{2^8 H}\}$. Denote by $\check{c}^m$ the value of $\check{c}$ at the beginning of interval $m$ and define $\check{c}_h^m = \check{c}(s_h^m, a_h^m)$. Define $\check{Q}_h^{\pi^\star,m}$ and $\check{V}_h^{\pi^\star,m}$ as the action-value function and value function w.r.t cost $c^m + 8\eta_m$, transition $P^m$, and policy $\pi_{k(m)}^\star$; and $C_{[i,j]} = \sum_{m=i}^{j} \sum_{h=1}^{H_m} C^m$. Let $\check{Q}_h^{\star,m}$ and $\check{V}_h^{\star,m}$ be the optimal value functions w.r.t cost function $c^m + 8\eta_m$ and transition function $P^m$. It is not hard to see that they can be defined recursively as follows: $\check{V}_{H+1}^{\star,m} = c_f$ and for $h \leq H$,

$$\check{Q}_h^{\star,m}(s,a) = c^m(s,a) + 8\eta_m + P_{s,a}^m \check{V}_{h+1}^{\star,m}, \qquad \check{V}_h^{\star,m}(s) = \min_a Q_h^{\star,m}(s,a).$$

For notational convenience, define $\check{Q}_{H+1}^m(s,a) = \check{V}_{H+1}^m(s)$, $\check{Q}_{H+1}^{\pi^\star,m}(s,a) = \check{V}_{H+1}^{\pi^\star,m}(s)$, and $\check{Q}_{H+1}^{\star,m}(s,a) = \check{V}_{H+1}^{\star,m}(s)$ for any $(s,a) \in \mathcal{S} \times \mathcal{A}$; let $L_c = L_{c,[1,K]}$ and $L_P = L_{P,[1,K]}$.

**Proof Sketch of Theorem 4** We give a high level idea on the analysis of the main theorem and also point out the key technical challenges. We decompose the regret as follows:

$$
\mathring{R}_K = \sum_{m=1}^{K} (C^m - \check{V}_1^m(s_1^m)) + \sum_{m=1}^{K} (\check{V}_1^m(s_1^m) - \check{V}_1^{\pi^\star,m}(s_1^m)) + 8T_\star \sum_{m=1}^{K} \eta_m
$$

$$
\lesssim \sum_{m=1}^{K} \sum_{h=1}^{H} \left( c_h^m - \widehat{c}_h^m + \check{V}_{h+1}^m(s_{h+1}^m) - \bar{P}_h^m \check{V}_{h+1}^m + b_h^m - 8\eta_m \right) \quad \text{(definition of } \check{V}_h^m(s_h^m))
$$

$$
+ \sum_{m=1}^{K} (\check{V}_1^m(s_1^m) - \check{V}_1^{\pi^\star,m}(s_1^m)) + 8T_\star \sum_{m=1}^{K} \eta_m.
$$

We bound the three terms above separately. For the second term, we first show that $\check{V}_1^m(s_1^m) - \check{V}_1^{\pi^\star,m}(s_1^m) \leq (\Delta_{c,m} + B\Delta_{P,m})T_\star$, where $\Delta_{c,m} = \Delta_{c,[i_m^c,m]}$, $\Delta_{P,m} = \Delta_{P,[i_m^P,m]}$ are the accumulated cost and transition non-stationarity since the last reset respectively. Although proving such a bound is straightforward when $\check{V}_h^m$ is indeed a value function (similar to Lemma 14), it is non-trivial under the UCBVI update rule as the bonus term $b$ depends on the next-step value function and can not be simply treated as part of the cost function. A key step here is to make use of the monotonic property (Lemma 48) of the bonus function; see Lemma 22 for more details. Now by the periodic resets of cost and transition counters (Line 4 and Line 5), the number of intervals between consecutive resets of cost and transition estimation is upper bounded by $W_c$ and $W_P$ respectively. Thus,

$$
\sum_{m=1}^{K} (\Delta_{c,m} + B\Delta_{P,m})T_\star \leq \sum_{m=1}^{K} (\Delta_{c,f^c(m)} + B\Delta_{P,f^P(m)})T_\star \leq (W_c\Delta_c + BW_P\Delta_P)T_\star
$$

$$
= \tilde{\mathcal{O}}\left( (B_\star S A T_\star \Delta_c)^{1/3} K^{2/3} + B_\star (S A T_\star \Delta_P)^{1/3} K^{2/3} + (\Delta_c + B_\star \Delta_P)T_\star \right).
$$

where the last step is simply by the chosen values of $W_c$ and $W_P$.

For the third term, we have:

$$
T_\star \sum_{m=1}^{K} \eta_m \leq T_\star \sum_{m=1}^{K} \left( \frac{c_1}{\sqrt{\nu_m^c}} + \frac{Bc_2}{\sqrt{\nu_m^P}} \right) = \tilde{\mathcal{O}}\left( T_\star \left( c_1 \sum_{i=1}^{L_c} \sqrt{M_i^c} + B_\star c_2 \sum_{i=1}^{L_P} \sqrt{M_i^P} \right) \right)
$$

$$
= \tilde{\mathcal{O}}\left( T_\star (c_1 \sqrt{L_c K} + B_\star c_2 \sqrt{L_P K}) \right) = \tilde{\mathcal{O}}\left( \sqrt{B_\star S A L_c K} + B_\star \sqrt{S A L_P K} \right),
$$

where $M_i^c$ (or $M_i^P$) is the number of intervals between the $i$-th and $(i+1)$-th reset of cost (or transition) estimation, and the second last step is by Cauchy-Schwarz inequality. Finally we bound the first term, simply by **Test 1** and **Test 2**, we have (only keeping the dominating terms)

$$
\sum_{m=1}^{K} \sum_{h=1}^{H} \left( c_h^m - \widehat{c}_h^m + \check{V}_{h+1}^m(s_{h+1}^m) - \bar{P}_h^m \check{V}_{h+1}^m + b_h^m - 8\eta_m \right)
$$

$$
= \sum_{i=1}^{L_c} \sum_{m \in \mathcal{I}_i^c} \sum_{h=1}^{H_m} (c_h^m - \widehat{c}_h^m) + \sum_{i=1}^{L_P} \sum_{m \in \mathcal{I}_i^P} \sum_{h=1}^{H_m} (\check{V}_{h+1}^m(s_{h+1}^m) - \bar{P}_h^m \check{V}_{h+1}^m) + \sum_{m=1}^{M'} \sum_{h=1}^{H_m} (b_h^m - 8\eta_m)
$$

$$
\lesssim \sum_{m=1}^{M'} \sum_{h=1}^{H_m} \left( \sqrt{\frac{\overline{c}_h^m}{\mathbf{M}_h^m}} + \sqrt{\frac{\mathbb{V}(\bar{P}_h^m, \check{V}_{h+1}^m)}{\mathbf{N}_h^m}} \right) = \tilde{\mathcal{O}}\left( \sqrt{B_\star S A L_c K} + B_\star \sqrt{S A L_P K} \right).
$$

where $\{\mathcal{I}_i^c\}_{i=1}^{L_c}$ (or $\{\mathcal{I}_i^P\}_{i=1}^{L_P}$) is a partition of $K$ episodes such that $\mathbf{M}$ (or $\mathbf{N}$) is reseted in the last interval of each $\mathcal{I}_i^c$ (or $\mathcal{I}_i^P$) for $i < L_c$ (or $i < L_P$) and the last interval of $\mathcal{I}_{L_c}^c$ (or $\mathcal{I}_{L_P}^P$) is $K$, and in the second last step we apply the definition of $\chi_m^c$ (Lemma 24) and $\chi_m^P$ (Lemma 25). Note that the regret of non-stationarity along the learner's trajectory is cancelled out by the negative correction term $-8\eta_m$. Now it suffices to bound $L_c$ and $L_P$. It can be shown that the reset rules of the non-stationarity tests guarantee that

$$
L_c = \tilde{\mathcal{O}}\left( K/W_c + B_\star K/W_P \right), \quad L_P = \tilde{\mathcal{O}}\left( K/W_P + K/(B_\star W_c) \right).
$$

Details are deferred to Lemma 26. Putting everything together completes the proof.

Next, we present three lemmas related to the optimism and magnitude (**Test 3**) of estimated value function.

**Lemma 21.** *With probability at least $1 - 2\delta$, for all $m \leq K$, $\check{Q}_h^m(s,a) \leq \check{Q}_h^{\star,m}(s,a) + (\Delta_{c,m} + B\Delta_{P,m})(H - h + 1)$.*

*Proof.* We prove this by induction on $h$. The base case of $h = H + 1$ is clearly true. For $h \leq H$, by **Test 3** and the induction step, we have $\check{V}_{h+1}^m(s) \leq \min\{B/2, \check{V}_{h+1}^{\star,m}(s) + (\Delta_{c,m} + B\Delta_{P,m})(H-h)\} \leq \check{V}_{h+1}^{\star,m}(s) + x_{h+1}^m \leq B$ where $x_h^m = \min\{B/2, (\Delta_{c,m} + B\Delta_{P,m})(H - h + 1)\}$ and $\check{V}_h^{\star,m}(s) \leq \check{V}_h^{\pi^\star,m}(s) \leq \frac{B}{4} + 8H\eta_m \leq \frac{B}{3}$. Thus, with probability at least $1 - 2\delta$,

$$
\check{c}^m(s,a) + \bar{P}_{s,a}^m \check{V}_{h+1}^m - b^m(s, a, \check{V}_{h+1}^m)
$$
$$
\leq \check{c}^m(s,a) + \bar{P}_{s,a}^m(\check{V}_{h+1}^{\star,m} + x_{h+1}^m) - b^m(s, a, \check{V}_{h+1}^{\star,m} + x_{h+1}^m) \qquad \text{(Lemma 48)}
$$
$$
\overset{\text{(i)}}{\leq} \check{c}^m(s,a) + \widetilde{P}_{s,a}^m(\check{V}_{h+1}^{\star,m} + x_{h+1}^m) \qquad \text{(Lemma 8)}
$$
$$
\leq c^m(s,a) + 8\eta_m + \Delta_{c,m} + P_{s,a}^m(\check{V}_{h+1}^{\star,m} + x_{h+1}^m) + B\Delta_{P,m} \qquad \text{(Lemma 5)}
$$
$$
\leq \check{Q}_h^{\star,m}(s,a) + (\Delta_{c,m} + B\Delta_{P,m})(H - h + 1).
$$

Note that in (i) we use the fact that $|\{\check{V}_h^{\star,m} + x_h^m\}_{m,h}| \leq (HK+1)^6$ since $|\{(c^m, P^m)\}_m| \leq K$, $|\{\rho_m^c\}_m| \leq K$, $|\{\rho_m^P\}_m| \leq K$, $|\{\Delta_{c,m}\}_m| \leq K+1$, and $|\{\Delta_{P,m}\}_m| \leq K+1$ ($\Delta_{c,m} = \Delta_{P,m} = 0$ when $m$ is not the first interval of some episode). $\square$

**Lemma 22.** *With probability at least $1 - 2\delta$, for all $m \leq K$, $\check{Q}_h^m(s,a) \leq \check{Q}_h^{\pi^\star,m}(s,a) + (\Delta_{c,m} + B\Delta_{P,m})T_h^{\pi^\star,m}(s,a)$.*

*Proof.* We prove this by induction on $h$. The base case of $h = H + 1$ is clearly true. For $h \leq H$, by **Test 3** and the induction step, we have $\check{V}_{h+1}^m(s) \leq \min\{B/2, \check{V}_{h+1}^{\pi^\star,m}(s) + (\Delta_{c,m} + B\Delta_{P,m})T_{h+1}^{\pi^\star,m}(s)\} \leq \check{V}_{h+1}^{\pi^\star,m}(s) + x_{h+1}^m(s) \leq B$ where $x_h^m(s) = \min\{B/2, (\Delta_{c,m} + B\Delta_{P,m})T_h^{\pi^\star,m}(s)\}$ and $\check{V}_h^{\pi^\star,m}(s) \leq \frac{B}{4} + 8\eta_m T_h^{\pi^\star,m}(s) \leq \frac{B}{4} + 8H\eta_m \leq \frac{B}{3}$. Thus, with probability at least $1 - 2\delta$,

$$
\check{c}^m(s,a) + \bar{P}_{s,a}^m \check{V}_{h+1}^m - b^m(s, a, \check{V}_{h+1}^m)
$$
$$
\leq \check{c}^m(s,a) + \bar{P}_{s,a}^m(\check{V}_{h+1}^{\pi^\star,m} + x_{h+1}^m) - b^m(s, a, \check{V}_{h+1}^{\pi^\star,m} + x_{h+1}^m) \qquad \text{(Lemma 48)}
$$
$$
\overset{\text{(i)}}{\leq} \check{c}^m(s,a) + \widetilde{P}_{s,a}^m(\check{V}_{h+1}^{\pi^\star,m} + x_{h+1}^m) \qquad \text{(Lemma 8)}
$$
$$
\leq c^m(s,a) + 8\eta_m + \Delta_{c,m} + P_{s,a}^m(\check{V}_{h+1}^{\pi^\star,m} + x_{h+1}^m) + B\Delta_{P,m} \qquad \text{(Lemma 5)}
$$
$$
\leq \check{Q}_h^{\pi^\star,m}(s,a) + (\Delta_{c,m} + B\Delta_{P,m})T_h^{\pi^\star,m}(s,a).
$$

Note that in (i) we use the fact that $|\{\check{V}_h^{\pi^\star,m} + x_h^m\}_{m,h}| \leq (HK+1)^6$ since $|\{V_h^{\pi^\star,m}\}_{m,h}| \leq HK+1$, $|\{\rho_m^c\}_m| \leq K$, $|\{\rho_m^P\}_m| \leq K$, $|\{\Delta_{c,m}\}_m| \leq K+1$, $|\{\Delta_{P,m}\}_m| \leq K+1$ ($\Delta_{c,m} = \Delta_{P,m} = 0$ when $m$ is not the first interval of some episode), and $|\{T_h^{\pi^\star,m}\}_{m,h}| \leq HK + 1$. $\square$

**Lemma 23.** *With probability at least $1 - 2\delta$, for all $m \leq K$, if $\Delta_{c,m} \leq \rho_m^c$ and $\Delta_{P,m} \leq \rho_m^P$, then $\check{Q}_h^m(s,a) \leq \check{Q}_h^{\pi^\star,m}(s,a) + \eta_m T_h^{\pi^\star,m}(s,a) \leq B/2$. Moreover, if **Test 3** fails in interval $m$, then $\Delta_{c,[i_m^c, m+1]} > g^c(\nu_m^c + 1)$ or $\Delta_{P,[i_m^P, m+1]} > g^P(\nu_m^P + 1)$.*

*Proof.* First note that $\check{Q}_h^{\pi^*,m}(s,a) \le \frac{B}{4} + 8\eta_m T_h^{\pi^*,m}(s,a) \le \frac{B}{4} + 8H\eta_m \le \frac{B}{3}$. We prove the first statement by induction on $h$. The base case of $h = H+1$ is clearly true. For $h \le H$, note that:

$$\check{c}^m(s,a) + \bar{P}_{s,a}^m \check{V}_{h+1}^m - b^m(s,a,\check{V}_{h+1}^m)$$

$$\le \check{c}^m(s,a) + \bar{P}_{s,a}^m(\check{V}_{h+1}^{\pi^*,m} + \eta_m T_{h+1}^{\pi^*,m}) - b^m(s,a,\check{V}_{h+1}^{\pi^*,m} + \eta_m T_{h+1}^{\pi^*,m})$$
$$\text{(induction step and Lemma 48)}$$

$$\overset{(i)}{\le} \check{c}^m(s,a) + \widetilde{P}_{s,a}^m(\check{V}_{h+1}^{\pi^*,m} + \eta_m T_{h+1}^{\pi^*,m}) \qquad\qquad\qquad\qquad \text{(Lemma 8)}$$

$$\le c^m(s,a) + 8\eta_m + \rho_m^c + P_{s,a}^m(\check{V}_{h+1}^{\pi^*,m} + \eta_m T_{h+1}^{\pi^*,m}) + \rho_m^P(B/3 + H\eta_m)$$
$$\text{(Lemma 5, } \Delta_{c,m} \le \rho_m^c \text{, and } \Delta_{P,m} \le \rho_m^P\text{)}$$

$$\le \check{Q}_h^{\pi^*,m}(s,a) + \eta_m T_h^{\pi^*,m}(s,a). \qquad\qquad\qquad\qquad (H\eta_m \le B/12)$$

Note that in (i) we use the fact that $|\{\check{V}_h^{\pi^*,m} + \eta_m T_h^{\pi^*,m}\}_{m,h}| \le (HK+1)^6$ since $|\{V_h^{\pi^*,m}\}_{m,h}| \le HK + 1$, $|\{\rho_m^c\}_m| \le K$, $\{\rho_m^P\}_m \le K$, and $|\{T_h^{\pi^*,m}\}_{m,h}| \le HK + 1$. The second statement is simply by the contraposition of the first statement. $\qquad\square$

The next two lemmas are about **Test 1** and **Test 2**.

**Lemma 24.** *With probability at least* $1 - 4\delta$*, for any* $M' \le K$*, if* $\Delta_{c,M'} \le \rho_{M'}^c$*, then*

$$\sum_{m=i_{M'}^c}^{M'} \sum_{h=1}^{H_m} (c_h^m - \widehat{c}_h^m) \le \tilde{\mathcal{O}}\left( \sqrt{C_{[i_{M'}^c, M']}} + \sum_{m=i_{M'}^c}^{M'} \sum_{h=1}^{H_m} \left( \sqrt{\frac{\overline{c}_h^m}{\mathbf{M}_h^m}} + \frac{1}{\mathbf{M}_h^m} \right) \right) + \sum_{m=i_{M'}^c}^{M'} \sum_{h=1}^{H_m} \rho_m^c \triangleq \chi_{M'}^c.$$

*Moreover, if* **Test 1** *fails in interval* $M'$*, then* $\Delta_{c,M'} > \rho_{M'}^c$*.*

*Proof.* Note that for any given $M' \le M$, without loss of generality, we can offset the intervals and assume $i_{M'}^c = 1$. Then with probability at least $1 - 4\delta$, for any $M' \le K$, assuming $i_{M'}^c = 1$ we have

$$\sum_{m=1}^{M'} \sum_{h=1}^{H_m} (c_h^m - \widehat{c}_h^m) = \sum_{m=1}^{M'} \sum_{h=1}^{H_m} (c_h^m - c^m(s_h^m, a_h^m)) + \sum_{m=1}^{M'} \sum_{h=1}^{H_m} (c^m(s_h^m, a_h^m) - \widehat{c}_h^m)$$

$$\le \tilde{\mathcal{O}}\left( \sqrt{C_{M'}} \right) + \sum_{m=1}^{M'} \sum_{h=1}^{H_m} (c^m(s_h^m, a_h^m) - \widehat{c}_h^m) \quad \text{(Lemma 49 and Lemma 50)}$$

$$\le \tilde{\mathcal{O}}\left( \sqrt{C_{M'}} + \sum_{m=1}^{M'} \sum_{h=1}^{H_m} \left( \sqrt{\frac{\overline{c}_h^m}{\mathbf{M}_h^m}} + \frac{1}{\mathbf{M}_h^m} \right) \right) + \sum_{m=1}^{M'} \sum_{h=1}^{H_m} \rho_m^c.$$
$$\text{(Lemma 3, and } \Delta_{c,m} \le \Delta_{c,M'} \le \rho_{M'}^c \le \rho_m^c\text{)}$$

The first statement is then proved by noting $i_{M'}^c = 1$. The second statement is simply by the contraposition of the first statement. $\qquad\square$

**Lemma 25.** *With probability at least* $1 - 16\delta$*, for any* $M' \le K$*, if* $\Delta_{c,[i_{M'}^P, M']} \le \bar{\rho}_{M'}^c \triangleq \min\{\frac{B_\star^{1.5} c_1}{\sqrt{\nu_{M'}^P}}, \frac{1}{2^8 H}\}$ *and* $\Delta_{P,M'} \le \rho_{M'}^P$*, then*

$$\sum_{m=i_{M'}^P}^{M'} \sum_{h=1}^{H_m} \left( \check{V}_{h+1}^m(s_{h+1}^m) - \bar{P}_h^m \check{V}_{h+1}^m \right) \le \tilde{\mathcal{O}}\left( \sqrt{\sum_{m=i_{M'}^P}^{M'} \sum_{h=1}^{H_m} \mathbb{V}(\bar{P}_h^m, \check{V}_{h+1}^m)} + \sum_{m=i_{M'}^P}^{M'} \sum_{h=1}^{H_m} \sqrt{\frac{\mathbb{V}(\bar{P}_h^m, \check{V}_{h+1}^m)}{\mathbf{N}_h^m}} \right)$$

$$+ \tilde{\mathcal{O}}\left( \sqrt{SA(B_\star + L_{c,[i_{M'}^P, M']}) C_{[i_{M'}^P, M']}} + \sqrt{B_\star SA\nu_{M'}^P} + B_\star^{2.5} S^2 AHL_{c,[i_{M'}^P, M']} \right) + 4 \sum_{m=i_{M'}^P}^{M'} \sum_{h=1}^{H_m} \eta_m \triangleq \chi_{M'}^P.$$

*Moreover, if* **Test 2** *fails in interval* $M'$*, then* $\Delta_{c,[i_{M'}^P, M']} > \bar{\rho}_{M'}^c$ *or* $\Delta_{P,M'} > \rho_{M'}^P$*.*

*Proof.* For any $M' \leq K$, without loss of generality, we can offset the intervals and assume $i_{M'}^P = 1$. Moreover, for any $m \leq M'$, we have $\Delta_{P,m} \leq \Delta_{P,M'} \leq \rho_{M'}^P \leq \rho_m^P$. Thus, with probability at least $1 - 2\delta$,

$$
\sum_{m=1}^{M'} \sum_{h=1}^{H_m} \left( \check{V}_{h+1}^m(s_{h+1}^m) - \bar{P}_h^m \check{V}_{h+1}^m \right)
$$

$$
\leq \sum_{m=1}^{M'} \sum_{h=1}^{H_m} (\check{V}_{h+1}^m(s_{h+1}^m) - P_h^m \check{V}_{h+1}^m) + \sum_{m=1}^{M'} \sum_{h=1}^{H_m} (\widetilde{P}_h^m - \bar{P}_h^m)\check{V}_{h+1}^m + \sum_{m=1}^{M'} \sum_{h=1}^{H_m} B(\rho_m^P + \mathbf{n}_h^m)
$$

$$
(P_h^m \check{V}_{h+1}^m \leq \widetilde{P}_h^m \check{V}_{h+1}^m + B(\Delta_{P,m} + \mathbf{n}_h^m) \text{ and } \Delta_{P,m} \leq \rho_m^P)
$$

$$
\leq \tilde{\mathcal{O}}\left( \sqrt{\sum_{m=1}^{M'} \sum_{h=1}^{H_m} \mathbb{V}(P_h^m, \check{V}_{h+1}^m) + B_\star SA} \right) + \sum_{m=1}^{M'} \sum_{h=1}^{H_m} (\widetilde{P}_h^m - \bar{P}_h^m)\check{V}_{h+1}^m + \sum_{m=1}^{M'} \sum_{h=1}^{H_m} B\rho_m^P
$$

$$
(\text{Lemma 49 and } \textstyle\sum_{m=1}^{M'} \sum_{h=1}^{H_m} \mathbf{n}_h^m \leq \sum_{m=1}^{M'} \sum_{h=1}^{H_m} \frac{1}{\mathbf{N}_h^m} \leq SA \text{ by } L_{P,M'} = 1)
$$

$$
\leq \tilde{\mathcal{O}}\left( \sqrt{\sum_{m=1}^{M'} \sum_{h=1}^{H_m} \mathbb{V}(\bar{P}_h^m, \check{V}_{h+1}^m) + B_\star SA} \right) + \sum_{m=1}^{M'} \sum_{h=1}^{H_m} (\widetilde{P}_h^m - \bar{P}_h^m)\check{V}_{h+1}^m + \sum_{m=1}^{M'} \sum_{h=1}^{H_m} 2B\rho_m^P,
$$

where the last inequality is by

$$
\mathbb{V}(P_h^m, \check{V}_{h+1}^m) \leq P_h^m (\check{V}_{h+1}^m - \bar{P}_h^m \check{V}_{h+1}^m)^2 \leq \widetilde{P}_h^m (\check{V}_{h+1}^m - \bar{P}_h^m \check{V}_{h+1}^m)^2 + B^2(\Delta_{P,m} + \mathbf{n}_h^m)
$$

$$
(\textstyle\frac{\sum_i p_i x_i}{\sum_i p_i} = \text{argmin}_z \sum_i p_i(x_i - z)^2)
$$

$$
\leq 2\mathbb{V}(\bar{P}_h^m, \check{V}_{h+1}^m) + \tilde{\mathcal{O}}\left( \frac{SB^2}{\mathbf{N}_h^m} \right) + B^2 \rho_m^P,
$$

$$
(\widetilde{P}_h^m(s') \leq 2\bar{P}_h^m(s') + \tfrac{1}{\mathbf{N}_h^m} \text{ by Lemma 50, } \mathbf{n}_h^m \leq \tfrac{1}{\mathbf{N}_h^m}, \text{ and } \Delta_{P,m} \leq \rho_m^P)
$$

Lemma 11, $L_{P,M'} = 1$, and AM-GM inequality. Now note that with probability at least $1 - 3\delta$,

$$
\sum_{m=1}^{M'} \sum_{h=1}^{H_m} (\widetilde{P}_h^m - \bar{P}_h^m)\check{V}_{h+1}^m = \sum_{m=1}^{M'} \sum_{h=1}^{H_m} \left( (\widetilde{P}_h^m - \bar{P}_h^m)\check{V}_{h+1}^{\star,m} + (\widetilde{P}_h^m - \bar{P}_h^m)(\check{V}_{h+1}^m - \check{V}_{h+1}^{\star,m}) \right)
$$

$$
\leq \tilde{\mathcal{O}}\left( \sum_{m=1}^{M'} \sum_{h=1}^{H_m} \left( \sqrt{\frac{\mathbb{V}(\bar{P}_h^m, \check{V}_{h+1}^{\star,m})}{\mathbf{N}_h^m}} + \frac{SB_\star}{\mathbf{N}_h^m} \right) + \sqrt{S^2 A \sum_{m=1}^{M'} \sum_{h=1}^{H_m} \mathbb{V}(P_h^m, \check{V}_{h+1}^m - \check{V}_{h+1}^{\star,m})} \right) + \sum_{m=1}^{M'} \sum_{h=1}^{H_m} \frac{B\rho_m^P}{32}
$$

$$
(\text{Lemma 8, Lemma 6, Cauchy-Schwarz inequality, Lemma 11, and } \Delta_{P,m} \leq \rho_m^P)
$$

$$
\leq \tilde{\mathcal{O}}\left( \sum_{m=1}^{M'} \sum_{h=1}^{H_m} \left( \sqrt{\frac{\mathbb{V}(\bar{P}_h^m, \check{V}_{h+1}^m)}{\mathbf{N}_h^m}} + \frac{SB_\star}{\mathbf{N}_h^m} \right) + \sqrt{S^2 A \sum_{m=1}^{M'} \sum_{h=1}^{H_m} \mathbb{V}(P_h^m, \check{V}_{h+1}^m - \check{V}_{h+1}^{\star,m})} \right) + \sum_{m=1}^{M'} \sum_{h=1}^{H_m} \frac{B\rho_m^P}{16},
$$

where in the last step we apply

$$
\sum_{m=1}^{M'} \sum_{h=1}^{H_m} \sqrt{\frac{\mathbb{V}(\bar{P}_h^m, \check{V}_{h+1}^{\star,m})}{\mathbf{N}_h^m}} \leq \sum_{m=1}^{M'} \sum_{h=1}^{H_m} \left( \sqrt{\frac{\mathbb{V}(\bar{P}_h^m, \check{V}_{h+1}^m)}{\mathbf{N}_h^m}} + \sqrt{\frac{\mathbb{V}(\bar{P}_h^m, \check{V}_{h+1}^m - \check{V}_{h+1}^{\star,m})}{\mathbf{N}_h^m}} \right)
$$

by $\sqrt{\mathrm{VAR}[X+Y]} \leq \sqrt{\mathrm{VAR}[X]} + \sqrt{\mathrm{VAR}[Y]}$ [Cohen et al., 2021, Lemma E.3] and

$$\sum_{m=1}^{M'} \sum_{h=1}^{H_m} \sqrt{\frac{\mathbb{V}(\bar{P}_h^m, \check{V}_{h+1}^m - \check{V}_{h+1}^{\star,m})}{\mathbf{N}_h^m}} \leq \sum_{m=1}^{M'} \sum_{h=1}^{H_m} \sqrt{\frac{\bar{P}_h^m((\check{V}_{h+1}^m - \check{V}_{h+1}^{\star,m}) - P_h^m(\check{V}_{h+1}^m - \check{V}_{h+1}^{\star,m}))^2}{\mathbf{N}_h^m}}$$

$$(\tfrac{\sum_i p_i x_i}{\sum_i p_i} = \operatorname{argmin}_z \sum_i p_i (x_i - z)^2)$$

$$\leq \sum_{m=1}^{M'} \sum_{h=1}^{H_m} \sqrt{\frac{2\widetilde{P}_h^m((\check{V}_{h+1}^m - \check{V}_{h+1}^{\star,m}) - P_h^m(\check{V}_{h+1}^m - \check{V}_{h+1}^{\star,m}))^2}{\mathbf{N}_h^m}} + \tilde{\mathcal{O}}\left(\sum_{m=1}^{M'} \sum_{h=1}^{H_m} \frac{B\sqrt{S}}{\mathbf{N}_h^m}\right)$$

$$(\bar{P}_h^m(s') \leq 2\widetilde{P}_h^m(s') + \tilde{\mathcal{O}}\left(\tfrac{1}{\mathbf{N}_h^m}\right) \text{ by Lemma 50})$$

$$\leq \sum_{m=1}^{M'} \sum_{h=1}^{H_m} \sqrt{\frac{2\mathbb{V}(P_h^m, \check{V}_{h+1}^m - \check{V}_{h+1}^{\star,m})}{\mathbf{N}_h^m}} + \tilde{\mathcal{O}}\left(\sum_{m=1}^{M'} \sum_{h=1}^{H_m} \frac{B\sqrt{S}}{\mathbf{N}_h^m} + \sum_{m=1}^{M'} \sum_{h=1}^{H_m} B\sqrt{\frac{\Delta_{P,m}}{\mathbf{N}_h^m}}\right)$$

$$\leq \tilde{\mathcal{O}}\left(\sqrt{SA \sum_{m=1}^{M'} \sum_{h=1}^{H_m} \mathbb{V}(P_h^m, \check{V}_{h+1}^m - \check{V}_{h+1}^{\star,m})} + \sum_{m=1}^{M'} \sum_{h=1}^{H_m} \frac{B\sqrt{S}}{\mathbf{N}_h^m}\right) + \sum_{m=1}^{M'} \sum_{h=1}^{H_m} \frac{B\rho_m^P}{32}.$$

(Cauchy-Schwarz inequality, Lemma 11, $L_{P,M'} = 1$, AM-GM inequality, and $\Delta_{P,m} \leq \rho_m^P$)

Now by Lemma 28, $L_{P,M'} = 1$, and AM-GM inequality, we have with probability $1 - 10\delta$,

$$\sqrt{S^2 A \sum_{m=1}^{M'} \sum_{h=1}^{H_m} \mathbb{V}(P_h^m, \check{V}_{h+1}^m - \check{V}_{h+1}^{\star,m})} \leq \tilde{\mathcal{O}}\left(\sqrt{SAL_{c,M'} C_{M'}} + \sqrt{B_\star SA(C_{M'} + M')}\right)$$

$$+ \tilde{\mathcal{O}}\left(B_\star S^2 A + B_\star S^{1.5} A L_{c,M'} + \sqrt{B_\star S^2 A \sum_{m=1}^{M'} (\Delta_{c,m} + B_\star \Delta_{P,m}) H}\right).$$

Moreover, by $i_m^c \geq i_m^P$ and $\nu_m^c \leq \nu_m^P$ due to the reset rules, we have $\Delta_{c,m} \leq \Delta_{c,[i_{M'}^P, m]} \leq \Delta_{c,[i_{M'}^P, M']} \leq \bar{\rho}_{M'}^c \leq \bar{\rho}_m^c \leq B_\star^{1.5} \min\{\frac{c_1}{\sqrt{\nu_m^P}}, \frac{1}{2^8 H}\} \leq B_\star^{1.5} \min\{\frac{c_1}{\sqrt{\nu_m^c}}, \frac{1}{2^8 H}\} \leq B_\star^{1.5} \rho_m^c$. Therefore, by $\Delta_{P,m} \leq \rho_m^P$ and AM-GM inequality,

$$\sqrt{B_\star S^2 A \sum_{m=1}^{M'} \sum_{h=1}^{H_m} (\Delta_{c,m} + B_\star \Delta_{P,m})} \leq \sqrt{B_\star^{2.5} S^2 AH \sum_{m=1}^{M'} (\rho_m^c + B_\star \rho_m^P)} \leq B_\star^{2.5} S^2 AH + \sum_{m=1}^{M'} \eta_m.$$

Plugging these back, and by Lemma 11, $L_{P,M'} = 1$, we obtain

$$\sum_{m=1}^{M'} \sum_{h=1}^{H_m} (\widetilde{P}_h^m - \bar{P}_h^m) \check{V}_{h+1}^m \leq \tilde{\mathcal{O}}\left(\sum_{m=1}^{M'} \sum_{h=1}^{H_m} \left(\sqrt{\frac{\mathbb{V}(\bar{P}_h^m, \check{V}_{h+1}^m)}{\mathbf{N}_h^m}}\right) + \sqrt{B_\star SA(C_{M'} + M')}\right)$$

$$+ \tilde{\mathcal{O}}\left(\sqrt{SAL_{c,M'} C_{M'}} + B_\star S^{1.5} A L_{c,M'} + B_\star^{2.5} S^2 AH\right) + 2 \sum_{m=1}^{M'} \sum_{h=1}^{H_m} \eta_m.$$

Plugging this back and noting $i_{M'}^P = 1$ completes the proof of the first statement. The second statement is simply by the contraposition of the first statement. $\qquad\square$

## E.1 Proof of Theorem 4

*Proof.* By $s_1^m = s_{\text{init}}$, we decompose the regret as follows, with probability at least $1 - 2\delta$,

$$\mathring{R}_K = \sum_{m=1}^{K} \left( \sum_{h=1}^{H_m} c_h^m + c_{H_m+1}^m - V_1^{\pi^\star,m}(s_1^m) \right)$$

$$= \sum_{m=1}^{K} \left( \sum_{h=1}^{H_m} c_h^m + c_{H_m+1}^m - \check{V}_1^m(s_1^m) \right) + \sum_{m=1}^{K} \left( \check{V}_1^m(s_1^m) - V_1^{\pi^\star,m}(s_1^m) \right) + 8T_\star \sum_{m=1}^{K} \eta_m$$

$$\leq \sum_{m=1}^{K} \left( \sum_{h=1}^{H_m} c_h^m + c_{H_m+1}^m - \check{V}_1^m(s_1^m) \right) + \sum_{m=1}^{K} (\Delta_{c,m} + B\Delta_{P,m})T_\star + 8T_\star \sum_{m=1}^{K} \eta_m$$

$$\text{(Lemma 22)}$$

We first bound the first and the third term above separately. For the third term, we have:

$$T_\star \sum_{m=1}^{K} \eta_m \leq T_\star \sum_{m=1}^{K} \left( \frac{c_1}{\sqrt{\nu_m^c}} + \frac{Bc_2}{\sqrt{\nu_m^P}} \right) = \tilde{\mathcal{O}} \left( T_\star \left( c_1 \sum_{i=1}^{L_c} \sqrt{M_i^c} + B_\star c_2 \sum_{i=1}^{L_P} \sqrt{M_i^P} \right) \right)$$

$$\left( \sum_{i=1}^{j} \frac{1}{\sqrt{i}} = \mathcal{O}(\sqrt{j}) \right)$$

$$= \tilde{\mathcal{O}} \left( T_\star (c_1 \sqrt{L_c K} + B_\star c_2 \sqrt{L_P K}) \right) = \tilde{\mathcal{O}} \left( \sqrt{B_\star SAL_c K} + B_\star \sqrt{SAL_P K} \right),$$

where $M_i^c$ (or $M_i^P$) is the number of intervals between the $i$-th and $(i+1)$-th reset of cost (or transition) estimation, and the second last step is by Cauchy-Schwarz inequality. For the first term, define $\{\mathcal{I}_i^c\}_{i=1}^{L_c}$ (or $\{\mathcal{I}_i^P\}_{i=1}^{L_P}$) as a partition of $K$ episodes such that $\mathbf{M}$ (or $\mathbf{N}$) is reset in the last interval of each $\mathcal{I}_i^c$ (or $\mathcal{I}_i^P$) for $i < L_c$ (or $i < L_P$) and the last interval of $\mathcal{I}_{L_c}^c$ (or $\mathcal{I}_{L_P}^P$) is $K$. Also let $L = L_c + L_P$. Then with probability at least $1 - 20\delta$,

$$\sum_{m=1}^{K} \left( \sum_{h=1}^{H_m} c_h^m + c_{H_m+1}^m - \check{V}_1^m(s_1^m) \right) \leq \sum_{m=1}^{K} \sum_{h=1}^{H_m} \left( c_h^m + \check{V}_{h+1}^m(s_{h+1}^m) - \check{V}_h^m(s_h^m) \right) + \tilde{\mathcal{O}}\left( B_\star SAL \right)$$

$$\text{(Lemma 12)}$$

$$\leq \sum_{m=1}^{K} \sum_{h=1}^{H_m} \left( c_h^m - \hat{c}_h^m + \check{V}_{h+1}^m(s_{h+1}^m) - \bar{P}_h^m \check{V}_{h+1}^m + b_h^m - 8\eta_m \right) + \tilde{\mathcal{O}}\left( B_\star SAL \right)$$

$$\text{(definition of } \check{V}_h^m(s_h^m))$$

$$= \sum_{i=1}^{L_c} \sum_{m \in \mathcal{I}_i^c} \sum_{h=1}^{H_m} (c_h^m - \hat{c}_h^m) + \sum_{i=1}^{L_P} \sum_{m \in \mathcal{I}_i^P} \sum_{h=1}^{H_m} (\check{V}_{h+1}^m(s_{h+1}^m) - \bar{P}_h^m \check{V}_{h+1}^m) + \sum_{m=1}^{K} \sum_{h=1}^{H_m} (b_h^m - 8\eta_m) + \tilde{\mathcal{O}}\left( B_\star SAL \right)$$

$$= \tilde{\mathcal{O}} \left( \sqrt{L_c C_K} + \sum_{m=1}^{K} \sum_{h=1}^{H_m} \left( \sqrt{\frac{\bar{c}_h^m}{\mathbf{M}_h^m}} + \frac{1}{\mathbf{M}_h^m} \right) + \sqrt{L_P \sum_{m=1}^{K} \sum_{h=1}^{H_m} \mathbb{V}(\bar{P}_h^m, \check{V}_{h+1}^m) + \sum_{m=1}^{K} \sum_{h=1}^{H_m} b_h^m} \right)$$

$$+ \tilde{\mathcal{O}} \left( B_\star^{2.5} S^2 AHL_c + \sqrt{B_\star SAL_P (C_K + K)} + \sqrt{SAL_c C_K} + HL_c + B_\star HL_P \right),$$

$$\text{(\textbf{Test 1} (Lemma 24), \textbf{Test 2} (Lemma 25), and Cauchy-Schwarz inequality)}$$

where $\tilde{\mathcal{O}}(HL_c + B_\star HL_P)$ is upper bound of the costs in intervals where **Test 1** fails or **Test 2** fails. By Lemma 3 and AM-GM inequality, with probability at least $1 - 3\delta$,

$$\sum_{m=1}^{K} \sum_{h=1}^{H_m} \left( \sqrt{\frac{\bar{c}_h^m}{\mathbf{M}_h^m}} + \frac{1}{\mathbf{M}_h^m} \right) = \tilde{\mathcal{O}} \left( SAHL_c + \sqrt{SAL_c C_K} \right) + \sum_{m=1}^{K} \Delta_{c,m}.$$

Following the proof of Lemma 10, we have $\sqrt{L_P \sum_{m=1}^{K} \sum_{h=1}^{H_m} \mathbb{V}(\bar{P}_h^m, \check{V}_{h+1}^m)}$ is dominated by the upper bound of $\sum_{m=1}^{M'} \sum_{h=1}^{H_m} b_h^m$. Thus with probability at least $1 - \delta$,

$$
\sqrt{L_P \sum_{m=1}^{K} \sum_{h=1}^{H_m} \mathbb{V}(\bar{P}_h^m, \check{V}_{h+1}^m)} + \sum_{m=1}^{K} \sum_{h=1}^{H_m} b_h^m
$$
$$
= \tilde{\mathcal{O}}\left( \sqrt{SAL_P \sum_{m=1}^{K} \sum_{h=1}^{H_m} \mathbb{V}(P_h^m, \check{V}_{h+1}^m) + B_\star S^{1.5} AL_P} + B_\star \sqrt{SAL_P \sum_{m=1}^{K} \sum_{h=1}^{H_m} \Delta_{P,m}} \right)
$$
$$
= \tilde{\mathcal{O}}\left( \sqrt{B_\star SAL_P (C_K + K)} + \sqrt{SAL_c C_K} + B_\star S^{1.5} AHL \right) + \sum_{m=1}^{K} (\Delta_{c,m} + B_\star \Delta_{P,m}),
$$

where in the last inequality we apply AM-GM inequality on $B_\star \sqrt{SAL_P \sum_{m=1}^{K} \sum_{h=1}^{H_m} \Delta_{P,m}}$, and note that with probability at least $1 - 11\delta$,

$$
\sqrt{SAL_P \sum_{m=1}^{K} \sum_{h=1}^{H_m} \mathbb{V}(P_h^m, \check{V}_{h+1}^m)}
$$
$$
= \tilde{\mathcal{O}}\left( \sqrt{SAL_P \sum_{m=1}^{K} \sum_{h=1}^{H_m} \mathbb{V}(P_h^m, \check{V}_{h+1}^{\star,m})} + \sqrt{SAL_P \sum_{m=1}^{K} \sum_{h=1}^{H_m} \mathbb{V}(P_h^m, \check{V}_{h+1}^m - \check{V}_{h+1}^{\star,m})} \right)
$$
$$
\text{(VAR}[X + Y] \leq 2\text{VAR}[X] + 2\text{VAR}[Y] \text{ and } \sqrt{a + b} \leq \sqrt{a} + \sqrt{b})
$$
$$
= \tilde{\mathcal{O}}\left( \sqrt{B_\star SAL_P (C_K + K)} + \sqrt{SAL_c C_K} + B_\star S^{1.5} AHL \right) + \sum_{m=1}^{M'} (\Delta_{c,m} + B_\star \Delta_{P,m}).
$$
$$
\text{(Lemma 27, Lemma 28, and AM-GM inequality)}
$$

Putting everything together, we have

$$
\mathring{R}_K = \tilde{\mathcal{O}}\left( \sqrt{SA(L_c + B_\star L_P)(C_K + B_\star K)} + B_\star^{2.5} S^2 AHL + \sum_{m=1}^{K} (\Delta_{c,m} + B_\star \Delta_{P,m}) T_\star \right).
$$

Now by $\mathring{R}_K \geq C_K - 4B_\star K$, solving a quadratic inequality (Lemma 45) w.r.t $C_K$ and plugging the bound on $C_K$ back, we obtain

$$
\mathring{R}_K = \tilde{\mathcal{O}}\left( \sqrt{B_\star SAL_c K} + B_\star \sqrt{SAL_P K} + B_\star^{2.5} S^2 AHL + \sum_{m=1}^{K} (\Delta_{c,m} + B_\star \Delta_{P,m}) T_\star \right).
$$

It suffices to bound the last term above. By the periodic resets of $\mathbf{M}$ and $\mathbf{N}$ (Line 4 and Line 5 of Algorithm 4), the number of intervals between consecutive resets of $\mathbf{M}$ and $\mathbf{N}$ are upper bounded by $W_c$ and $W_P$ respectively. Thus,

$$
\sum_{m=1}^{K} (\Delta_{c,m} + B_\star \Delta_{P,m}) T_\star \leq \sum_{m=1}^{K} (\Delta_{c,f^c(m)} + B_\star \Delta_{P,f^P(m)}) T_\star \leq (W_c \Delta_c + B_\star W_P \Delta_P) T_\star
$$
$$
= \tilde{\mathcal{O}}\left( (B_\star SAT_\star \Delta_c)^{1/3} K^{2/3} + B_\star (SAT_\star \Delta_P)^{1/3} K^{2/3} + (\Delta_c + B_\star \Delta_P) T_\star \right),
$$

where the last step is simply by the chosen values of $W_c$ and $W_P$. Plugging this back and applying Lemma 26 completes the proof. $\qquad\square$

**Lemma 26.** *With probability at least $1 - 2\delta$, Algorithm 4 with $p = 1/B_\star$ ensures*

$$
L_c = \tilde{\mathcal{O}}\left( (B_\star SA)^{-1/3} (T_\star \Delta_c)^{2/3} K^{1/3} + B_\star (SA)^{-1/3} (T_\star \Delta_P)^{2/3} K^{1/3} + H(\Delta_c + B_\star \Delta_P) \right),
$$
$$
L_P = \tilde{\mathcal{O}}\left( (B_\star SA)^{-1/3} (T_\star \Delta_c)^{2/3} K^{1/3} / B_\star + (SA)^{-1/3} (T_\star \Delta_P)^{2/3} K^{1/3} + H(\Delta_c + \Delta_P) \right).
$$

*Proof.* We consider the number of resets of $\mathbf{M}$ and $\mathbf{N}$ from each test separately. By Lemma 24 and Lemma 13, there are at most $\tilde{\mathcal{O}}((c_1^{-1}\Delta_c)^{2/3}K^{1/3} + H\Delta_c)$ resets of $\mathbf{M}$ triggered by **Test 1**. By Lemma 25 and Lemma 13, there are at most $\tilde{\mathcal{O}}(((B_\star^{-1.5}c_1^{-1}\Delta_c)^{2/3} + (c_2^{-1}\Delta_P)^{2/3})K^{1/3} + H(\Delta_c + \Delta_P))$ resets of $\mathbf{M}$ and $\mathbf{N}$ triggered by **Test 2**.

Next, we consider **Test 3**. Define $\mathbb{I}_m^c = \mathbb{I}\{\Delta_{c,[i_m^c,m+1]} > g^c(\nu_m^c + 1)\}$ and $\mathbb{I}_m^P = \mathbb{I}\{\Delta_{P,[i_m^P,m+1]} > g^P(\nu_m^P + 1)\}$. Note that whenever **Test 3** fails in interval $m$, we have $\mathbb{I}_m^c = 1$ or $\mathbb{I}_m^P = 1$ by Lemma 23. We partition $K$ intervals into segments $\mathcal{I}_1, \ldots, \mathcal{I}_{N_c}$, such that in the last interval of each $\mathcal{I}_i$ with $i < N_c$ denoted by $m$, **Test 3** fails and $\mathbb{I}_m^c = 1$. Since $\nu^c$ is reset whenever **Test 3** fails, we have $\Delta_{\mathcal{I}_i \cup \{m+1\}} \geq \Delta_{[i_m^c,m+1]} > g^c(\nu_m^c + 1) \geq g^c(|\mathcal{I}_i| + 1)$. By Lemma 13, we obtain $N_c = \tilde{\mathcal{O}}((c_1^{-1}\Delta_c)^{2/3}K^{1/3} + H\Delta_c)$.

Now define $\mathbb{A}_m$ as the indicator that **Test 3** fails in interval $m$ and $\mathbb{I}_m^P = 1$. Also define $\mathbb{A}_m'$ as the indicator that **Test 3** fails and $\mathbf{N}$ is reset in interval $m$, and $\mathbb{I}_m^P = 1$. We then partition $K$ intervals into segments $\mathcal{I}_1', \ldots, \mathcal{I}_{N_P}'$, such that in the last interval of each $\mathcal{I}_i'$ with $i < N_P$ denoted by $m$, $\mathbb{A}_m' = 1$. Since $\nu^P$ is reset in interval $m$ when $\mathbb{A}_m' = 1$, we have $\Delta_{\mathcal{I}_i' \cup \{m+1\}} \geq \Delta_{[i_m^P,m+1]} > g^P(\nu_m^P + 1) \geq g^P(|\mathcal{I}_i'| + 1)$. By Lemma 13, we have $N_P = \tilde{\mathcal{O}}((c_2^{-1}\Delta_P)^{2/3}K^{1/3} + H\Delta_P)$. Moreover, by Lemma 50 and the reset rule of **Test 3**, we have $p\sum_m \mathbb{A}_m = \tilde{\mathcal{O}}(\sum_m \mathbb{A}_m')$ with probability at least $1 - \delta$, which gives $\sum_m \mathbb{A}_m = \tilde{\mathcal{O}}(N_P/p)$.

Since $\mathbb{I}_m^c = 1$ or $\mathbb{I}_m^P = 1$ when **Test 3** fails in interval $m$, the total number of times that **Test 3** fails $N_3 \leq N_c + \sum_m \mathbb{A}_m = \tilde{\mathcal{O}}((c_1^{-1}\Delta_c)^{2/3}K^{1/3} + B_\star(c_2^{-1}\Delta_P)^{2/3}K^{1/3} + H(\Delta_c + B_\star\Delta_P))$. Now by the reset rule of **Test 3**, the number of times $\mathbf{M}$ is reset due to **Test 3** is upper bounded by $N_3$, and the number of times $\mathbf{N}$ is reset due to **Test 3** is upper bounded by $\tilde{\mathcal{O}}(pN_3)$ with probability at least $1 - \delta$ by Lemma 50. Finally, by Line 4 and Line 5 of Algorithm 4, there are at most $\frac{K}{W_c}$ resets of $\mathbf{M}$ and $\frac{K}{W_P}$ resets of $\mathbf{N}$ respectively due to periodic restarts. Putting all cases together, we have

$$L_c = \tilde{\mathcal{O}}\left((c_1^{-1}\Delta_c)^{2/3}K^{1/3} + B_\star(c_2^{-1}\Delta_P)^{2/3})K^{1/3} + H(\Delta_c + B_\star\Delta_P) + K/W_c\right)$$
$$= \tilde{\mathcal{O}}\left((B_\star SA)^{-1/3}(T_\star\Delta_c)^{2/3}K^{1/3} + B_\star(SA)^{-1/3}(T_\star\Delta_P)^{2/3}K^{1/3} + H(\Delta_c + B_\star\Delta_P)\right),$$

and

$$L_P = \tilde{\mathcal{O}}\left(\frac{1}{B_\star}(c_1^{-1}\Delta_c)^{2/3}K^{1/3} + (c_2^{-1}\Delta_P)^{2/3}K^{1/3} + H(\Delta_c + \Delta_P) + K/W_P\right)$$
$$= \tilde{\mathcal{O}}\left(\frac{(B_\star SA)^{-1/3}(T_\star\Delta_c)^{2/3}K^{1/3}}{B_\star} + (SA)^{-1/3}(T_\star\Delta_P)^{2/3}K^{1/3} + H(\Delta_c + \Delta_P)\right).$$

This completes the proof. □

### E.2 Auxiliary Lemmas

**Lemma 27.** *With probability at least $1 - \delta$, for any $M' \leq K$, $\sum_{m=1}^{M'}\sum_{h=1}^{H_m}\mathbb{V}(P_h^m, \check{V}_{h+1}^{\star,m}) = \tilde{\mathcal{O}}\left(B_\star C_{M'} + B_\star M' + B_\star^2\right).$*

*Proof.* Applying Lemma 9 with $\left\|\check{V}_h^{\star,m}\right\|_\infty \leq B$, with probability at least $1 - \delta$,

$$\sum_{m=1}^{M'}\sum_{h=1}^{H_m}\mathbb{V}(P_h^m, \check{V}_{h+1}^{\star,m})$$
$$= \tilde{\mathcal{O}}\left(\sum_{m=1}^{M'}\check{V}_{H_m+1}^{\star,m}(s_{H_m+1}^m)^2 + \sum_{m=1}^{M'}\sum_{h=1}^{H_m}B_\star(\check{V}_h^{\star,m}(s_h^m) - P_h^m\check{V}_{h+1}^{\star,m})_+ + B_\star^2\right)$$
$$= \tilde{\mathcal{O}}\left(B_\star C_{M'} + B_\star M' + B_\star^2\right),$$

where in the last step we apply
$$(\check{V}_h^{\star,m}(s_h^m) - P_h^m\check{V}_{h+1}^{\star,m})_+ \leq (\check{Q}_h^{\star,m}(s_h^m,a_h^m) - P_h^m\check{V}_{h+1}^{\star,m})_+ \leq c^m(s_h^m,a_h^m) + 8\eta_m$$
$$\leq c^m(s_h^m,a_h^m) + 1/H,$$

and also Lemma 50. □

**Lemma 28.** *With probability at least* $1 - 10\delta$, *for any* $M' \leq K$, $\sum_{m=1}^{M'} \sum_{h=1}^{H_m} \mathbb{V}(P_h^m, \check{V}_{h+1}^{\star,m} - \check{V}_{h+1}^m) = \tilde{\mathcal{O}}(B_\star \sqrt{SAL_{c,M'}C_{M'}} + B_\star \sqrt{B_\star SAL_{P,M'}(C_{M'} + M')} + B_\star^2 S^2 AL_{P,M'} + B_\star^2 SAL_{c,M'} + \sum_{m=1}^{M'} B_\star(\Delta_{c,m} + B_\star\Delta_{P,m})H)$.

*Proof.* Let $z_h^m = \min\{B/2, (\Delta_{c,m} + B\Delta_{P,m})H\}\mathbb{I}\{h \leq H\}$. By Lemma 21, we have $\check{V}_h^{\star,m}(s) + z_h^m \geq \check{V}_h^m(s)$. Moreover, by Lemma 12, we have

$$\sum_{m=1}^{M'} (\check{V}_{H_m+1}^{\star,m}(s_{H_m+1}^m) + z_{H_m+1}^m - \check{V}_{H_m+1}^m(s_{H_m+1}^m))^2$$

$$\leq \sum_{m=1}^{M'} (z_{H_m+1}^m)^2 \mathbb{I}\{s_{H_m+1}^m = g\} + 64B_\star^2 \sum_{m=1}^{M'} \mathbb{I}\{H_m < H, s_{H_m+1}^m \neq g\}$$

$$= \tilde{\mathcal{O}}\left(B_\star \sum_{m=1}^{M'} (\Delta_{c,m} + B_\star\Delta_{P,m})H + B_\star^2 SAL_{M'}\right).$$

and

$$(\ast) = \sum_{m=1}^{M'} B_\star \sum_{h=1}^{H_m} (\check{V}_h^{\star,m}(s_h^m) - \check{V}_h^m(s_h^m) - P_h^m\check{V}_{h+1}^{\star,m} + P_h^m\check{V}_{h+1}^m + z_h^m - z_{h+1}^m)_+$$

$$\leq \sum_{m=1}^{M'} B_\star \sum_{h=1}^{H_m} \left(c^m(s_h^m, a_h^m) + 8\eta_m + \widetilde{P}_h^m\check{V}_{h+1}^m - \check{V}_h^m(s_h^m) + B(\Delta_{P,m} + \mathbf{n}_h^m)\right)_+ + B_\star \sum_{m=1}^{M'} (z_1^m - z_{H_m+1}^m)$$

$$\quad (\check{V}_h^{\star,m}(s_h^m) \leq \check{Q}_h^{\star,m}(s_h^m, a_h^m), z_h^m \geq z_{h+1}^m, \text{ and } P_{h+1}^m\check{V}_{h+1}^m \leq \widetilde{P}_{h+1}^m\check{V}_{h+1}^m + B(\Delta_{P,m} + \mathbf{n}_h^m))$$

$$\leq \sum_{m=1}^{M'} B_\star \sum_{h=1}^{H_m} (c^m(s_h^m, a_h^m) - \hat{c}_h^m + (\widetilde{P}_h^m - \bar{P}_h^m)\check{V}_{h+1}^{\star,m} + (\widetilde{P}_h^m - \bar{P}_h^m)(\check{V}_{h+1}^m - \check{V}_{h+1}^{\star,m}) + b_h^m)_+$$

$$\quad + \tilde{\mathcal{O}}\left(\sum_{m=1}^{M'} B_\star(\Delta_{c,m} + B_\star\Delta_{P,m})H + B_\star^2 \sum_{m=1}^{M'} \sum_{h=1}^{H_m} \mathbf{n}_h^m\right). \qquad \text{(definition of } \check{V}_h^m(s_h^m))$$

Now by Lemma 3, Lemma 8, Lemma 6, and $\mathbf{n}_h^m \leq \frac{1}{\mathbf{N}_h^m}$, we continue with

$$(\ast) = \tilde{\mathcal{O}}\left(B_\star\left(\sqrt{SAL_{c,M'}C_{M'}} + SAL_{c,M'} + \sum_{m=1}^{M'} \sum_{h=1}^{H_m}\left(\sqrt{\frac{\mathbb{V}(P_h^m, \check{V}_{h+1}^{\star,m})}{\mathbf{N}_h^m}} + \sqrt{\frac{S\mathbb{V}(P_h^m, \check{V}_{h+1}^m - \check{V}_{h+1}^{\star,m})}{\mathbf{N}_h^m}}\right)\right)\right)$$

$$\quad + \tilde{\mathcal{O}}\left(\sum_{m=1}^{M'} \sum_{h=1}^{H_m} \frac{B_\star^2 S}{\mathbf{N}_h^m} + \sum_{m=1}^{M'} B_\star(\Delta_{c,m} + B_\star\Delta_{P,m})H + B_\star \sum_{m=1}^{M'} \sum_{h=1}^{H_m} b_h^m\right)$$

$$= \tilde{\mathcal{O}}\left(B_\star\sqrt{SAL_{c,M'}C_{M'}} + B_\star SAL_{c,M'} + B_\star\sqrt{SAL_{P,M'} \sum_{m=1}^{M'} \sum_{h=1}^{H_m} \mathbb{V}(P_h^m, \check{V}_{h+1}^{\star,m}) + B_\star^2 S^2 AL_{P,M'}}\right)$$

$$\quad + \tilde{\mathcal{O}}\left(B_\star\sqrt{S^2 AL_{P,M'} \sum_{m=1}^{M'} \sum_{h=1}^{H_m} \mathbb{V}(P_h^m, \check{V}_{h+1}^{\star,m} - \check{V}_{h+1}^m)} + \sum_{m=1}^{M'} B_\star(\Delta_{c,m} + B_\star\Delta_{P,m})H\right),$$

where in the last step we apply Cauchy-Schwarz inequality, Lemma 11, Lemma 10, $\text{VAR}[X + Y] \leq 2\text{VAR}[X] + 2\text{VAR}[Y]$, and AM-GM inequality. Finally, by Lemma 27, we continue with

$$(\ast) = \tilde{\mathcal{O}}\left(B_\star\sqrt{SAL_{c,M'}C_{M'}} + B_\star SAL_{c,M'} + B_\star\sqrt{B_\star SAL_{P,M'}(C_{M'} + M')} + B_\star^2 S^2 AL_{P,M'}\right)$$

$$\quad + \tilde{\mathcal{O}}\left(B_\star\sqrt{S^2 AL_{P,M'} \sum_{m=1}^{M'} \sum_{h=1}^{H_m} \mathbb{V}(P_h^m, \check{V}_{h+1}^{\star,m} - \check{V}_{h+1}^m)} + \sum_{m=1}^{M'} B_\star(\Delta_{c,m} + B_\star\Delta_{P,m})H\right).$$

Applying Lemma 9 on value functions $\{\check{V}_h^{\star,m} + z_h^m - \check{V}_h^m\}_{m,h}$ (constant offset does not change the variance) and plugging in the bounds above, we have

$$\sum_{m=1}^{M'}\sum_{h=1}^{H_m}\mathbb{V}(P_h^m, \check{V}_{h+1}^{\star,m} - \check{V}_{h+1}^m) = \sum_{m=1}^{M'}\sum_{h=1}^{H_m}\mathbb{V}(P_h^m, \check{V}_{h+1}^{\star,m} + z_h^m - \check{V}_{h+1}^m)$$

$$= \tilde{\mathcal{O}}\left(B_\star\sqrt{SAL_{c,M'}C_{M'}} + B_\star^2 SAL_{c,M'} + B_\star\sqrt{B_\star SAL_{P,M'}(C_{M'}+M')} + B_\star^2 S^2 AL_{P,M'}\right)$$

$$+ \tilde{\mathcal{O}}\left(B_\star\sqrt{S^2 AL_{P,M'}\sum_{m=1}^{M'}\sum_{h=1}^{H_m}\mathbb{V}(P_h^m, \check{V}_{h+1}^{\star,m} - \check{V}_{h+1}^m)} + \sum_{m=1}^{M'}B_\star(\Delta_{c,m} + B_\star\Delta_{P,m})H\right).$$

Then solving a quadratic inequality w.r.t $\sum_{m=1}^{M'}\sum_{h=1}^{H_m}\mathbb{V}(P_h^m, \check{V}_{h+1}^{\star,m} - \check{V}_{h+1}^m)$ (Lemma 45) completes the proof. □

### E.3  Proof of Theorem 5

We first prove a general regret guarantee of Algorithm 5, from which Theorem 5 is a direct corollary.

**Theorem 11.** *Suppose $\mathfrak{A}_1$ ensures $\mathring{R}_K \leq R^1$ when $s_1^m = s_{init}$ for $m \leq K$, and $\mathfrak{A}_2$ ensures $R_{K'} \leq R^2(K')$ for any $K' \leq K$ such that $R^2(k)$ is sub-linear w.r.t $k$. Then Algorithm 5 ensures $R_K = \tilde{\mathcal{O}}(R^1)$ (ignoring lower order terms).*

*Proof.* Let $\mathcal{I}_k$ be the set of intervals in episode $k$, and $m_i^k$ be the $i$-th interval of episode $k$ (if exists). The regret is decomposed as:

$$R_K = \sum_{k=1}^{K}\left[\sum_{h=1}^{H_{m_1^k}}c_h^{m_1^k} + c_{H_{m_1^k}+1}^{m_1^k} - V_k^\star(s_1^k)\right] + \sum_{k=1}^{K}\left[\sum_{m\in\mathcal{I}_k\setminus\{m_1^k\}}\sum_{h=1}^{H_m}c_h^m - c_{H_{m_1^k}+1}^{m_1^k}\right].$$

Note that $V_1^{\pi^\star,m_1^k}(s_1^{m_1^k}) \leq V_k^\star(s_1^k) + B_\star/K$ by Lemma 46. Therefore,

$$\sum_{k=1}^{K}\left[\sum_{h=1}^{H_{m_1^k}}c_h^{m_1^k} + c_{H_{m_1^k}+1}^{m_1^k} - V_k^\star(s_1^k)\right] \leq \sum_{k=1}^{K}\left[\sum_{h=1}^{H_{m_1^k}}c_h^{m_1^k} + c_{H_{m_1^k}+1}^{m_1^k} - V_1^{\pi^\star,m_1^k}(s_1^{m_1^k})\right] + B_\star$$

$$\leq R^1 + B_\star.$$

For the second term, note that $c_{H_{m_1^k}+1}^{m_1^k} = 2B_\star$ if $s_1^{m_2^k}$ exists. Define $K_f = \sum_{k=1}^{K}\mathbb{I}\{|\mathcal{I}_k| > 1\}$, we have (define $s_1^{m_2^k} = g$ if $m_2^k$ does not exist)

$$\sum_{k=1}^{K}\left[\sum_{m\in\mathcal{I}_k\setminus\{m_1^k\}}\sum_{h=1}^{H_m}c_h^m - c_{H_{m_1^k}+1}^{m_1^k}\right] \leq \sum_{k=1}^{K}\left(\sum_{m\in\mathcal{I}_k\setminus\{m_1^k\}}\sum_{h=1}^{H_m}c_h^m - V_k^\star(s_1^{m_2^k})\right) - B_\star K_f$$

$$\leq R^2(K_f) - B_\star K_f,$$

which is a lower order term since $R^2(K_f)$ is sub-linear w.r.t $K_f$. Putting everything together completes the proof. □

We are now ready to prove Theorem 5.

*Proof.* We simply apply Theorem 11 with $R^1$ determined by Theorem 4 and $R^2$ determined by Theorem 3. □

---

**Algorithm 6** MVP-Base

---

**Parameters:** failure probability $\delta$.

**Initialize:** $\widehat{\chi} \leftarrow 0$, and for all $(s, a, s')$, $\mathbf{C}(s, a) \leftarrow 0$, $\mathbf{M}(s, a) \leftarrow 0$, $\mathbf{N}(s, a) \leftarrow 0$, $\mathbf{N}(s, a, s') \leftarrow 0$.

**Initialize:** Update(1).

**for** $m = 1, \ldots, M$ **do**

    **for** $h = 1, \ldots, H$ **do**

        Play action $a_h^m = \mathrm{argmin}_a \check{Q}_h(s_h^m, a)$, receive cost $c_h^m$ and next state $s_{h+1}^m$.

        $\mathbf{C}(s_h^m, a_h^m) \leftarrow c_h^m$, $\mathbf{M}(s_h^m, a_h^m) \overset{+}{\leftarrow} 1$, $\mathbf{N}(s_h^m, a_h^m) \overset{+}{\leftarrow} 1$, $\mathbf{N}(s_h^m, a_h^m, s_{h+1}^m) \overset{+}{\leftarrow} 1$.

        **if** $s_{h+1}^m = g$ *or* $\mathbf{M}(s_h^m, a_h^m) = 2^l$ *or* $\mathbf{N}(s_h^m, a_h^m) = 2^l$ *for some integer* $l \geq 0$ **then**

            **break** (which starts a new interval).

    $\widehat{\chi} \overset{+}{\leftarrow} C^m - \check{V}_1(s_1^m)$.

1    **if** $\widehat{\chi} > \chi_m$ *(defined in Lemma 31)* **then** terminate. **(Test 1)**

    Update($m + 1$).

2    **if** $\left\| \check{V}_h \right\| > B/2$ *for some $h$ (**Test 2**)* **then** terminate.

**Procedure** Update($m$)

    $\check{V}_{H+1}(s) \leftarrow 2B_\star \mathbb{I}\{s \neq g\}$, $\check{V}_h(g) \leftarrow 0$ for all $h \leq H$, and $\iota \leftarrow 2^{11} \cdot \ln\left(\frac{2SAHKm}{\delta}\right)$.

3    $\eta \leftarrow \min\{\frac{B_\star S \sqrt{A}}{T_\star \sqrt{m}}, \frac{1}{2^8 H}\}$.

    **for** *all* $(s, a)$ **do**

        $\mathbf{N}^+(s, a) \leftarrow \max\{1, \mathbf{N}(s, a)\}$, $\mathbf{M}^+(s, a) \leftarrow \max\{1, \mathbf{M}(s, a)\}$, $\bar{c}(s, a) \leftarrow \frac{\mathbf{C}(s,a)}{\mathbf{M}^+(s,a)}$,

        $\bar{P}_{s,a}(\cdot) \leftarrow \frac{\mathbf{N}(s,a,\cdot)}{\mathbf{N}^+(s,a)}$, $\widehat{c}(s, a) \leftarrow \max\left\{0, \bar{c}(s, a) - \sqrt{\frac{\bar{c}(s,a)\iota}{\mathbf{M}^+(s,a)}} - \frac{\iota}{\mathbf{M}^+(s,a)}\right\}$,

4        $\check{c}(s, a) \leftarrow \widehat{c}(s, a) + 8\eta$.

    **for** $h = H, \ldots, 1$ **do**

        $b_h(s, a) \leftarrow \max\left\{7\sqrt{\frac{\mathbb{V}(\bar{P}_{s,a}, \check{V}_{h+1})\iota}{\mathbf{N}^+(s,a)}}, \frac{49B\sqrt{S}\iota}{\mathbf{N}^+(s,a)}\right\}$ for all $(s, a)$.

        $\check{Q}_h(s, a) = \max\{0, \check{c}(s, a) + \bar{P}_{s,a}\check{V}_{h+1} - b_h(s, a)\}$ all $(s, a)$.

        $\check{V}_h(s) = \mathrm{argmin}_a \check{Q}_h(s, a)$ for all $s$.

---

# F  Omitted Details in Section 7

In this section, we present all proofs and details of learning without the knowledge of non-stationarity. We first provide a base algorithm in Appendix F.1. The rest of this section then discusses the meta algorithm MASTER adopted from [Wei and Luo, 2021], and its regret guarantee combining with the base algorithm.

## F.1  Base Algorithm

We first present the base algorithm used in MASTER (Algorithm 6). The main idea is again incorporating a correction term to penalize long horizon policy and has the effect of cancelling the non-stationarity along the learner's trajectory when it is not too large (Line 3). When the non-stationarity is large, on the other hand, we detect it through two non-stationary tests (Line 1 and Line 2), and reset the knowledge of the environment (more details to follow).

**Test 1** is a combination of the first two tests of Algorithm 4, which directly checks whether the estimated regret is too large. This is also similar to the second test of the MASTER algorithm [Wei and Luo, 2021]. **Test 2** is the same as the third test of Algorithm 4, which guards the magnitude of the estimated value function. When tests fail, the algorithm directly terminate instead of resetting some accumulators. Note that the status of $\mathbf{M}$ and $\mathbf{N}$ are completely identical in this algorithm, but we still maintain them separately so that the auxiliary lemmas in Appendix A are still applicable. The rest of the algorithm largely follows the design of Algorithm 2.

**Notations** Note that here $\mathbf{M}$ and $\mathbf{N}$ are only reset at the initialization step. Thus, $i_m^c = i_m^P = 1$, $L_{c,m} = L_{P,m} = 1$, $\Delta_{c,m} = \Delta_{c,[1,m]}$ and $\Delta_{P,m} = \Delta_{P,[1,m]}$. Let $\Delta'_m = (\Delta_{c,m} + B\Delta_{P,m})$ and

denote by $\eta_m$, $\check{Q}_h^m$, $\check{V}_h^m$ the value of $\eta$, $\check{Q}_h$, and $\check{V}_h$ at the beginning of interval $m$. Denote by $\check{c}^m$ the value of $\check{c}$ at the beginning of interval $m$ and define $\check{c}_h^m = \check{c}(s_h^m, a_h^m)$. Also define $\check{Q}_h^{\pi^\star, m}$ and $\check{V}_h^{\pi^\star, m}$ as the action-value function and value function w.r.t cost $c^m(s, a) + 8\eta_m$, transition $P^m$, and policy $\pi_{k(m)}^\star$.

**Lemma 29.** *With probability at least $1 - 2\delta$, if [Algorithm 6] does not terminate up to interval $m \leq K$, then $\check{Q}_h^m(s, a) \leq \check{Q}_h^{\pi^\star, m}(s, a) + \Delta_m' T_h^{\pi^\star, m}(s, a)$.*

*Proof.* We prove this by induction on $h$. The base case of $h = H + 1$ is clearly true. For $h \leq H$, by **Test 2** and the induction step, we have $\check{V}_{h+1}^m(s) \leq \min\{B/2, \check{V}_{h+1}^{\pi^\star, m}(s) + \Delta_m' T_{h+1}^{\pi^\star, m}(s)\} \leq \check{V}_{h+1}^{\pi^\star, m}(s) + x_{h+1}^m(s) \leq B$ where $x_h^m(s) = \min\{B/2, \Delta_m' T_h^{\pi^\star, m}(s)\}$. Thus,

$$\check{c}^m(s, a) + \bar{P}_{s,a}^m \check{V}_{h+1}^m - b^m(s, a, \check{V}_{h+1}^m)$$
$$\leq \check{c}^m(s, a) + \bar{P}_{s,a}^m(\check{V}_{h+1}^{\pi^\star, m} + x_{h+1}^m) - b^m(s, a, \check{V}_{h+1}^{\pi^\star, m} + x_{h+1}^m) \qquad \text{(Lemma 48)}$$
$$\overset{(i)}{\leq} \check{c}^m(s, a) + \widetilde{P}_{s,a}^m(\check{V}_{h+1}^{\pi^\star, m} + x_{h+1}^m) \qquad \text{(Lemma 8)}$$
$$\leq c^m(s, a) + 8\eta_m + \Delta_{c,m} + P_{s,a}^m(\check{V}_{h+1}^{\pi^\star, m} + x_{h+1}^m) + \Delta_{P,m} B \qquad \text{(Lemma 5)}$$
$$\leq \check{Q}_h^{\pi^\star, m}(s, a) + \Delta_m' T_h^{\pi^\star, m}(s, a).$$

Note that in (i) we use the fact that $|\{\check{V}_h^{\pi^\star, m} + x_h^m\}_{m,h}| \leq (HK+1)^6$ since $|\{V_h^{\pi^\star, m}\}_{m,h}| \leq HK+1$, $|\{\eta_m\}_m| \leq K+1$, $|\{\Delta_m'\}_m| \leq K+1$, and $|\{T_h^{\pi^\star, m}\}_{m,h}| \leq HK+1$. $\qquad \square$

**Lemma 30.** *With probability at least $1 - 2\delta$, for all $m \leq K$, if $\Delta_m' \leq \eta_m$, then $\check{Q}_h^m(s, a) \leq \check{Q}_h^{\pi^\star, m}(s, a) + \eta_m T_h^{\pi^\star, m}(s, a) \leq B/2$. Moreover, if **Test 2** fails in interval $m$, then $\Delta_{m+1}' > \eta_{m+1}$.*

*Proof.* First note that $\check{Q}_h^{\pi^\star, m}(s, a) \leq \frac{B}{4} + 8\eta_m T_h^{\pi^\star, m}(s, a) \leq \frac{B}{4} + 8H\eta_m \leq \frac{B}{3}$. We prove the first statement by induction on $h$. The base case of $h = H + 1$ is clearly true. For $h \leq H$, note that:

$$\check{c}^m(s, a) + \bar{P}_{s,a}^m \check{V}_{h+1}^m - b^m(s, a, \check{V}_{h+1}^m)$$
$$\leq \check{c}^m(s, a) + \bar{P}_{s,a}^m(\check{V}_{h+1}^{\pi^\star, m} + \eta_m T_{h+1}^{\pi^\star, m}) - b^m(s, a, \check{V}_{h+1}^{\pi^\star, m} + \eta_m T_{h+1}^{\pi^\star, m})$$
$$\text{(induction step and Lemma 48)}$$
$$\overset{(i)}{\leq} \check{c}^m(s, a) + \widetilde{P}_{s,a}^m(\check{V}_{h+1}^{\pi^\star, m} + \eta_m T_{h+1}^{\pi^\star, m}) \qquad \text{(Lemma 8)}$$
$$\leq c^m(s, a) + 8\eta_m + \Delta_{c,m} + P_{s,a}^m(\check{V}_{h+1}^{\pi^\star, m} + \eta_m T_{h+1}^{\pi^\star, m}) + \Delta_{P,m}(B/3 + H\eta_m) \qquad \text{(Lemma 5)}$$
$$\leq \check{Q}_h^{\pi^\star, m}(s, a) + \eta_m T_h^{\pi^\star, m}(s, a). \qquad \text{($H\eta_m \leq B/12$ and $\Delta_m' \leq \eta_m$)}$$

Note that in (i) we use the fact that $|\{\check{V}_h^{\pi^\star, m} + \eta_m T_h^{\pi^\star, m}\}_{m,h}| \leq (HK+1)^6$ since $|\{V_h^{\pi^\star, m}\}_{m,h}| \leq HK+1$, $|\{\eta_m\}_m| \leq K+1$, and $|\{T_h^{\pi^\star, m}\}_{m,h}| \leq HK+1$. The second statement is simply by the contraposition of the first statement. $\qquad \square$

**Lemma 31.** *With probability at least $1 - 12\delta$, for any $M' \leq K$, if $\Delta_{M'}' \leq \eta_{M'}$, then*

$$\sum_{m=1}^{M'} \left( \sum_{h=1}^{H_m} c_h^m + c_{H_m+1}^m - \check{V}_1^m(s_1^m) \right) = \tilde{\mathcal{O}}\left( B_\star S\sqrt{AM'} + B_\star S^2 A \right) \triangleq \chi_{M'}.$$

*Moreover, if **Test 1** fails in interval $m$, then $\Delta_m' > \eta_m$.*

*Proof.* By $\Delta'_{M'} \leq \eta_{M'}$ and Lemma 30, the algorithm will not terminate by **Test 2** before interval $M'$ with probability at least $1 - 2\delta$. Then with probability at least $1 - 4\delta$,

$$\sum_{m=1}^{M'} \left( \sum_{h=1}^{H_m} c_h^m + c_{H_m+1}^m - \check{V}_1^m(s_1^m) \right)$$

$$\leq \sum_{m=1}^{M'} \sum_{h=1}^{H_m} \left( c_h^m + \check{V}_{h+1}^m(s_{h+1}^m) - \check{V}_h^m(s_h^m) \right) + \tilde{\mathcal{O}}\left(B_\star SA\right) \qquad \text{(Lemma 12 and } L_{M'} = \mathcal{O}(1))$$

$$\leq \sum_{m=1}^{M'} \sum_{h=1}^{H_m} \left( c_h^m - \hat{c}_h^m + \check{V}_{h+1}^m(s_{h+1}^m) - P_h^m \check{V}_{h+1}^m + (P_h^m - \bar{P}_h^m)\check{V}_{h+1}^m + b_h^m - 8\eta_m \right) + \tilde{\mathcal{O}}\left(B_\star SA\right)$$

$$\text{(definition of } \check{V}_h^m(s_h^m))$$

$$\leq \tilde{\mathcal{O}}\left( \sqrt{SAC_{M'}} + \sqrt{\sum_{m=1}^{M'} \sum_{h=1}^{H_m} \mathbb{V}(P_h^m, \check{V}_{h+1}^m) + B_\star SA} \right)$$

$$+ \sum_{m=1}^{M'} \sum_{h=1}^{H_m} \left( (\widetilde{P}_h^m - \bar{P}_h^m)\check{V}_{h+1}^m + B\mathbf{n}_h^m + b_h^m - 5\eta_m \right),$$

where in the last inequality we apply Lemma 3, $i_{M'}^c = i_{M'}^P = 1$, $\Delta'_{M'} \leq \eta_{M'}$, $P_h^m \check{V}_{h+1}^m \leq \widetilde{P}_h^m \check{V}_{h+1}^m + B(\mathbf{n}_h^m + \Delta_{P,m})$, Lemma 49 and Lemma 50 on both $\sum_{m=1}^{M'} \sum_{h=1}^{H_m} (c_h^m - c^m(s_h^m, a_h^m))$, and Lemma 49 on $\sum_{m=1}^{M'} \sum_{h=1}^{H_m} (\check{V}_{h+1}^m(s_{h+1}^m) - P_h^m \check{V}_{h+1}^m)$. Now note that with probability at least $1 - 6\delta$,

$$\sum_{m=1}^{M'} \sum_{h=1}^{H_m} ((\widetilde{P}_h^m - \bar{P}_h^m)\check{V}_{h+1}^m + b_h^m + B\mathbf{n}_h^m) + \tilde{\mathcal{O}}\left( \sqrt{\sum_{m=1}^{M'} \sum_{h=1}^{H_m} \mathbb{V}(P_h^m, \check{V}_{h+1}^m)} \right)$$

$$= \tilde{\mathcal{O}}\left( \sqrt{S^2 A \sum_{m=1}^{M'} \sum_{h=1}^{H_m} \mathbb{V}(P_h^m, \check{V}_{h+1}^m) + B_\star S^2 A} \right) + \sum_{m=1}^{M'} \sum_{h=1}^{H_m} b_h^m + \sum_{m=1}^{M'} \sum_{h=1}^{H_m} \frac{B\Delta_{P,m}}{64}$$

$$\text{(}\mathbf{n}_h^m \leq \frac{1}{\mathbf{N}_h^m}\text{, Lemma 6, Cauchy-Schwarz inequality, Lemma 11, and } L_{P,M'} = 1)$$

$$= \tilde{\mathcal{O}}\left( \sqrt{S^2 A \sum_{m=1}^{M'} \sum_{h=1}^{H_m} \mathbb{V}(P_h^m, \check{V}_{h+1}^m) + B_\star S^2 A} \right) + \sum_{m=1}^{M'} \sum_{h=1}^{H_m} \frac{B\Delta_{P,m}}{32}.$$

$$\text{(Lemma 10, } L_{P,M'} = 1\text{, and AM-GM inequality)}$$

$$= \tilde{\mathcal{O}}\left( \sqrt{B_\star S^2 A(C_{M'} + M')} + B_\star S^2 A \right) + \sum_{m=1}^{M'} \sum_{h=1}^{H_m} \frac{\Delta'_m}{16}. \quad \text{(Lemma 32 and AM-GM inequality)}$$

Plugging this back and by $\Delta'_{M'} \leq \eta_{M'}$, we have

$$C_{M'} - \sum_{m=1}^{M'} \check{V}_1^m(s_1^m) = \tilde{\mathcal{O}}\left( \sqrt{B_\star S^2 A(C_{M'} + M')} + B_\star S^2 A \right).$$

Solving a quadratic inequality w.r.t $C_{M'}$ (Lemma 45), we have $C_{M'} = \tilde{\mathcal{O}}(B_\star M' + \sqrt{B_\star S^2 AM'} + B_\star S^2 A)$. Plugging this back completes the proof of the first statement. The second statement is simply by the contraposition of the first statement. □

**Theorem 12.** *Suppose Algorithm 4 does not terminate up to interval $M' \leq K$ (including $M'$) and $s_1^m = s_{init}$ for $m \leq M'$. Then with probability at least $1 - 2\delta$, $\mathring{R}_{M'} = \tilde{\mathcal{O}}(B_\star S\sqrt{AM'} + B_\star S^2 A + \sum_{m=1}^{M'} \Delta'_m T_\star)$.*

*Proof.* We decompose the regret as follows:

$$
\mathring{R}_{M'} = \sum_{m=1}^{M'} \left( \sum_{h=1}^{H_m} c_h^m + c_{H_m+1}^m - V_1^{\pi^\star,m}(s_1^m) \right)
$$

$$
= \sum_{m=1}^{M'} \left( \sum_{h=1}^{H_m} c_h^m + c_{H_m+1}^m - \check{V}_1^m(s_1^m) \right) + \sum_{m=1}^{M'} \left( \check{V}_1^m(s_1^m) - \check{V}_1^{\pi^\star,m}(s_1^m) \right) + 8T_\star \sum_{m=1}^{M'} \eta_m
$$

$$
\leq \chi_{M'} + \sum_{m=1}^{M'} \Delta_m' T_\star + 8T_\star \sum_{m=1}^{M'} \eta_m. \qquad \text{(\textbf{Test 2} and Lemma 29)}
$$

Plugging in the definition of $\chi_{M'}$ and $\eta_m$ completes the proof. $\qquad \square$

**Lemma 32.** *With probability at least $1 - 4\delta$, $\sum_{m=1}^{M'} \sum_{h=1}^{H_m} \mathbb{V}(P_h^m, \check{V}_{h+1}^m) = \tilde{\mathcal{O}}(B_\star(C_{M'} + M') + B_\star^2 S^2 A + B_\star \sum_{m=1}^{M'} \sum_{h=1}^{H_m} \Delta_m')$ for any $M' \leq K$.*

*Proof.* Applying Lemma 9 with $\left\| \check{V}_h^m \right\|_\infty \leq B$ (**Test 2**), with probability at least $1 - \delta$,

$$
\sum_{m=1}^{M'} \sum_{h=1}^{H_m} \mathbb{V}(P_h^m, \check{V}_{h+1}^m)
$$

$$
= \tilde{\mathcal{O}} \left( \sum_{m=1}^{M'} \check{V}_{H_m+1}^m(s_{H_m+1}^m)^2 + \sum_{m=1}^{M'} \sum_{h=1}^{H_m} B_\star(\check{V}_h^m(s_h^m) - P_h^m \check{V}_{h+1}^m)_+ + B_\star^2 \right)
$$

$$
= \tilde{\mathcal{O}} \left( B_\star(C_{M'} + M') + B_\star \sqrt{S^2 A \sum_{m=1}^{M'} \sum_{h=1}^{H_m} \mathbb{V}(P_h^m, \check{V}_{h+1}^m)} + B_\star^2 S^2 A + B_\star \sum_{m=1}^{M'} \sum_{h=1}^{H_m} \Delta_m' \right),
$$

where in the last step we apply

$$
\sum_{m=1}^{M'} \sum_{h=1}^{H_m} (\check{V}_h^m(s_h^m) - P_h^m \check{V}_{h+1}^m)_+ = \sum_{m=1}^{M'} \sum_{h=1}^{H_m} (\check{Q}_h^m(s_h^m, a_h^m) - P_h^m \check{V}_{h+1}^m)_+
$$

$$
\leq \sum_{m=1}^{M'} \sum_{h=1}^{H_m} (\check{c}_h^m + (\bar{P}_h^m - \widetilde{P}_h^m)\check{V}_{h+1}^m + B\Delta_{P,m})_+
$$

$$
\qquad\qquad ((a)_+ - (b)_+ \leq (a - b)_+, \text{ definition of } \check{Q}_h^m, \text{ and } b_h^m \geq 0)
$$

$$
\leq \sum_{m=1}^{M'} \sum_{h=1}^{H_m} c^m(s_h^m, a_h^m) + M' + \tilde{\mathcal{O}} \left( \sqrt{S^2 A \sum_{m=1}^{M'} \sum_{h=1}^{H_m} \mathbb{V}(P_h^m, \check{V}_{h+1}^m) + B_\star S^2 A} \right) + 2 \sum_{m=1}^{M'} \sum_{h=1}^{H_m} \Delta_m'
$$

$$
\qquad\qquad \text{(Lemma 5, } 8\eta_m \leq \tfrac{1}{H}, \text{ Lemma 6, Cauchy-Schwarz inequality, and Lemma 11)}
$$

$$
\leq \tilde{\mathcal{O}} \left( \sum_{m=1}^{M'} \sum_{h=1}^{H_m} c_h^m + M' + \sqrt{S^2 A \sum_{m=1}^{M'} \sum_{h=1}^{H_m} \mathbb{V}(P_h^m, \check{V}_{h+1}^m) + B_\star S^2 A} \right) + 2 \sum_{m=1}^{M'} \sum_{h=1}^{H_m} \Delta_m'.
$$

$$
\qquad\qquad \text{(Lemma 50)}
$$

Solving a quadratic inequality w.r.t $\sum_{m=1}^{M'} \sum_{h=1}^{H_m} \mathbb{V}(P_h^m, \check{V}_{h+1}^m)$ (Lemma 45) completes the proof. $\qquad \square$

## F.2 Preliminaries

Here we adopt the MASTER algorithm in [Wei and Luo, 2021] to our finite-horizon approximation scheme. There are several issues we need to address: 1) under the protocol of Algorithm 1, the total number of intervals and the non-stationarity in each interval are not fixed before learning start; besides, we need to prove an anytime regret guarantee, so that it can translate back to a regret

guarantee on the original SSP (see Lemma 16); 2) when the base algorithm has a regret guarantee $\mathring{R}_m \le \min\{c_1\sqrt{m} + c_2, c_3 m\}$ without non-stationarity, the original MASTER algorithm ensures a dynamic regret whose dominating term scale with $c_1 + c_2 c_3/c_1$; this is undesirable as $c_3 = \tilde{\mathcal{O}}(T_{\max})$ in our case, and ideally we want $c_3 = \tilde{\mathcal{O}}(B_\star)$; 3) when base algorithms incorporate correction term, the original analysis of the non-stationarity tests breaks as discussed in Section 7; 4) The analysis in [Wei and Luo, 2021, Lemma 17] that bounds the cost of non-stationary detection only works for oblivious adversary. Our modified MASTER algorithm (Algorithm 8) manages to address all these issues.

**Setup**  To give a general result, we define the dynamic regret for the first $M'$ intervals as $\widetilde{R}_{M'} = \sum_{m=1}^{M'}(C^m - f_m^\star)$, where the choice of benchmark $\{f_m^\star\}_{m=1}^{M'}$ is flexible depending on the problem and the algorithm.

**Notations**  For any interval $\mathcal{I} = [s, e]$, define $\Delta_{\mathcal{I}} = \sum_{m=s}^{e-1}\Delta(m)$ and $L_{\mathcal{I}} = 1 + \sum_{m=s}^{e-1}\mathbb{I}\{\Delta(m) \ne 0\}$, where $\Delta(m) \in \mathbb{R}_+^{\mathbb{N}_+}$ is some non-stationarity measure satisfying $|f_{m+1}^\star - f_m^\star| \le \Delta(m)$.

We make the following assumption on the base algorithm used in the MASTER algorithm, and then show two algorithms satisfying the assumption.

**Assumption 1.** *Base algorithm $\mathfrak{A}$ with failure probability $\delta$ on intervals $[1, M']$ outputs an estimate $\widetilde{f}_m$ at the beginning of interval $m \le M'$ if it does not terminate before interval $m$. Moreover, there exists a non-decreasing function $R(m) = \min\{c_1\sqrt{m} + c_2, c_3 m\}$ with $c_3 \ge 1$ and non-stationarity measure $\Delta$ such that $r(m) = R(m)/m$ is non-increasing, $r(m) \ge \frac{1}{\sqrt{m}}$, $\widetilde{f}_m \le c_4 \le c_3$ for all $m$, and with probability at least $1 - \delta$, for any $m \le M'$, as long as $\Delta_{[1,m]} \le r(m)$ and $\mathfrak{A}$ does not terminate up to interval $m$ (including $m$), without knowing $\Delta_{[1,m]}$ we have:*

$$\widetilde{f}_m \le f_m^\star + r(m), \quad \sum_{\tau=1}^m \left(C^\tau - \widetilde{f}_\tau\right) \le R(m), \ and \quad \sum_{\tau=1}^m (f_\tau^\star - C^\tau) \le R(m).$$

**Lemma 33.** *Algorithm 2 with arbitrary initial state for each interval satisfies Assumption 1 with $f_m^\star = V_1^{\star,m}(s_1^m)$, $\widetilde{f}_m = V_1^m(s_1^m)$, $\Delta(m) = \tilde{\mathcal{O}}((\Delta_{c,[m,m+1]} + B\Delta_{P,[m,m+1]})H)$, $R(m) = \tilde{\mathcal{O}}(\min\{B_\star S\sqrt{Am} + B_\star S^2 A, Hm\})$, and $c_4 = \tilde{\mathcal{O}}(B_\star)$.*

*Proof.* The first two properties are simply by Lemma 18 and Theorem 9 with $L_{c,m} = L_{P,m} = 1$ and $\Delta_{[1,m]} \le r(m)$ with a large enough constant hidden in $\tilde{\mathcal{O}}(\cdot)$ in the definition of $\Delta(m)$. For the third property, with high probability,

$$\sum_{\tau=1}^m (f_\tau^\star - C^\tau) = \sum_{\tau=1}^m (V_1^{\star,\tau}(s_1^m) - C^\tau) = \tilde{\mathcal{O}}\left(\sqrt{B_\star \sum_{\tau=1}^m C^\tau} + B_\star\right). \qquad \text{(Lemma 35)}$$

$$= \tilde{\mathcal{O}}\left(\sqrt{B_\star \left(\sum_{\tau=1}^m \widetilde{f}_\tau + R(m)\right)} + B_\star\right) \le \tilde{\mathcal{O}}\left(B_\star\sqrt{m} + B_\star\right) + \frac{1}{2}R(m).$$
$$\text{(the second property, } V_1^m(s_1^m) = \tilde{\mathcal{O}}(B_\star)\text{, and AM-GM inequality)}$$

Plugging in the definition of $R(m)$ completes the proof (again with a large enough constant hidden in $\tilde{\mathcal{O}}(\cdot)$ in the definition of $R(m)$). $\qquad \square$

**Lemma 34.** *Algorithm 6 with $m \le K$ and $s_1^m = s_{init}$ satisfies Assumption 1 with $f_m^\star = V_1^{\pi^\star,m}(s_{init})$, $\widetilde{f}_m = \check{V}_1^m(s_1^m)$, $\Delta(m) = \tilde{\mathcal{O}}((\Delta_{c,[m,m+1]} + B\Delta_{P,[m,m+1]})T_\star)$, $R(m) = \tilde{\mathcal{O}}(\min\{B_\star S\sqrt{Am} + B_\star S^2 A, Hm\})$, and $c_4 = \tilde{\mathcal{O}}(B_\star)$.*

*Proof.* For the first property, by Lemma 29, $\Delta_{[1,m]} \le r(m)$ and a large enough constant hidden in $\tilde{\mathcal{O}}(\cdot)$ in the definition of $\Delta(m)$, we have

$$\widetilde{f}_m = \check{V}_1^m(s_1^m) \le \check{V}_1^{\pi^\star,m}(s_1^m) + \Delta_m' T_\star \le f_m^\star + 8T_\star\eta_m + \tilde{\mathcal{O}}\left(\Delta_{[1,m]}\right) \le f_m^\star + r(m).$$

---
**Algorithm 7** MALG
---
**Input:** order $n$, regret density function $r$.

**for** $l = 0, \ldots, n$ **do**

    **for** $m \in \{0, 2^l, 2 \cdot 2^l, \ldots, 2^n - 2^l\}$ **do**

        With probability $\frac{r(2^n)}{r(2^m)}$, assigns a new base algorithm on intervals $[m+1, m+2^l]$.

**for** *each interval* $m$ **do**

    Let $\mathfrak{A}$ be the algorithm that covers interval $m$ with shortest scheduled length, output $\widetilde{g}_m = \widetilde{f}_m^{\mathfrak{A}}$ (which is the $\widetilde{f}_m$ output by $\mathfrak{A}$), follow $\mathfrak{A}$'s decision, and update $\mathfrak{A}$ with environment's feedback.

    **if** $\mathfrak{A}$ *terminates* **then** terminate.

---

The second property is simply by **Test 2** (Lemma 31) of Algorithm 6 (again with a large enough constant hidden in $\tilde{\mathcal{O}}(\cdot)$ in the definition of $R(m)$). For the third property,

$$\sum_{\tau=1}^{m}(f_\tau^\star - C^\tau) = \sum_{\tau=1}^{m}(V_1^{\pi^\star,\tau}(s_{\text{init}}) - C^\tau) \leq \frac{B_\star m}{K} + \sum_{\tau=1}^{m}(V_1^{\star,\tau}(s_{\text{init}}) - C^\tau) \leq R(m),$$

where the first inequality is by Lemma 46 and the last step follows similar arguments as in Lemma 33. $\square$

**Lemma 35.** *With probability at least* $1 - 3\delta$, *for any* $m \leq M$, $\sum_{\tau=1}^{m}(V_1^{\star,\tau}(s_1^\tau) - C^\tau) = \tilde{\mathcal{O}}(\sqrt{B_\star \sum_{\tau=1}^{m} C^\tau} + B_\star)$.

*Proof.* With probability at least $1 - 3\delta$,

$$\sum_{\tau=1}^{m}(V_1^{\star,\tau}(s_1^m) - C^\tau) \leq \sum_{\tau=1}^{m}\sum_{h=1}^{H_\tau}(V_h^{\star,\tau}(s_h^\tau) - V_{h+1}^{\star,\tau}(s_{h+1}^\tau) - c_h^\tau) \qquad (V_{H_\tau+1}^{\star,\tau}(s_{H_\tau+1}^\tau) \leq c_{H_\tau+1}^\tau)$$

$$\leq \sum_{\tau=1}^{m}\sum_{h=1}^{H_\tau}(P_h^\tau V_{h+1}^{\star,\tau} - V_{h+1}^{\star,\tau}(s_{h+1}^\tau)) = \tilde{\mathcal{O}}\left(\sqrt{\sum_{\tau=1}^{m}\sum_{h=1}^{H_\tau}\mathbb{V}(P_h^\tau, V_{h+1}^{\star,\tau})} + B_\star\right)$$

$$\qquad\qquad (V_h^{\star,\tau}(s_h^\tau) \leq Q_h^{\star,\tau}(s_h^\tau, a_h^\tau) \text{ and Lemma 49})$$

$$= \tilde{\mathcal{O}}\left(\sqrt{B_\star \sum_{\tau=1}^{m} C^\tau} + B_\star\right). \qquad\qquad \text{(Lemma 19)}$$

$\square$

### F.3  MALG: Multi-Scale Learning with Base Algorithm

Following [Wei and Luo, 2021, Section 3], we first introduce MALG (Algorithm 7), which runs multiple instances of base algorithms in a multi-scale manner. We then combine MALG with non-stationarity detection to obtain the MASTER algorithm in Appendix F.4. We always run MALG on a segment (an interval of intervals) of length $2^n$ for some integer $n$, which we call a *block*. Since we want to obtain an anytime regret guarantee, the failure probability of base algorithms and MALG need to be adjusted adaptively. Specifically, if an MALG instance is scheduled on intervals $[M_\dagger - 2^n + 1, M_\dagger]$, then the regret guarantee of this MALG instance and the failure probability of base algorithms it maintains depends on $M_\dagger$. However, we ignore the dependency on $M_\dagger$ in algorithms and analysis since the regret bound only has logarithmic dependency on $M_\dagger$.

We show that MALG ensures a multi-scale regret guarantee in the following lemma. Below we say an algorithm is of order $l$ if it is scheduled on a segment of length $2^l$. Also denote by $\widetilde{f}_m^{\mathfrak{A}}$ the $\widetilde{f}_m$ output by $\mathfrak{A}$.

**Lemma 36.** *For a given* $M_\dagger \geq 1$, *let* $\widehat{n} = \log_2 M_\dagger + 1$ *and* $\widehat{R}(m) = 2^{10}\widehat{n}\ln(2M_\dagger/\delta)R(m)$. *Algorithm 7 scheduled on* $[M_\dagger - 2^n + 1, M_\dagger]$ *with input* $n \leq \log_2 M_\dagger$ *guarantees for any* $\mathfrak{A}$ *it maintains and any* $m \in [\mathfrak{A}.s, \mathfrak{A}.e]$, *as long as* $\Delta_{[\mathfrak{A}.s,m]} \leq r(m')$ *where* $m' = m - \mathfrak{A}.s + 1$ *and all*

*base algorithms it maintains do not terminate up to interval $m$ (including $m$), we have with high probability:*

$$\widetilde{g}_m \leq f_m^\star + r(m''), \quad \sum_{\tau=\mathfrak{A}.s}^{m} (C^\tau - \widetilde{g}_\tau) \leq \widehat{R}(m'), \quad \text{and} \quad \sum_{\tau=\mathfrak{A}.s}^{m} (f_\tau^\star - C^\tau) \leq \widehat{R}(m'),$$

*where $m''$ is the number of intervals that $\mathfrak{A}'$ is active up to interval $m$, and $\mathfrak{A}'$ is the active algorithm in interval $m$.*

*Proof.* Fix a base algorithm $\mathfrak{A}$ and $m \in [\mathfrak{A}.s, \mathfrak{A}.e]$. Suppose $\mathfrak{A}'$ is active in interval $m$, which implies $[\mathfrak{A}'.s, \mathfrak{A}'.e] \subseteq [\mathfrak{A}.s, \mathfrak{A}.e]$. For the first statement, note that $\Delta_{[\mathfrak{A}'.s, m]} \leq \Delta_{[\mathfrak{A}.s, m]} \leq r(m') \leq r(m'')$ since $r$ is non-increasing. Thus, by the guarantee of $\mathfrak{A}'$ (Assumption 1), we have

$$\widetilde{g}_m \leq f_m^\star + r(m'').$$

For the second statement, first note that:

$$\sum_{\tau=\mathfrak{A}.s}^{m} (C^\tau - \widetilde{g}_\tau) = \sum_{l=0}^{n} \sum_{\mathfrak{A}' \in \mathcal{S}_l} \sum_{\tau=\mathfrak{A}.s}^{m} (C^\tau - \widetilde{f}_\tau^{\mathfrak{A}'}) \mathbb{I}\{\mathfrak{A}' \text{ is active at } \tau\},$$

where $\mathcal{S}_l$ is the set of base algorithms of order $l$ which starts within $[\mathfrak{A}.s, m]$. For a fix $l$, suppose $\mathcal{S}_l = \{\mathfrak{A}_1', \ldots, \mathfrak{A}_N'\}$, and define $\mathcal{I}_i = [\mathfrak{A}.s, m] \cap [\mathfrak{A}_i'.s, \mathfrak{A}_i'.e]$. Note that $\{\mathcal{I}_i\}_{i=1}^N$ are disjoint, and $\Delta_{\mathcal{I}_i} \leq \Delta_{[\mathfrak{A}.s, m]} \leq r(m') \leq r(|\mathcal{I}_i|)$. Moreover, $[\mathfrak{A}_i'.s, \mathfrak{A}_i'.e] \subseteq [\mathfrak{A}.s, \mathfrak{A}.e]$ if $\mathfrak{A}_i'$ is active at some interval within $[\mathfrak{A}.s, m]$. Therefore, by the the guarantee of $\mathfrak{A}_i'$ (Assumption 1) we have:

$$\sum_{i=1}^{N} \sum_{\tau=\mathfrak{A}.s}^{m} (C^\tau - \widetilde{f}_\tau^{\mathfrak{A}_i'}) \mathbb{I}\{\mathfrak{A}_i' \text{ is active at } \tau\} \leq \sum_{i=1}^{N} R(|\mathcal{I}_i|) \leq N \cdot R(\min\{2^l, m'\}).$$

Now we need to bound $N$. Note that $\mathbb{E}[N] \leq \frac{r(2^n)}{r(2^l)}(\frac{m'}{2^l} + 1)$ by the scheduling rule. By Lemma 50, with probability at least $1 - \frac{\delta}{(2M_\dagger)^6}$ (simply choose a small enough failure probability such that the failure probability over all $M_\dagger \geq 1$ and all base algorithms is bounded), $N \leq 2\mathbb{E}[N] + 2^8 \ln(2M_\dagger/\delta) \leq \frac{2r(2^n)}{r(2^l)} \frac{m'}{2^l} + 258 \ln(2M_\dagger/\delta)$ and

$$N \cdot R(\min\{2^l, m'\}) \leq \left( \frac{2r(2^n)}{r(2^l)} \frac{m'}{2^l} + 258 \ln(2M_\dagger/\delta) \right) R(\min\{2^l, m'\})$$

$$\leq \left( \frac{2R(m')}{R(2^l)} + 258 \ln(2M_\dagger/\delta) \right) R(\min\{2^l, m'\}) \leq 2^9 \ln(2M_\dagger/\delta) R(m').$$

$$(r(2^n) \leq r(m'))$$

Summing over $l$ and by $n+1 \leq \widehat{n}$ proves the second statement. For the third statement, by Lemma 35,

$$\sum_{\tau=\mathfrak{A}.s}^{m} (f_\tau^\star - C^\tau) = \tilde{\mathcal{O}}\left( \sqrt{B_\star \sum_{\tau=\mathfrak{A}.s}^{m} C^\tau} + B_\star \right) = \tilde{\mathcal{O}}\left( \sqrt{B_\star \left( \sum_{\tau=\mathfrak{A}.s}^{m} \widetilde{g}_\tau + \widehat{R}(m') \right)} + B_\star \right)$$

(the second statement)

$$\leq \tilde{\mathcal{O}}\left( B_\star \sqrt{m'} \right) + \frac{1}{2}\widehat{R}(m') \leq \widehat{R}(m').$$

$$(\widetilde{g}_\tau \leq c_4 = \tilde{\mathcal{O}}(B_\star) \text{ and AM-GM inequality})$$

This completes the proof. $\qquad\square$

### F.4 Non-stationarity Detection: Single Block Regret Analysis

Now we introduce the MASTER algorithm (Algorithm 8) that performs non-stationarity tests and restarts. We first show the regret bound on a single block of order $n$ (of length $2^n$) that starts from $m_n$ and ends on $E_n$. Clearly $E_n \leq m_n + 2^n - 1$ since it may terminate earlier than planned. Also let $M_\dagger = m_n + 2^n - 1$ be the planned last interval. Define $\widehat{r}(m) = \widehat{R}(m)/m$, $\alpha_l = r(2^l)$, $\widehat{\alpha}_l = \widehat{r}(2^l)$, and $l_0 = \max_l\{12\widehat{\alpha}_{l-1} > c_4\}$. We divide the whole block $[m_n, E_n]$ into near-stationary segments $\mathcal{I}_1, \ldots, \mathcal{I}_\ell$ with $\mathcal{I}_i = [s_i, e_i]$, such that $\Delta_{\mathcal{I}_i} \leq r(|\mathcal{I}_i|)$ and $\Delta_{[s_i, e_i+1]} > r(|\mathcal{I}_i| + 1)$ for $i < \ell$. Note that the partition depends on the learner's behavior, but whether $m \in \mathcal{I}_i$ is determined at the beginning of interval $m$ before interaction starts. In the following lemma we give a bound on $\ell$.

---

**Algorithm 8** MASTER

---

**Input:** $\widehat{r}(\cdot)$ (defined in Appendix F.4).
**Initialize:** $m \leftarrow 1$.

1 **for** $n = 0, 1, \dots$ **do**

    Set $m_n \leftarrow m$, and initialize a MALG (Algorithm 7) instance on $[m_n, m_n + 2^n - 1]$.

    **while** $m < m_n + 2^n$ **do**

        Receive $\widetilde{g}_m$ from MALG, follow MALG's decision, and suffer $C^m$.

2         Update MALG and set $U_m^l = \max_{\tau \in [m_n + 2^l - 1, m]} \widetilde{g}_\tau^l$ for all $0 \leq l \leq n$, where $\widetilde{g}_\tau^l = \frac{1}{2^l} \sum_{\tau' = \tau - 2^l + 1}^{\tau} \widetilde{g}_{\tau'}$ and $U_m^l = 0$ if $m < m_n + 2^l - 1$.

        Perform **Test 1** and **Test 2**, and increment $m \leftarrow m + 1$.

        **if** *either test fails or MALG terminates* **then** restart from Line 1

---

3 **Test 1**: If $m = \mathfrak{A}.e$ for some order-$l$ $\mathfrak{A}$ and $\frac{1}{2^l} \sum_{\tau = \mathfrak{A}.s}^{\mathfrak{A}.e} C^\tau \leq U_m^l - 9\widehat{r}(2^l)$, return *fail*.

  **Test 2**: If $\frac{1}{m - m_n + 1} \sum_{\tau = m_n}^{m} (C^\tau - \widetilde{g}_\tau) \geq 3\widehat{r}(m - m_n + 1)$. return *fail*.

---

**Lemma 37.** *Let* $\mathcal{J} = [m_n, E_n]$. *We have* $\ell \leq L_\mathcal{J}$ *and* $\ell \leq 1 + (2c_1^{-1} \Delta_\mathcal{J})^{2/3} |\mathcal{J}|^{1/3} + c_3^{-1} \Delta_\mathcal{J}$.

*Proof.* The first statement is clearly true. For the second statement follows from Lemma 13. $\square$

We also define $\widetilde{g}_\tau^l = \frac{1}{2^l} \sum_{\tau' = \tau - 2^l + 1}^{\tau} \widetilde{g}_{\tau'}$ and $f_\tau^{\star, l} = \frac{1}{2^l} \sum_{\tau' = \tau - 2^l + 1}^{\tau} f_{\tau'}^\star$ for $\tau \geq m_n + 2^l - 1$. We first show a running average version of the first statement in Lemma 36.

**Lemma 38.** *For any* $\tau \geq m_n + 2^l - 1$, *if for any* $m \in [\tau - 2^l + 1, \tau]$, $\Delta_{[\mathfrak{A}.s, m]} \leq r(m - \mathfrak{A}.s + 1)$ *where* $\mathfrak{A}$ *is the base algorithm of MALG active in interval* $m$, *then* $\widetilde{g}_\tau^l \leq f_\tau^{\star, l} + \widehat{\alpha}_l$ *with high probability.*

*Proof.* The case of $l = 0$ is clearly true by Lemma 36. For $l > 0$, we have

$$
\widetilde{g}_\tau^l = \frac{1}{2^l} \sum_{\tau' = \tau - 2^l + 1}^{\tau} \widetilde{g}_{\tau'} = \frac{1}{2^l} \sum_{\tau' = \tau - 2^l + 1}^{\tau} \sum_{l'=0}^{n} \sum_{\mathfrak{A}' \in \mathcal{S}_{l'}} \widehat{f}_{\tau'}^{\mathfrak{A}'} \mathbb{I}\{\mathfrak{A}' \text{ is active at } \tau'\}
$$

$$
\leq \frac{1}{2^l} \sum_{\tau' = \tau - 2^l + 1}^{\tau} \sum_{l'=0}^{n} \sum_{\mathfrak{A}' \in \mathcal{S}_{l'}} (f_{\tau'}^\star + r(m_{\tau'}^{\mathfrak{A}'})) \mathbb{I}\{\mathfrak{A}' \text{ is active at } \tau'\}
$$

$$
(\Delta_{[\mathfrak{A}'.s, m]} \leq r(m - \mathfrak{A}'.s + 1) \leq r(m_{\tau'}^{\mathfrak{A}'}) \text{ and Assumption 1})
$$

$$
\leq f_\tau^{\star, l} + \frac{1}{2^l} \sum_{l'=0}^{n} \sum_{\mathfrak{A}' \in \mathcal{S}_{l'}} \sum_{\tau' = \tau - 2^l + 1}^{\tau} r(m_{\tau'}^{\mathfrak{A}'}) \mathbb{I}\{\mathfrak{A}' \text{ is active at } \tau'\}
$$

$$
\leq f_\tau^{\star, l} + \frac{2}{2^l} \sum_{l'=0}^{n} |\mathcal{S}_{l'}| R(\min\{2^l, 2^{l'}\}),
$$

where $m_{\tau'}^{\mathfrak{A}'}$ is the number of intervals that $\mathfrak{A}'$ is active up to $\tau'$, $\mathcal{S}_{l'}$ is the set of order $l'$ base algorithms that intersect with $[\tau - 2^l + 1, \tau]$, and in the last inequality we use the fact that for any $m \geq 1$,

$$
\sum_{\tau=1}^{m} r(\tau) = \sum_{\tau=1}^{m} \min \left\{ \frac{c_1}{\sqrt{\tau}} + \frac{c_2}{\tau}, c_3 \right\} \leq \min \left\{ \sum_{\tau=1}^{m} \left( \frac{c_1}{\sqrt{\tau}} + \frac{c_2}{\tau} \right), c_3 m \right\} \leq 2R(m).
$$

For $l' \geq l$, we have $|\mathcal{S}_{l'}| \leq 2$. For $l' < l$, note that $\mathbb{E}[|\mathcal{S}_{l'}|] \leq \frac{r(2^n)}{r(2^{l'})}(2^{l-l'} + 1)$. By Lemma 50, with high probability, $|\mathcal{S}_{l'}| \leq 2\mathbb{E}[|\mathcal{S}_{l'}|] + 2^8 \ln(2M_\dagger/\delta) \leq \frac{2R(2^l)}{R(2^{l'})} + 258 \ln(2M_\dagger/\delta)$. Plugging these back, we obtain

$$
\frac{2}{2^l} \sum_{l'=0}^{n} |\mathcal{S}_{l'}| R(\min\{2^l, 2^{l'}\}) \leq \frac{2}{2^l} \sum_{l'=0}^{l-1} \left( \frac{2R(2^l)}{R(2^{l'})} + 258 \ln(2M_\dagger/\delta) \right) R(2^{l'}) + 4 \sum_{l'=l}^{n} \alpha_l \leq \widehat{\alpha}_l.
$$

This completes the proof. $\square$

Now we show the guarantee of non-stationarity detection on a single block $[m_n, E_n]$. Define $\tau_i(l)$ as the smallest interval $\tau \in \mathcal{I}_i^l \triangleq [s_i + 2^l - 1, e_i]$ ($\tau_i(l) = e_i + 1$ if such an interval does not exist) such that $\widetilde{g}_\tau^l - f_\tau^{\star,l} > 12\widehat{\alpha}_l$, and $\xi_i(l) = e_i - \tau_i(l) + 1$.

**Lemma 39.** *Let the event in [Lemma 36](#) hold. Then with high probability,*

$$\sum_{\tau=m_n}^{E_n} (C^\tau - \widetilde{g}_\tau) \le 3\widehat{R}(E_n - m_n + 1) + c_3,$$

$$\sum_{\tau=m_n}^{E_n} (\widetilde{g}_\tau - f_\tau^\star) \le \tilde{\mathcal{O}}\left(\sum_{i=1}^{\ell} \widehat{R}(|\mathcal{I}_i|)\right) + 2^{10} \sum_{l=l_0}^{n} \frac{\alpha_l}{\alpha_n} \widehat{R}(2^l) \ln(2M_\dagger/\delta).$$

*Proof.* The first statement trivially holds by **Test 2** and the estimated regret in a single interval is at most $c_3$ ([Assumption 1](#)). For the second statement, define $d_\tau^l = \widetilde{g}_\tau^l - f_\tau^{\star,l}$. For a particular $\mathcal{I}_i$ and any $l \ge 0$, let $\mathcal{I}_i' = \mathcal{I}_i \cap [\tau_i(l) - 1]$. If $|\mathcal{I}_i'| \le 2 \cdot 2^{l+1}$, then clearly $\sum_{\tau \in \mathcal{I}_i^l, \tau < \tau_i(l)} d_\tau^l \le |\mathcal{I}_i'| \cdot 12\widehat{\alpha}_l \le \min\{|\mathcal{I}_i|, 2 \cdot 2^{l+1}\} \cdot 12\widehat{\alpha}_l$. If $|\mathcal{I}_i'| > 2 \cdot 2^{l+1}$, then $\mathcal{I}_i'$ can be partitioned into three segments $\mathcal{H}_i^0 = [s_i, s_i + 2^{l+1} - 1]$, $\mathcal{H}_i^1 = [\tau_i(l) - 2^{l+1}, \tau_i(l) - 1]$, and $\mathcal{H}_i^2 = [s_i + 2^{l+1}, \tau_i(l) - 2^{l+1} - 1]$. Note that for $\tau \in \mathcal{H}_i^2$, the weight of $d_\tau^0$ within the sum $\sum_{\tau \in \mathcal{I}_i^{l+1}, \tau < \tau_i(l)} d_\tau^{l+1}$ is 1. Therefore, $\sum_{\tau \in \mathcal{H}_i^2} d_\tau^0 \le \sum_{\tau \in \mathcal{I}_i^{l+1}, \tau < \tau_i(l)} d_\tau^{l+1}$. Moreover, $\sum_{\tau \in \mathcal{H}_i^0 \cup \mathcal{H}_i^1} d_\tau^0 = 2^l(d_{s_i+2^l-1}^l + d_{s_i+2^{l+1}-1}^l + d_{\tau_i(l)-1}^l + d_{\tau_i(l)-2^l-1}^l) \le 2 \cdot 2^{l+1} \cdot 12\widehat{\alpha}_l = \min\{|\mathcal{I}_i|, 2 \cdot 2^{l+1}\} \cdot 12\widehat{\alpha}_l$. This gives

$$\sum_{\tau \in \mathcal{I}_i^l, \tau < \tau_i(l)} d_\tau^l \le \sum_{\tau \in \mathcal{I}_i'} d_\tau^0 = \left(\sum_{\tau \in \mathcal{H}_i^2} + \sum_{\tau \in \mathcal{H}_i^0 \cup \mathcal{H}_i^1}\right) d_\tau^0$$

$$\le \sum_{\tau \in \mathcal{I}_i^{l+1}, \tau < \tau_i(l)} d_\tau^{l+1} + \min\{|\mathcal{I}_i|, 2 \cdot 2^{l+1}\} \cdot 12\widehat{\alpha}_l$$

$$\le \sum_{\tau \in \mathcal{I}_i^{l+1}, \tau < \tau_i(l+1)} d_\tau^{l+1} + 12\widehat{\alpha}_l \xi_i(l+1) + \min\{|\mathcal{I}_i|, 2^l\} \cdot 48\widehat{\alpha}_l.$$

$$\left(d_\tau^{l+1} = \tfrac{1}{2}(d_\tau^l + d_{\tau-2^l}^l)\right)$$

$$\le \sum_{\tau \in \mathcal{I}_i^{l+1}, \tau < \tau_i(l+1)} d_\tau^{l+1} + 24\widehat{\alpha}_{l+1} \xi_i(l+1) + \min\{|\mathcal{I}_i|, 2^l\} \cdot 48\widehat{\alpha}_l.$$

$$\left(\widehat{\alpha}_l = \tfrac{\widehat{R}(2^l)}{2^l} \le \tfrac{2\widehat{R}(2^{l+1})}{2^{l+1}} \le 2\widehat{\alpha}_{l+1}\right)$$

Combining the two cases above, we have

$$\sum_{\tau \in \mathcal{I}_i^l, \tau < \tau_i(l)} d_\tau^l \le \sum_{\tau \in \mathcal{I}_i^{l+1}, \tau < \tau_i(l+1)} d_\tau^{l+1} + 24\widehat{\alpha}_{l+1} \xi_i(l+1) + \min\{|\mathcal{I}_i|, 2^l\} \cdot 48\widehat{\alpha}_l.$$

Applying this recursively, we have for a given $\mathcal{I}_i$,

$$\sum_{\tau \in \mathcal{I}_i} (\widetilde{g}_\tau - f_\tau^\star) = \sum_{\tau \in \mathcal{I}_i^0, \tau < \tau_i(0)} d_\tau^0 \le \sum_{\tau \in \mathcal{I}_i^n, \tau < \tau_i(n)} d_\tau^n + 24 \sum_{l=0}^{n-1} \widehat{\alpha}_{l+1} \xi_i(l+1) + 48 \sum_{l=0}^{n-1} \widehat{R}(\min\{|\mathcal{I}_i|, 2^l\})$$

$$(\min\{|\mathcal{I}_i|, 2^l\}\widehat{\alpha}_l \le \widehat{r}(\min\{|\mathcal{I}_i|, 2^l\}) \min\{|\mathcal{I}_i|, 2^l\} = \widehat{R}(\min\{|\mathcal{I}_i|, 2^l\}))$$

$$\le 12|\mathcal{I}_i|\widehat{\alpha}_n + 24 \sum_{l=1}^{n} \widehat{\alpha}_l \xi_i(l) + \tilde{\mathcal{O}}\left(\widehat{R}(|\mathcal{I}_i|)\right) \le 24 \sum_{l=1}^{n} \widehat{\alpha}_l \xi_i(l) + \tilde{\mathcal{O}}\left(\widehat{R}(|\mathcal{I}_i|)\right).$$

Summing over all $i$ and by $l < l_0 \implies 12\widehat{\alpha}_l > c_4 \implies \xi_i(l) = 0$, we have:

$$\sum_{\tau=m_n}^{E_n} (\widetilde{g}_\tau - f_\tau^\star) \le \tilde{\mathcal{O}}\left(\sum_{i=1}^{\ell} R(|\mathcal{I}_i|)\right) + 24 \sum_{l=l_0}^{n} \sum_{i=1}^{\ell} \widehat{\alpha}_l \xi_i(l).$$

Now note that for any fixed $l$, we have:

$$\sum_{i=1}^{\ell} \widehat{\alpha}_l \xi_i(l) = \widehat{\alpha}_l \sum_{i=1}^{\ell} \min\{\xi_i(l), 4 \cdot 2^l\} + \widehat{\alpha}_l \sum_{i=1}^{\ell} (\xi_i(l) - 4 \cdot 2^l)_+$$

$$\leq 4 \sum_{i=1}^{\ell} \left( \widehat{R}(|\mathcal{I}_i|) + \frac{4\alpha_l}{\alpha_n} \widehat{R}(2^l) \ln(2M_\dagger/\delta) \right).$$

(Lemma 40 and $\widehat{\alpha}_l \min\{\xi_i(l), 4 \cdot 2^l\} \leq 4\widehat{r}(\min\{\xi_i(l), 2^l\}) \min\{\xi_i(l), 2^l\} = 4\widehat{R}(\min\{\xi_i(l), 2^l\}))$
Putting everything together completes the proof. $\qquad \square$

**Lemma 40.** *For any $l \leq n$, $\sum_{i=1}^{\ell} \widehat{\alpha}_l(\xi_i(l) - 4 \cdot 2^l)_+ \leq \frac{4\alpha_l}{\alpha_n} \widehat{R}(2^l) \ln(2M_\dagger/\delta)$ with high probability.*

*Proof.* Denote by $A_l$ the number of candidate starting points of an order-$l$ algorithm in $[\tau_i(l), e_i - 2 \cdot 2^l]$ for some $i$. Note that this quantity is lower bounded by $\sum_{i=1}^{\ell} (\xi_i(l) - 4 \cdot 2^l)_+/2^l$. Moreover, if in interval $m \in [\tau_i(l), e_i - 2 \cdot 2^l]$, an order-$l$ algorithm $\mathfrak{A}$ starts, then **Test 1** is performed at $m + 2^l - 1 \leq e_i$, and **Test 1** returns *fail* with high probability because

$$\frac{1}{2^l} \sum_{\tau=\mathfrak{A}.s}^{\mathfrak{A}.e} C^\tau \leq \frac{1}{2^l} \sum_{\tau=\mathfrak{A}.s}^{\mathfrak{A}.e} \widetilde{g}_\tau + \widehat{\alpha}_l \qquad (\Delta_{[\mathfrak{A}.s, \mathfrak{A}.e]} \leq \Delta_{\mathcal{I}_i} \leq r(|\mathcal{I}_i|) \leq r(2^l) \text{ and Lemma 36})$$

$$\leq \frac{1}{2^l} \sum_{\tau=\mathfrak{A}.s}^{\mathfrak{A}.e} f_\tau^\star + 2\widehat{\alpha}_l$$

(Lemma 38, $\Delta_{\mathcal{I}_i} \leq r(|\mathcal{I}_i|)$, and $[\mathfrak{A}'.s, \mathfrak{A}'.e] \subseteq [\mathfrak{A}.s, \mathfrak{A}.e]$ if $\mathfrak{A}'$ is active within $[\mathfrak{A}.s, \mathfrak{A}.e]$)

$$\leq f_{\tau_i(l)}^{\star, l} + 2\widehat{\alpha}_l + \Delta_{\mathcal{I}_i}$$

$$\leq \widetilde{g}_{\tau_i(l)}^l - 12\widehat{\alpha}_l + 3\widehat{\alpha}_l \leq \widetilde{g}_{\tau_i(l)}^l - 9\widehat{\alpha}_l. \qquad (\Delta_{\mathcal{I}_i} \leq r(|\mathcal{I}_i|) \leq r(2^l) \leq \widehat{\alpha}_l)$$

This is a contradiction by the definition of $E_n$. Therefore, all candidate starting points of order-$l$ algorithm in $[\tau_i(l), e_i - 2 \cdot 2^l]$ does not instantiate an order-$l$ algorithm. Let $X_m = \{m \in [\tau_i(l), e_i - 2 \cdot 2^l] \text{ for some } i\}$, $X'_m = \{m \in [\tau_i(l), e_i] \text{ for some } i\}$, $Y_m = \{(m - m_n) \mod 2^l = 0\}$ and $Z_m = \{\nexists \text{ order-}l \text{ } \mathfrak{A}' \text{ such that } \mathfrak{A}'.s = m\}$, we have

$$A_l = \sum_{m=m_n}^{m_n+2^m-1} \mathbb{I}\{X_m, Y_m\} = \sum_{m=m_n}^{m_n+2^m-1} \mathbb{I}\{X_m, Y_m, Z_m\} \leq \sum_{m=m_n}^{m_n+2^m-1} \mathbb{I}\{X'_m, Y_m, Z_m\}.$$

Note that conditioned on $X'_m \cap Y_m$, the event $Z_m$ happens with a constant probability $1 - \frac{\alpha_n}{\alpha_l}$. Moreover, $Z_m = 0$ implies $X'_{m'} = 0$ for $m' > m$. Therefore, $\sum_{m=m_n}^{m_n+2^m-1} \mathbb{I}\{X'_m, Y_m, Z_m\}$ counts the number of trials up to the first success with success probability $\frac{\alpha_n}{\alpha_l}$ of each trial. Then with probability at least $1 - \delta/(2M_\dagger^2)$, we have $A_l \leq \frac{4\alpha_l}{\alpha_n} \ln(2M_\dagger/\delta)$. Thus,

$$\sum_{i=1}^{\ell} \widehat{\alpha}_l(\xi_i(l) - 4 \cdot 2^l)_+ \leq \widehat{\alpha}_l 2^l \cdot \frac{4\alpha_l}{\alpha_n} \ln(2M_\dagger/\delta) \leq \frac{4\alpha_l}{\alpha_n} \widehat{R}(2^l) \ln(2M_\dagger/\delta).$$

This completes the proof. $\qquad \square$

Now we present the regret guarantee in a single block.
**Lemma 41.** *Within a single block $\mathcal{J} = [m_n, E_n]$, we have*

$$\sum_{m \in \mathcal{J}} (C^m - f_m^\star) = \tilde{\mathcal{O}} \left( c_1 \sqrt{\ell |\mathcal{J}|} + c_2 \ell + \left( c_1 + \frac{c_2 c_4}{c_1} \right) 2^{n/2} + \frac{c_2^2}{c_3} + c_3 \right).$$

*Proof.* By Lemma 39, we have

$$\sum_{m \in \mathcal{J}} (C^m - f_m^\star) = \sum_{m \in \mathcal{J}} (C^m - \widetilde{g}_m) + \sum_{m \in \mathcal{J}} (\widetilde{g}_m - f_m^\star)$$

$$= \tilde{\mathcal{O}} \left( \widehat{R}(|\mathcal{J}|) + \sum_{i=1}^{\ell} \widehat{R}(|\mathcal{I}_i|) + \sum_{l=l_0}^{n} \frac{\alpha_l}{\alpha_n} \widehat{R}(2^l) + c_3 \right).$$

Note that by Cauchy-Schwarz inequality:

$$\widehat{R}(|\mathcal{J}|) + \sum_{i=1}^{\ell} \widehat{R}(|\mathcal{I}_i|) = \tilde{\mathcal{O}}\left(\left(c_1\sqrt{|\mathcal{J}|} + c_2\right) + \sum_{i=1}^{\ell}\left(c_1\sqrt{|\mathcal{I}_i|} + c_2\right)\right) = \tilde{\mathcal{O}}\left(c_1\sqrt{\ell|\mathcal{J}|} + c_2\ell\right).$$

Moreover, by the definition of $l_0$, we have $12\widehat{\alpha}_{l_0} = 2^{10}\widehat{n}\ln(2M_\dagger/\delta)\min\{\frac{c_1}{\sqrt{2^{l_0}}} + \frac{c_2}{2^{l_0}}, c_3\} \leq c_4$, which implies $c_2 \leq c_4 2^{l_0}$ by $c_4 \leq c_3$. Now for any $l \geq l_0$,

$$
\begin{aligned}
\frac{\alpha_l}{\alpha_n}\widehat{R}(2^l) &= \tilde{\mathcal{O}}\left(\frac{R(2^l)^2}{R(2^n)}2^{n-l}\right) = \tilde{\mathcal{O}}\left(\frac{c_1^2 2^l + c_2^2}{c_1 2^{n/2} + c_2}2^{n-l} + \frac{c_1^2 2^l + c_2^2}{c_3 2^n}2^{n-l}\right)\\
&= \tilde{\mathcal{O}}\left(c_1 2^{n/2} + \frac{c_2 c_4 2^{l_0}}{c_1}2^{n/2-l} + \frac{c_1^2}{c_3} + \frac{c_2^2}{c_3}2^{-l}\right)\\
&= \tilde{\mathcal{O}}\left(\left(c_1 + \frac{c_2 c_4}{c_1}\right)2^{n/2} + \frac{c_1^2}{c_3} + \frac{c_2^2}{c_3}2^{-l}\right) = \tilde{\mathcal{O}}\left(\left(c_1 + \frac{c_2 c_4}{c_1}\right)2^{n/2} + \frac{c_2^2}{c_3}2^{-l}\right),
\end{aligned}
$$

where in the last inequality we assume $c_1 \leq c_3 2^{n/2}$ without loss of generality and have $\frac{c_1^2}{c_3} \leq c_1 2^{n/2}$ (note that if $c_1 > c_3 2^{n/2}$, then $c_1 2^{n/2} > c_3 2^n$ and the regret bound is vacuous). Summing over $l$ and putting everything together, we obtain:

$$\sum_{m\in\mathcal{J}}(C^m - f_m^\star) = \tilde{\mathcal{O}}\left(c_1\sqrt{\ell|\mathcal{J}|} + c_2\ell + \left(c_1 + \frac{c_2 c_4}{c_1}\right)2^{n/2} + \frac{c_2^2}{c_3} + c_3\right).$$

$\square$

### F.5 Single Epoch Regret Analysis

We call $[m_0, E]$ an *epoch* if $m_0$ is the first interval after restart from Line 1 or $m_0 = 1$, and $E$ is the first interval where a restart after interval $m$ is triggered. The regret guarantee in a single epoch is shown in the following lemma.

**Lemma 42.** *Let $\mathcal{E}$ be an epoch, then $\sum_{m\in\mathcal{E}}(C^m - f_m^\star) = \tilde{\mathcal{O}}(c_1\sqrt{\ell_{\mathcal{E}}|\mathcal{E}|} + c_2\ell_{\mathcal{E}} + (c_1 + \frac{c_2 c_4}{c_1})\sqrt{|\mathcal{E}|} + \frac{c_2^2}{c_3} + c_3)$, where $\ell_{\mathcal{E}} = \tilde{\mathcal{O}}(1 + (c_1^{-1}\Delta_{\mathcal{E}})^{2/3}|\mathcal{E}|^{1/3} + c_3^{-1}\Delta_{\mathcal{E}})$ and $\ell_{\mathcal{E}} = \tilde{\mathcal{O}}(L_{\mathcal{E}})$.*

*Proof.* Suppose $\mathcal{E}$ consists of blocks $\mathcal{J}_1, \ldots, \mathcal{J}_n$ and the number of near stationary segments (as discussed in Appendix F.4) in $\mathcal{J}_i$ is $\ell_i$. Then, $|\mathcal{E}| = \Theta(2^n)$, and by Lemma 41 and Cauchy-Schwarz inequality,

$$
\begin{aligned}
\sum_{m\in\mathcal{E}}(C^m - f_m^\star) &= \tilde{\mathcal{O}}\left(c_1\sum_{i=1}^{n}\sqrt{\ell_i|\mathcal{J}_i|} + c_2\sum_{i=1}^{n}\ell_i + \left(c_1 + \frac{c_2 c_4}{c_1}\right)2^{n/2} + \frac{c_2^2}{c_3} + c_3\right)\\
&= \tilde{\mathcal{O}}\left(c_1\sqrt{\sum_{i=1}^{n}\ell_i|\mathcal{E}|} + c_2\sum_{i=1}^{n}\ell_i + \left(c_1 + \frac{c_2 c_4}{c_1}\right)\sqrt{|\mathcal{E}|} + \frac{c_2^2}{c_3} + c_3\right).
\end{aligned}
$$

Finally by Lemma 37 and Hölder's inequality, $\sum_{i=1}^{n}\ell_i = \tilde{\mathcal{O}}(1 + (c_1^{-1}\Delta_{\mathcal{E}})^{2/3}|\mathcal{E}|^{1/3} + c_3^{-1}\Delta_{\mathcal{E}})$ and $\sum_{i=1}^{n}\ell_i = \tilde{\mathcal{O}}(L_{\mathcal{E}})$. $\square$

### F.6 Full Regret Guarantee

To derive the full regret guarantee of the MASTER algorithm (Algorithm 8), we first bound the number of epochs by the following two lemmas. Define $\mathfrak{N}_{[1,M']}$ as the number of times MALG terminates within $[1, M']$

**Lemma 43.** *Let $m$ be in an epoch starting from interval $m_0$. If $\Delta_{[m_0,m]} \leq r(m - m_0 + 1)$, then no restart would be triggered by **Test 1** or **Test 2** in interval $m$ with high probability.*

*Proof.* We first show that **Test 1** would not fail. Let $m = \mathfrak{A}.e$ where $\mathfrak{A}$ is any order-$l$ base algorithm in a block of order $n$ starting from $m_n$. Then with high probability,

$$
\begin{aligned}
U_m^l &= \max_{\tau \in [m_n + 2^l - 1, m]} \widetilde{g}_\tau^l \leq \max_{\tau \in [m_n + 2^l - 1, m]} f_\tau^{\star, l} + \widehat{r}(2^l) \qquad \text{(Lemma 38)} \\
&\leq \frac{1}{2^l} \sum_{\tau = \mathfrak{A}.s}^{m} f_\tau^\star + \widehat{r}(2^l) + \Delta_{[m_n, m]} \\
&\leq \frac{1}{2^l} \sum_{\tau = \mathfrak{A}.s}^{m} C^\tau + 2\widehat{r}(2^l) + \Delta_{[m_n, m]} \leq \frac{1}{2^l} \sum_{\tau = \mathfrak{A}.s}^{m} C^\tau + 3\widehat{r}(2^l).
\end{aligned}
$$
$$\text{(Lemma 36 and } \Delta_{[m_n, m]} \leq \Delta_{[m_0, m]} \leq r(m - m_0 + 1) \leq r(2^l))$$

Thus, **Test 1** would not fail. For **Test 2**, by Lemma 36 and $\Delta_{[m_n, m]} \leq \Delta_{[m_0, m]} \leq r(m - m_0 + 1) \leq r(m - m_n + 1)$:

$$
\sum_{\tau = m_n}^{m} (C^\tau - \widetilde{g}_\tau) \leq \widehat{R}(m - m_n + 1),
$$

Thus, **Test 2** also would not fail. $\qquad\square$

**Lemma 44.** *Assuming that MALG does not terminate without non-stationarity, with high probability, the number of epochs within $[1, M']$ is upper bounded by $L_{[1, M']}$ and $1 + (2c_1^{-1}\Delta_{[1, M']})^{2/3} M'^{1/3} + c_3^{-1}\Delta_{[1, M']} + \mathfrak{N}_{[1, M']}$.*

*Proof.* The first upper bound is clearly true by partitioning $[1, M']$ into segments without non-stationarity. For the second upper bound, by Lemma 43, if an epoch $[m_0, E]$ is not the last epoch, then $\Delta_{[m_0, E]} > r(E - m_0 + 1)$ or MALG terminates with high probability. Applying Lemma 13 completes the proof. $\qquad\square$

**Theorem 13.** *If Assumption 1 holds, then MASTER (Algorithm 8) ensures with high probability (ignoring lower order terms), for any $M' \geq 1$:*

$$
\widetilde{R}_{M'} = \tilde{\mathcal{O}}\left( \left( c_1 + \frac{c_2 c_4}{c_1} \right) \sqrt{L_{[1, M']} M'} \right) \text{ and}
$$
$$
\widetilde{R}_{M'} = \tilde{\mathcal{O}}\left( \left( c_1 + \frac{c_2 c_4}{c_1} \right) \sqrt{(\mathfrak{N}_{[1, M']} + 1) M'} + \left( c_1^{2/3} + \frac{c_2 c_4}{c_1^{4/3}} \right) \Delta_{[1, M']}^{1/3} M'^{2/3} \right).
$$

*Proof.* Let $\mathcal{E}_1, \ldots, \mathcal{E}_N$ be epochs in $[1, M']$ and $\mathcal{E} = \bigcup_{i=1}^{N} \mathcal{E}_i$. Then by Lemma 42 and Cauchy-Schwarz inequality, we have:

$$
\begin{aligned}
\widetilde{R}_{M'} &= \tilde{\mathcal{O}}\left( \sum_{i=1}^{N} \left( c_1 \sqrt{\ell_{\mathcal{E}_i} |\mathcal{E}_i|} + c_2 \ell_{\mathcal{E}_i} + \left( c_1 + \frac{c_2 c_4}{c_1} \right) \sqrt{|\mathcal{E}_i|} + \frac{c_2^2}{c_3} + c_3 \right) \right) \\
&= \tilde{\mathcal{O}}\left( c_1 \sqrt{\ell_{\mathcal{E}} M'} + c_2 \ell_{\mathcal{E}} + \left( c_1 + \frac{c_2 c_4}{c_1} \right) \sqrt{N M'} + \left( \frac{c_2^2}{c_3} + c_3 \right) N \right),
\end{aligned}
$$

where $\ell_{\mathcal{E}} = \sum_{i=1}^{N} \ell_{\mathcal{E}_i}$. Below we assume sub-linear $L_{[1, M']}, \Delta_{[1, M']}$ and only write down dominating terms. For $L$-dependent bound, note that $N \leq L_{[1, M']}$ by Lemma 44 and $\ell_{\mathcal{E}} \leq N + L_{[1, M']} = \tilde{\mathcal{O}}(L_{[1, M']})$ by Lemma 42. Thus, $c_2 \ell_{\mathcal{E}} + (\frac{c_2^2}{c_3} + c_3) N$ is a lower order term, and

$$
\widetilde{R}_{M'} = \tilde{\mathcal{O}}\left( \left( c_1 + \frac{c_2 c_4}{c_1} \right) \sqrt{L_{[1, M']} M'} \right).
$$

For $\Delta$-dependent bound, note that by Lemma 42, Hölder's inequality, and Lemma 44,

$$
\begin{aligned}
\ell_{\mathcal{E}} &= \tilde{\mathcal{O}}\left( N + (c_1^{-1}\Delta_{[1, M']})^{2/3} M'^{1/3} + c_3^{-1}\Delta_{[1, M']} \right) \\
&= \tilde{\mathcal{O}}\left( \mathfrak{N}_{[1, M']} + 1 + (c_1^{-1}\Delta_{[1, M']})^{2/3} M'^{1/3} + c_3^{-1}\Delta_{[1, M']} \right).
\end{aligned}
$$

Ignoring lower order term of the form $\sqrt{\Delta_{[1,M']}M'}$, we have

$$c_1\sqrt{\ell_\mathcal{E}M'} + \left(c_1 + \frac{c_2 c_4}{c_1}\right)\sqrt{NM'}$$

$$= \tilde{\mathcal{O}}\left(\left(c_1 + \frac{c_2 c_4}{c_1}\right)\sqrt{\left(\mathfrak{N}_{[1,M']} + 1 + (c_1^{-1}\Delta_{[1,M']})^{2/3}M'^{1/3} + c_3^{-1}\Delta_{[1,M']}\right)M'}\right)$$

$$= \tilde{\mathcal{O}}\left(\left(c_1 + \frac{c_2 c_4}{c_1}\right)\sqrt{(\mathfrak{N}_{[1,M']} + 1)M'} + \left(c_1^{2/3} + \frac{c_2 c_4}{c_1^{4/3}}\right)\Delta_{[1,M']}^{1/3}M'^{2/3}\right).$$

The remaining $c_2\ell_\mathcal{E} + (\frac{c_2^2}{c_3} + c_3)N$ is again a lower order term. $\qquad\square$

### F.7 Proof of Theorem 6

We are ready to present the regret guarantee of the MASTER algorithm combining with different base algorithms. Recall $L = 1 + \sum_{k=1}^{K-1}\mathbb{I}\{P_{k+1} \neq P_k \text{ or } c_{k+1} \neq c_k\}$.

**Theorem 14.** *Let $\mathfrak{A}$ be Algorithm 8 with Algorithm 2 as base algorithm. Then Algorithm 1 with $\mathfrak{A}$ ensures with high probability, for any $K' \in [K]$,*

$$R_{K'} = \tilde{\mathcal{O}}\left(\min\left\{B_\star S\sqrt{ALK'}, B_\star S\sqrt{AK'} + (B_\star^2 S^2 A(\Delta_c + B_\star\Delta_P)T_{\max})^{1/3}K'^{2/3}\right\}\right).$$

*Proof.* By Lemma 33 and Theorem 13 with $\mathfrak{N}_{[1,M]} = 0$, we have for any $M' \leq M$,

$$\mathring{R}_{M'} \leq \widetilde{R}_{M'} = \tilde{\mathcal{O}}\left(\min\left\{B_\star S\sqrt{AL_{[1,M]}M'}, B_\star S\sqrt{AM'} + (B_\star^2 S^2 A\Delta_{[1,M]})^{1/3}M'^{2/3}\right\}\right),$$

where $L_{[1,M]} = L$ and $\Delta_{[1,M]} = \tilde{\mathcal{O}}((\Delta_c + B_\star\Delta_P)T_{\max})$. Applying Lemma 16, we have for any $K' \in [K]$ (ignoring lower order terms),

$$\mathring{R}_{M_{K'}} = \tilde{\mathcal{O}}\left(\min\left\{B_\star S\sqrt{ALK'}, B_\star S\sqrt{AK'} + (B_\star^2 S^2 A(\Delta_c + B_\star\Delta_P)T_{\max})^{1/3}K'^{2/3}\right\}\right).$$

Applying Lemma 15 completes the proof. $\qquad\square$

We are now ready to prove Theorem 6.

*Proof of Theorem 6.* By Lemma 30 and Lemma 31, when Algorithm 6 terminates in interval $E$ where $[m_0, E]$ is an epoch, we have $\Delta'_{[m_0, E+1]} > \eta_{E-m_0+2}$. Therefore, $\mathfrak{N}_{[1,K]} = \tilde{\mathcal{O}}(1 + (B_\star^2 S^2 A)^{-1/3}(T_\star\Delta'_{[1,K]})^{2/3}K^{1/3} + H\Delta'_{[1,K]})$ by Lemma 13 and the definition of $\eta_m$. Then by Lemma 34 and Theorem 13, we have $\mathfrak{A}_1$ ensures when $s_1^m = s_{\text{init}}$ for $m \leq K$,

$$\mathring{R}_K = \widetilde{R}_K = \tilde{\mathcal{O}}\left(\min\left\{B_\star S\sqrt{ALK}, B_\star S\sqrt{AK} + (B_\star^2 S^2 A(\Delta_c + B_\star\Delta_P)T_\star)^{1/3}K^{2/3}\right\}\right),$$

where we apply $\Delta'_{[1,K]} = \tilde{\mathcal{O}}(\Delta_c + B_\star\Delta_P)$, $L_{[1,K]} = L$, and $\Delta_{[1,K]} = \tilde{\mathcal{O}}((\Delta_c + B_\star\Delta_P)T_\star)$. Moreover, by Theorem 14, $\mathfrak{A}_2$ ensures $R_{K'}$ being sub-linear w.r.t $K'$ for any $K' \in [K]$. Applying Theorem 11 completes the proof. $\qquad\square$

## G  Auxiliary Lemmas

**Lemma 45.** *[Chen et al., 2022b, Lemma 48] $x \leq a\sqrt{x} + b$ implies $x \leq (a + \sqrt{b})^2 \leq 2a^2 + 2b$.*

**Lemma 46.** *[Rosenberg and Mansour, 2021, Lemma 6] Let $\pi$ be a policy whose expected hitting time starting from any state is at most $\tau$. Then for any $\delta \in (0,1)$, with probability at least $1 - \delta$, it takes no more than $4\tau\ln\frac{2}{\delta}$ steps to reach the goal state following $\pi$.*

**Lemma 47.** *[Chen et al., 2021a, Lemma 30] For any random variable $X$ with $\|X\|_\infty \leq C$, we have $\text{VAR}[X^2] \leq 4C^2\text{VAR}[X]$.*

**Lemma 48.** *([Chen et al., 2021a, Lemma 31]) Define $\Upsilon = \{v \in [0,B]^{\mathcal{S}_+} : v(g) = 0\}$. Let $f : \Delta_{\mathcal{S}_+} \times \Upsilon \times \mathbb{R}^+ \times \mathbb{R}^+ \times \mathbb{R}^+ \to \mathbb{R}^+$ with $f(p,v,n,B,\iota) = pv - \max\left\{c_1\sqrt{\frac{\mathbb{V}(p,v)\iota}{n}}, c_2\frac{B\iota}{n}\right\}$ with $c_1^2 \leq c_2$. Then $f$ satisfies for all $p \in \Delta_{\mathcal{S}_+}, v \in \Upsilon$ and $n, \iota > 0$,*

1. *$f(p,v,n,B,\iota)$ is non-decreasing in $v(s)$, that is,*

$$\forall v, v' \in \Upsilon, v(s) \leq v'(s), \forall s \in \mathcal{S}^+ \implies f(p,v,n,B,\iota) \leq f(p,v',n,B,\iota);$$

2. *$f(p,v,n,B,\iota) \leq pv - \frac{c_1}{2}\sqrt{\frac{\mathbb{V}(p,v)\iota}{n}} - \frac{c_2}{2}\frac{B\iota}{n}$.*

**Lemma 49** (Any interval Freedman's inequality). *Let $\{X_i\}_{i=1}^{\infty}$ be a martingale difference sequence w.r.t the filtration $\{\mathcal{F}_i\}_{i=0}^{\infty}$ and $|X_i| \leq B$ for some $B > 0$. Then with probability at least $1 - \delta$, for all $1 \leq l \leq n$ simultaneously,*

$$\left|\sum_{i=l}^{n} X_i\right| \leq 3\sqrt{\sum_{i=l}^{n}\mathbb{E}[X_i^2|\mathcal{F}_{i-1}]\ln\frac{16B^2n^5}{\delta}} + 2B\ln\frac{16B^2n^5}{\delta} \tag{5}$$

$$\leq 3\sqrt{2\sum_{i=l}^{n}X_i^2\ln\frac{16B^2n^5}{\delta}} + 18B\ln\frac{16B^2n^5}{\delta}. \tag{6}$$

*Proof.* For each $l \geq 1$, by [Chen et al., 2022a, Lemma 38], with probability at least $1 - \frac{\delta}{4l^2}$, Eq. (5) holds for all $n \geq l$. Then by Lemma 50, with probability at least $1 - \frac{\delta}{4l^2}$, Eq. (6) holds for all $n \geq l$. Applying a union bound over $l$ completes the proof. $\square$

**Lemma 50.** *Suppose $\{X_i\}_{i=1}^{\infty}$ is a sequence of random variables w.r.t the filtration $\{\mathcal{F}_i\}_{i=0}^{\infty}$ and satisfies $X_i \in [0,B]$ for some $B > 0$. Then with probability at least $1 - \delta$, for all $1 \leq l \leq n$ simultaneously,*

$$\sum_{i=l}^{n}\mathbb{E}[X_i|\mathcal{F}_{i-1}] \leq 2\sum_{i=l}^{n}X_i + 12B\ln\frac{2n}{\delta},$$

$$\sum_{i=l}^{n}X_i \leq 2\sum_{i=l}^{n}\mathbb{E}[X_i|\mathcal{F}_{i-1}] + 24B\ln\frac{2n}{\delta}.$$

*Proof.* For each $l \geq 1$, by [Chen et al., 2022a, Lemma 39], with probability at least $1 - \frac{\delta}{2l^2}$, the two inequalities above hold for all $n \geq l$. Taking a union bound over $l$ completes the proof. $\square$