# OpenReview forum: "Near-Optimal Goal-Oriented Reinforcement Learning in Non-Stationary Environments"
_NeurIPS.cc/2022/Conference — NeurIPS 2022 Accept_

### Official Review · Reviewer_Cqin · 2022-07-10

**Rating:** 7
**Confidence:** 4
**Soundness:** 4 excellent
**Presentation:** 4 excellent
**Contribution:** 3 good

**Summary:**

This paper initiates the study of learning stochastic shortest path in the nonstationary tabular setting. Specifically, this paper assumes the cost function and the transition kernel vary with respect to the time. It considers the case where the magnitudes of the total change in the cost and transition functions are bounded. Under such a setting, this paper proves several results. First, a lower bound that depends on the magnitudes of the total change of the cost and transition functions. Then it proposed an algorithm that is nearly optimal.


**Questions:**

I have one high level question: Do you think there is any fundamental difference between using finite-horizon approximation and using infinite-horizon-discounted approximation (e.g. [1] ) to solve SSP?
It would be good if the authors can give some thoughts.

[1]: Stochastic Shortest Path: Minimax, Parameter-Free and Towards Horizon-Free Regret


**Limitations:**

No special concerns.

**Strengths And Weaknesses:**

Originality: As far as I know, this paper is the first to consider the nonstationary SSP problem and gives a regret lower bound and regret upper bound. The main algorithm borrows ideas from some well-known existing RL algorithms, but also faces several unique challenges. Still I think t

Quality: The quality is high. The problem is well-defined, and the mathematics looks solid to me.

Clarity: The writing is good. Despite being a very technical paper, the main text is easy to follow.

Significance: SSP has been a quite important problem in RL in these two years. This paper makes a solid contribution to this area by considering a new problem setting (non-stationary change in cost and transition with bounded magnitude) and gives both a lower and upper bounds.

---

> ### Author Response · Authors · 2022-07-31
> **To Reviewer Cqin**
>
> **I have one high level question: Do you think there is any fundamental difference between using finite-horizon approximation and using infinite-horizon-discounted approximation (e.g. [1] ) to solve SSP? It would be good if the authors can give some thoughts.**
> That's a great question. Finite-horizon approximation and infinite-horizon approximation indeed have some crucial differences. First, finite-horizon approximation enjoys smaller approximation error as the error shrinks exponentially w.r.t the horizon (see Lemma 6 of [2]), while the approximation error of infinite-horizon approximation only goes down linearly w.r.t the discounting factor (see Lemma 14 of [1]). As a result, finite-horizon approximation often leads to optimal regret guarantee [3, 4], while infinite-horizon-discounted approximation is often suboptimal [5] unless it achieves horizon-free regret bound as in [1]. The advantage of infinite-horizon-discounted approximation, however, is that it gives stationary policies, while the finite-horizon approximation often gives non-stationary policies and increases the space complexity. A recent paper [6] proposes a new approximation scheme that enjoys small approximation error and produces nearly stationary policies, which could possibly also be applied to our setting. Their Section 3 also provides a detailed comparison between finite-horizon approximation and infinite-horizon-discounted approximation.
>
> [1] Tarbouriech, Jean, Runlong Zhou, Simon S. Du, Matteo Pirotta, Michal Valko, and Alessandro Lazaric. "Stochastic shortest path: Minimax, parameter-free and towards horizon-free regret."
>
> [2] Rosenberg, Aviv, and Yishay Mansour. "Stochastic shortest path with adversarially changing costs."
>
> [3] Chen, Liyu, Haipeng Luo, and Chen-Yu Wei. "Minimax regret for stochastic shortest path with adversarial costs and known transition."
>
> [4] Chen, Liyu, Mehdi Jafarnia-Jahromi, Rahul Jain, and Haipeng Luo. "Implicit finite-horizon approximation and efficient optimal algorithms for stochastic shortest path."
>
> [5] Wei, Chen-Yu, Mehdi Jafarnia Jahromi, Haipeng Luo, Hiteshi Sharma, and Rahul Jain. "Model-free reinforcement learning in infinite-horizon average-reward markov decision processes."
>
> [6] Chen, Liyu, Haipeng Luo, and Aviv Rosenberg. "Policy Optimization for Stochastic Shortest Path."

---

### Official Review · Reviewer_rKHa · 2022-07-11

**Rating:** 6
**Confidence:** 4
**Soundness:** 3 good
**Presentation:** 2 fair
**Contribution:** 3 good

**Summary:**

This paper studies dynamic regret minimization for goal-oriented reinforcement learning modeled by a non-stationary stochastic shortest path problem with changing cost and transition functions.

**Questions:**

Please see above.

**Limitations:**

Please see above.

**Strengths And Weaknesses:**

The paper extends the results of [Wei et al., 2021], with dynamism in amount of changes of the cost and transition functions.

The key weakness is that the model difference as compared to [Wei et al., 2021] is not explained, and the novelty as compared to [Wei et al., 2021] is not explained.

---

> ### Author Response · Authors · 2022-07-31
> **To reviewer rKHa**
>
> **The key weakness is that the model difference as compared to [Wei et al., 2021] is not explained, and the novelty as compared to [Wei et al., 2021] is not explained.**
>
> 1. Model difference: [Wei et al., 2021] considers a general reinforcement learning framework where the learner is given a policy set $\Pi$ and the environment decides $K$ reward functions $f_1,\ldots,f_K:\Pi\rightarrow[0,1]$ obliviously before learning starts. Then in each episode $k=1,\ldots,K$, the learner chooses a policy $\pi_k$ and receives a noisy reward $R_k$ with mean $f_k(\pi_k)$. The goal of the learner is to minimize the dynamic regret defined as $\sum_k(\max_{\pi\in\Pi} f_k(\pi) - R_k)$.
> Note that the non-stationary SSP problem that we study does not fall into this framework, as the learner may follow several different policies within each episode. This is necessary since under unknown and changing transition, the learner may not be able to identify a proper policy (which reaches the goal state with probability $1$) at the beginning of an episode, and committing to a single policy within an episode may lead to infinite regret. We resolve this issue by finite-horizon approximation, which enables us to commit to a single policy within each episode of the finite-horizon MDP. Note that even after the finite-horizon approximation, the problem still does not fall into the framework of [Wei et al., 2021], as both the total number of intervals and the cost (transition) function depend on the interaction history and are thus not obliviously chosen. In the revision, we will emphasize these differences in Section 4 along with the discussion of finite-horizon approximation.
> 2. Novelty: for unknown non-stationarity, our Algorithm 8 is a variant of the MASTER algorithm proposed in [Wei et al., 2021]. As described in the paragraph starting from line 289, the novelty of this algorithm is that it adopts a different Test 1, which is essential for the usage of correction term in the base algorithm (Algorithm 6). Specifically, we maintain multiple running averages of estimated value functions with different scales, and each of them detects different levels of non-stationarity. This requires a new analysis on bounding the regret on a single block (Lemma 39 in contrast to Lemma 16 of [Wei et al., 2021]). Another novelty of our analysis is that our Lemma 40 generalizes Lemma 17 of [Wei et al., 2021] so that we can bound the cost of non-stationarity detection even under adaptive adversary. We will emphasize these points in our revision.
> Finally, we also develop several new techniques to achieve minimax optimal regret under known non-stationarity (Algorithm 4), such as separate non-stationarity tests for cost and transition functions, the combination of correction term and non-stationarity tests, a novel set of reset rules, etc. All of these are new compared to [Wei et al., 2021].

---

### Official Review · Reviewer_DAsW · 2022-07-15

**Rating:** 7
**Confidence:** 4
**Soundness:** 3 good
**Presentation:** 3 good
**Contribution:** 4 excellent

**Summary:**

This paper studies dynamic regret minimization for goal-oriented reinforcement learning modeled by a non-stationary stochastic shortest path problem with changing cost and transition functions. They propose several algorithms, with corresponding upper bounds, and lower bounds for this setting.

**Questions:**

Can the results be applied to more general settings, for example linear MDP?

**Limitations:**

The authors have adequately addressed the limitations and potential negative societal impact of their work.

**Strengths And Weaknesses:**

The problem this paper studies is novel and of great significance for RL theory. And the theoretical results and analysis are sound and complete. Overall, this is a good paper for RL theory.

---

> ### Author Response · Authors · 2022-07-31
> **To reviewer DAsW**
>
> **Can the results be applied to more general settings, for example linear MDP?** This is a great question. Some components of our proposed algorithms can be directly applied to linear MDP, such as finite-horizon approximation [1], periodic restart [2], and non-stationarity tests [3]. With these components, it is already possible to develop an algorithm that achieves $\widetilde{O}(\Delta^{1/3}K^{2/3})$ dynamic regret (but maybe sub-optimal in other parameters). Parts requiring more thinking include how to inspect the norm of estimated value functions (used in adaptive confidence widening of Algorithm 2 and Test 3 of Algorithm 4), and how to construct variance-aware confidence bound. We believe that studying non-stationary SSP under linear or even general function approximation is an important future direction.
>
> [1] Chen, Liyu, Rahul Jain, and Haipeng Luo. "Improved no-regret algorithms for stochastic shortest path with linear MDP."
>
> [2] Zhou, Huozhi, Jinglin Chen, Lav R. Varshney, and Ashish Jagmohan. "Nonstationary reinforcement learning with linear function approximation."
>
> [3] Wei, Chen-Yu, and Haipeng Luo. "Non-stationary reinforcement learning without prior knowledge: An optimal black-box approach."

---

### Meta-Review · Area_Chair_B9qN · 2022-08-31

**Recommendation:** Accept
**Confidence:** Certain

**Metareview:**

This paper studies dynamic regret minimization for goal-oriented reinforcement learning modeled by a non-stationary stochastic shortest path problem with changing cost and transition functions. This paper proposes several algorithms, with nearly matching upper bounds and lower bounds when Delta_c, and Delta_p are known, and a slightly suboptimal dynamic regret when they are not known. All reviewers are convinced by the contribution of this paper, and I recommend acceptance.

**Award:**

No

---

### Decision · Program_Chairs · 2022-09-14

Accept